# Comparative Plant Transcriptome Profiling of *Arabidopsis thaliana* Col-0 and *Camelina sativa* var. *Celine* Infested with *Myzus persicae* Aphids Acquiring Circulative and Noncirculative Viruses Reveals Virus- and Plant-Specific Alterations Relevant to Aphid Feeding Behavior and Transmission

Quentin Chesnais,[a] Victor Golyaev,[b] Amandine Velt,[a] Camille Rustenholz,[a] Véronique Brault,[a] Mikhail M. Pooggin,[b] Martin Drucker[a]

[a]SVQV, UMR1131, INRAE Centre Grand Est–Colmar, Université Strasbourg, Strasbourg, France
[b]PHIM Plant Health Institute, Université Montpellier, INRAE, CIRAD, IRD, Institut Agro, Montpellier, France

Quentin Chesnais and Victor Golyaev contributed equally to this article. Co-first authors are listed in alphabetical order.

**ABSTRACT** Evidence is accumulating that plant viruses alter host plant traits in ways that modify their insect vectors' behavior. These alterations often enhance virus transmission, which has led to the hypothesis that these effects are manipulations caused by viral adaptation. However, we lack a mechanistic understanding of the genetic basis of these indirect, plant-mediated effects on vectors, their dependence on the plant host, and their relation to the mode of virus transmission. Transcriptome profiling of *Arabidopsis thaliana* and *Camelina sativa* plants infected with turnip yellows virus (TuYV) or cauliflower mosaic virus (CaMV) and infested with the common aphid vector *Myzus persicae* revealed strong virus- and host-specific differences in gene expression patterns. CaMV infection caused more severe effects on the phenotype of both plant hosts than did TuYV infection, and the severity of symptoms correlated strongly with the proportion of differentially expressed genes, especially photosynthesis genes. Accordingly, CaMV infection modified aphid behavior and fecundity more strongly than did infection with TuYV. Overall, infection with CaMV, relying on the noncirculative transmission mode, tends to have effects on metabolic pathways, with strong potential implications for insect vector-plant host interactions (e.g., photosynthesis, jasmonic acid, ethylene, and glucosinolate biosynthetic processes), while TuYV, using the circulative transmission mode, alters these pathways only weakly. These virus-induced deregulations of genes that are related to plant physiology and defense responses might impact both aphid probing and feeding behavior on infected host plants, with potentially distinct effects on virus transmission.

**IMPORTANCE** Plant viruses change the phenotype of their plant hosts. Some of the changes impact interactions of the plant with insects that feed on the plants and transmit these viruses. These modifications may result in better virus transmission. We examine here the transcriptomes of two plant species infected with two viruses with different transmission modes to work out whether there are plant species-specific and transmission mode-specific transcriptome changes. Our results show that both are the case.

**KEYWORDS** caulimovirus, polerovirus, aphid vector, transmission, feeding behavior, insect-plant interactions, transcriptome profiling, RNA-seq, plant viruses

Address correspondence to Martin Drucker, martin.drucker@inrae.fr, or Mikhail M. Pooggin, mikhail.pooggin@inrae.fr.

The authors declare no conflict of interest.

**M**ost known plant viruses rely on vectors for transmission to a new host (for an example, see reference 1). Insects that feed on plant phloem sap, such as whiteflies and aphids, are important vectors transmitting at least 500 virus species (2). The high virus

transmission capacity is due to their particular nondestructive feeding behavior that allows virus acquisition from, and inoculation into, the cytoplasm and/or the phloem sap of a new host plant. In fact, aphids alighting on a new plant will first evaluate the potential host for suitability by exploratory intracellular punctures into the epidermis and underlying tissues. If the plant is accepted, aphids plunge their needle-like mouthparts, the so-called stylets, for prolonged feeding phases into the sieve cells whose sap constitutes their principal nutritive source (for a review, see reference 3). Aphids secrete different saliva types during both the probing and feeding phases that contain effector molecules controlling interactions with the plant and susceptibility (4).

Viruses are classified according to two principal modes of transmission (for a review, see reference 5). Circulative viruses such as turnip yellows virus (TuYV; genus *Polerovirus*) are acquired by aphid vectors from the phloem sap of infected plants. They bind to specific receptors on the intestine epithelium (6), traverse the intestine, and cycle through the hemocoel to subsequently reach and invade the salivary glands (7). New hosts are inoculated when viruliferous aphids migrate between plants and inoculate the virus during salivation phases into the phloem, the only tissue where TuYV and many other circulative viruses are able to replicate. For this mode of transmission, virus acquisition and inoculation periods are rather long (in the range of several hours to days), requiring that aphids settle for sustained periods on the plants. On the other hand, despite the fact that poleroviruses do not seem to replicate in the vector, aphids having acquired poleroviruses remain infectious during their entire life span. Therefore, this transmission mode is also referred to as persistent transmission.

The transmission mode of noncirculative viruses such as cucumber mosaic virus (CMV, genus *Cucumovirus*), which are also transmitted by aphids and other hemipteran vectors, is entirely different (for a review, see reference 8). They are mostly acquired and inoculated during early probing phases (i.e., intracellular penetrations in the epidermis and mesophyll tissues [9]). They do not invade aphid cells but are retained externally in the mouthparts (stylets and/or esophagus), from where they are also released into a new host. For this reason, noncirculative viruses are acquired and inoculated within seconds to minutes, and vectors retain and transmit the virus only for a limited time (minutes range). Therefore, this transmission mode is also named nonpersistent transmission. Some noncirculative viruses may be retained by the vectors for several hours and are referred to as semipersistent viruses. The aphid-transmitted cauliflower mosaic virus (CaMV; genus *Caulimovirus*) belongs to this group (10, 11).

Available data indicate that many viruses modify host traits (i.e., color, volatiles, primary/secondary metabolites, etc.) in ways that are conducive to their transmission (3, 12, 13). Theoretical considerations suggest that these modifications depend on the virus species and, in particular, on the mode of virus transmission by vectors (3, 13). Circulative, persistent, and phloem-restricted viruses should benefit from faster access of vectors to the phloem and longer feeding, which would promote both virus acquisition and inoculation. In addition, these viruses would tend to improve the nutrient quality of the host and, consequently, vector fitness and fecundity, concomitant with an increased number of viruliferous vectors (3, 12, 14). Both modifications have been reported for aphid-transmitted luteoviruses (15). Nonpersistent or semipersistent noncirculative viruses with their fast transmission kinetics are expected to benefit from the attraction of vectors to infected plants, followed by a rapid dispersion, before the virus is lost from the vector during subsequent salivation events. The best-studied example is CMV, where altered volatiles incite aphids to alight on infected plants and acquire CMV before the poor taste and low nutritive value encourage the aphids to leave and transmit the virus to other (healthy) host plants (16–18).

While there is overwhelming evidence that some viruses do induce plant phenotype manipulation in ways that are conducive to their own transmission, many significant knowledge gaps remain (discussed in reference 19). In particular, the mechanisms

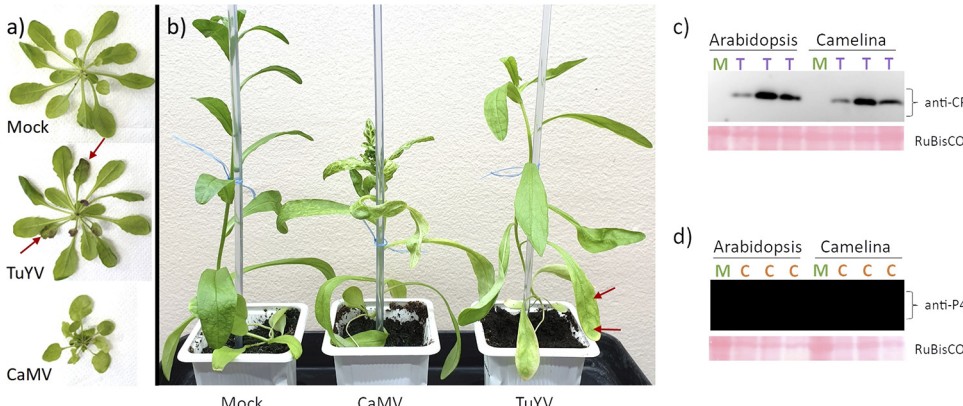

**FIG 1** Phenotypes of CaMV- and TuYV-infected plants and analysis of viral load. (a and b) *A. thaliana* Col-0 (a) and *C. sativa* var. *Celine* (b) plants 21 days after inoculation with the indicated virus or after mock inoculation. The red arrows point to purple-colored leaves in TuYV-infected *A. thaliana* Col-0 (a) and a yellowed leaf in TuYV-infected *C. sativa* var. *Celine* (b), respectively. (c and d) Western blot analysis of TuYV CP coat protein (c) and CaMV coat protein P4 (d). On each lane, a total extract from a different plant was loaded. Ponceau staining of the small ribulose 1,5-bisphosphate carboxylase/oxygenase (RubisCO) subunit is shown as a loading control. M, mock inoculated; T, TuYV infected; C, CaMV infected.

and pathways by which viruses alter aspects of the host phenotype and the virus components mediating these alterations remain poorly understood (19, 20).

In the present study, we addressed these shortcomings and initiated an analysis of the effects of two viruses, TuYV and CaMV, belonging to two different transmission categories, on transcriptomic profiles in two susceptible host plant species (*Arabidopsis thaliana* and *Camelina sativa*, both family *Brassicaceae*) and on changes in insect vector feeding behavior and performances. We selected the green peach aphid *Myzus persicae* as a vector because it transmits both TuYV and CaMV and infests both plant hosts. We chose two different plant species as virus hosts, as previous studies have highlighted potential host-specific effects of viruses on host plant traits and vector performance (21, 22). In addition, their phylogenetic proximity allows rather easy gene-to-gene comparisons. In fact, the *C. sativa* genome is highly similar to the *A. thaliana* genome and might have arisen from the hybridization of three diploid ancestors of *A. thaliana* (23). For this reason, its genome is allohexaploid. Over 70% of *C. sativa* genes are syntenically orthologous to *A. thaliana* genes (24), facilitating genomic studies of *C. sativa*.

## RESULTS AND DISCUSSION

**Plant phenotypes.** We used, in this study, 5-week-old *A. thaliana* Col-0 or *C. sativa* var. *Celine* plants that had been inoculated with CaMV or TuYV 3 weeks before analysis. We used plants at this age and infection state because they displayed symptoms but not yet necrosis. In both *A. thaliana* Col-0 and *C. sativa* var. *Celine* plants, CaMV caused severe leaf curling and mosaic and vein chlorosis as well as dwarfism (Fig. 1). TuYV-infected *A. thaliana* Col-0 and *C. sativa* var. *Celine* plants were smaller than mock-inoculated plants but showed no leaf deformation or bleaching. Older TuYV-infected *A. thaliana* Col-0 leaves turned purple, probably due to stress-induced anthocyanin accumulation, as previously reported for infection of *A. thaliana* Col-0 with another polerovirus, *Brassica* yellows virus (25). The purple coloring was first visible on the abaxial leaf surface and progressed slowly until covering the entire leaf very late in infection. Old leaves of TuYV-infected *C. sativa* var. *Celine* displayed mild yellowing symptoms, primarily on the leaf border.

Western blot analysis showed that TuYV accumulated, as previously reported (26), to similar levels in *A. thaliana* Col-0 and *C. sativa* var. *Celine*. Thus, there was no obvious link between TuYV accumulation and severity of symptoms since, despite comparable TuYV loads in *A. thaliana* Col-0 and *C. sativa* var. *Celine*, disease symptoms were stronger in *C. sativa* var. *Celine* than in *A. thaliana* Col-0 (Fig. 1; compare the stunted phenotype of TuYV-infected *C. sativa* var. *Celine* with the weak phenotype of TuYV-infected

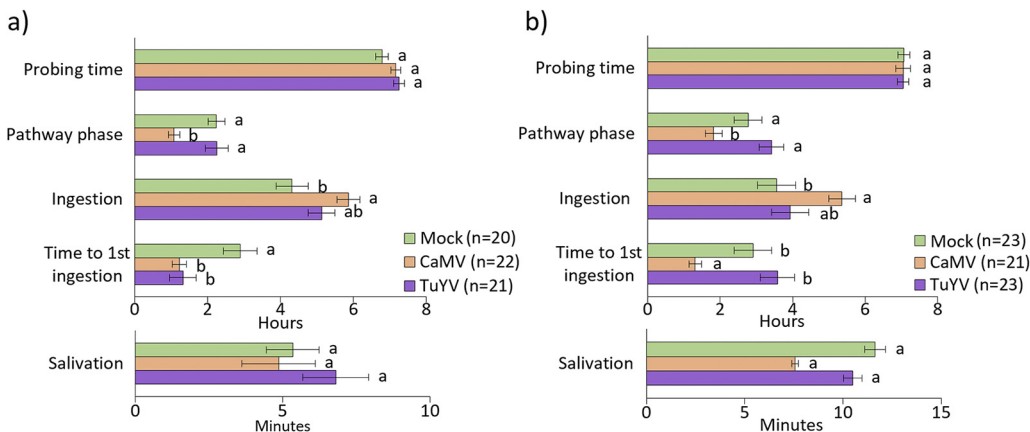

**FIG 2** Aphid feeding behavior parameters recorded by EPG on 5-week-old mock-inoculated, TuYV- or CaMV-infected *A. thaliana* Col-0 (a) and *C. sativa* var. *Celine* (b). Different letters indicate significant differences between plants as tested by GLM followed by pairwise comparisons using emmeans (*P* < 0.05; method, Tukey; *n* = 20 to 23).

*A. thaliana* Col-0). CaMV loads were higher in *C. sativa* var. *Celine* than in *A. thaliana* Col-0; whether this correlated with symptom expression was difficult to access because of the severe phenotype in both hosts.

**Aphid feeding behavior and fecundity.** We used the electrical penetration graph technique (EPG) to compare aphid probing and feeding behavior on *A. thaliana* Col-0 and *C. sativa* var. *Celine* infected or not with CaMV or TuYV (Fig. 2a and b). The total probing time was similar for all six conditions, and the aphids were active for approximately 7 h during the 8-h observation period. The pathway phase and the time until the first phloem ingestion were, in general, longer on *C. sativa* var. *Celine* for all three conditions, whereas the ingestion phase was longer on *A. thaliana* Col-0. The most important difference was the salivation time, which was extended on *C. sativa* var. *Celine* (50 to 100% longer than on *A. thaliana* Col-0), independently of the infection status.

CaMV infection changed aphid behavior similarly in both plant hosts. The pathway phase and the time to first phloem ingestion were decreased, whereas phloem ingestion was increased. Salivation time was not affected by the infection status of the two plant species. Infection with TuYV had no major effect. The only behavioral parameter significantly affected by TuYV was the time until first phloem ingestion, which was reduced by half on TuYV-infected *A. thaliana* Col-0 but not on TuYV-infected *C. sativa* var. *Celine*, compared to mock-inoculated plants. This is in contrast with CaMV infection for which the time to first phloem sap ingestion was reduced on both hosts. Previous EPG experiments on *A. thaliana* Col-0 (27) and *C. sativa* var. *Celine* (28) have reported neutral to slightly positive effects of TuYV infection on aphid probing and feeding behavior and highlighted also host-specific viral effects on plant quality and vector behavior (21).

Taken together, the significantly reduced time until first phloem ingestion and the longer phloem ingestion observed on infected *A. thaliana* Col-0 might contribute to better acquisition of phloem-restricted TuYV. The other transmission-related feeding parameters were only marginally modified on TuYV-infected plants, whereas CaMV infection altered aphid feeding more strongly. The reduced pathway phase and the increased phloem ingestion might also facilitate CaMV acquisition from phloem tissues. These alterations are expected to be detrimental for noncirculative viruses (such as the nonpersistent potyviruses) that are acquired during intracellular penetrations occurring in the pathway phase, but lost if the aphid stylets reach the phloem sap (29). However, this does not apply to CaMV, acquired efficiently from phloem sap as well as mesophyll and epidermis cells (30).

Infection with CaMV reduced aphid fecundity significantly in both plant host species (Fig. 3a and b) compared to mock-inoculated plants (generalized linear model

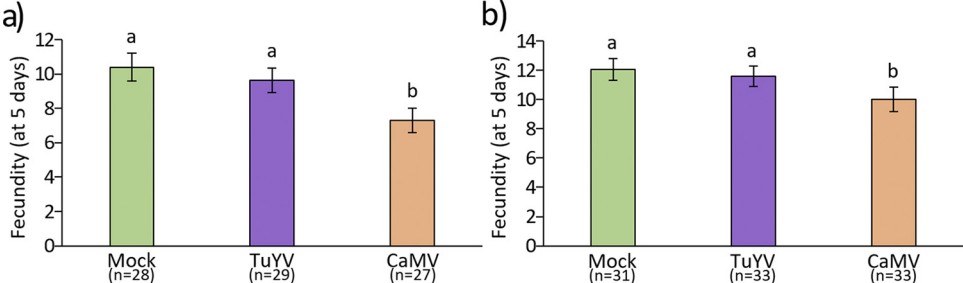

**FIG 3** Aphid fecundity 5 days after deposit (one aphid per plant) on 5-week-old mock-inoculated, TuYV- or CaMV-infected *A. thaliana* Col-0 (a) and *C. sativa* var. *Celine* (b). Different letters indicate significant differences between plants as tested by GLM followed by pairwise comparisons using emmeans ($P < 0.05$; method, Tukey; $n = 27$ to 33).

[GLM], $\chi^2 = 0.0007$ and $\chi^2 = 0.0409$ for *A. thaliana* Col-0 and *C. sativa* var. *Celine*, respectively) and correlated with the strong symptoms of infected plants. Fecundity was unchanged on TuYV-infected *A. thaliana* Col-0 and *C. sativa* var. *Celine*. This indicated that the severe (but less strong than CaMV infection) phenotype of TuYV-infected *C. sativa* var. *Celine* did not interfere with aphid fecundity.

**Quality of RNA and sequence alignment data.** Roughly 29 to 35 million reads were obtained for *A. thaliana* Col-0 mRNA sequencing (mRNA-seq) data sets, of which >80% could be aligned for mock-inoculated and TuYV-infected samples and 80% for CaMV-infected samples (see Table S2 in the supplemental material). For *C. sativa* var. *Celine*, 28 to 38 million reads were obtained, and 61 to 71% of the reads could be aligned (Table S2). Principal-component analysis of the transcriptome sequencing (RNA-seq) (Fig. 4a and b) for both plant species indicated that the three biological replicates per condition clustered well together and that the different conditions (mock inoculated or infected with either virus) were well separated. Thus, the reads were of excellent quality and suited for transcriptome analysis.

For 8 selected *A. thaliana* Col-0 genes, the trends of gene deregulations detected in the transcriptome data could be reproduced by an alternative analysis method, reverse transcriptase quantitative PCR (RT-qPCR) (Fig. S1). All 8 genes followed the same trend using either method for CaMV-infected *A. thaliana* Col-0 and all except two (At_AOS and At_EDS5) for TuYV-infected *A. thaliana* Col-0. The discrepancies were probably due to the rather weak expression changes, which are sometimes difficult to detect by RT-qPCR due to its intrinsic exponential amplification kinetics.

Quantification of viral RNA loads by counting viral reads normalized per million of total plant reads in each sample revealed that CaMV pregenomic RNA (pgRNA) and TuYV genomic RNA (gRNA; both represented by forward reads) (Data Set S1) accumulated to comparable levels in each of the three biological replicates except for one of the three TuYV-*A. thaliana* Col-0 replicates, which showed a lower number of normalized viral reads. The data confirmed further that the mock-inoculated plants were not cross-contaminated. Note that because TuYV gRNA is not polyadenylated (unlike CaMV pgRNA), the poly(A) enrichment step of the Illumina library preparation protocol should have led to its depletion. Therefore, TuYV accumulation, as judged by our RNA-seq data, might not be accurate, and indeed, we found a difference between Western blot results and RNA-seq data (Data Set S1). Concerning CaMV, its loads were lower in *A. thaliana* Col-0 than in *C. sativa* var. *Celine* (ca. 1.5 times) (Data Set S1), in line with the Western blot results (Fig. 1d).

**CaMV modifies expression of far more genes than TuYV.** We determined the number of differentially expressed genes (DEGs) in infected hosts (Fig. 4c to e). Far more DEGs were detected in *C. sativa* var. *Celine* than in *A. thaliana* Col-0. This was in part due to its allohexaploid genome consisting of three *A. thaliana* Col-0-like genomes coding for almost 90,000 genes (24) compared to *A. thaliana* Col-0's diploid genome containing about 28,000 coding genes (http://ensembl.gramene.org/Arabidopsis_thaliana/Info/Annotation/#assembly, last accessed 17 December 2021). This means that for many *A. thaliana* Col-0 genes, there

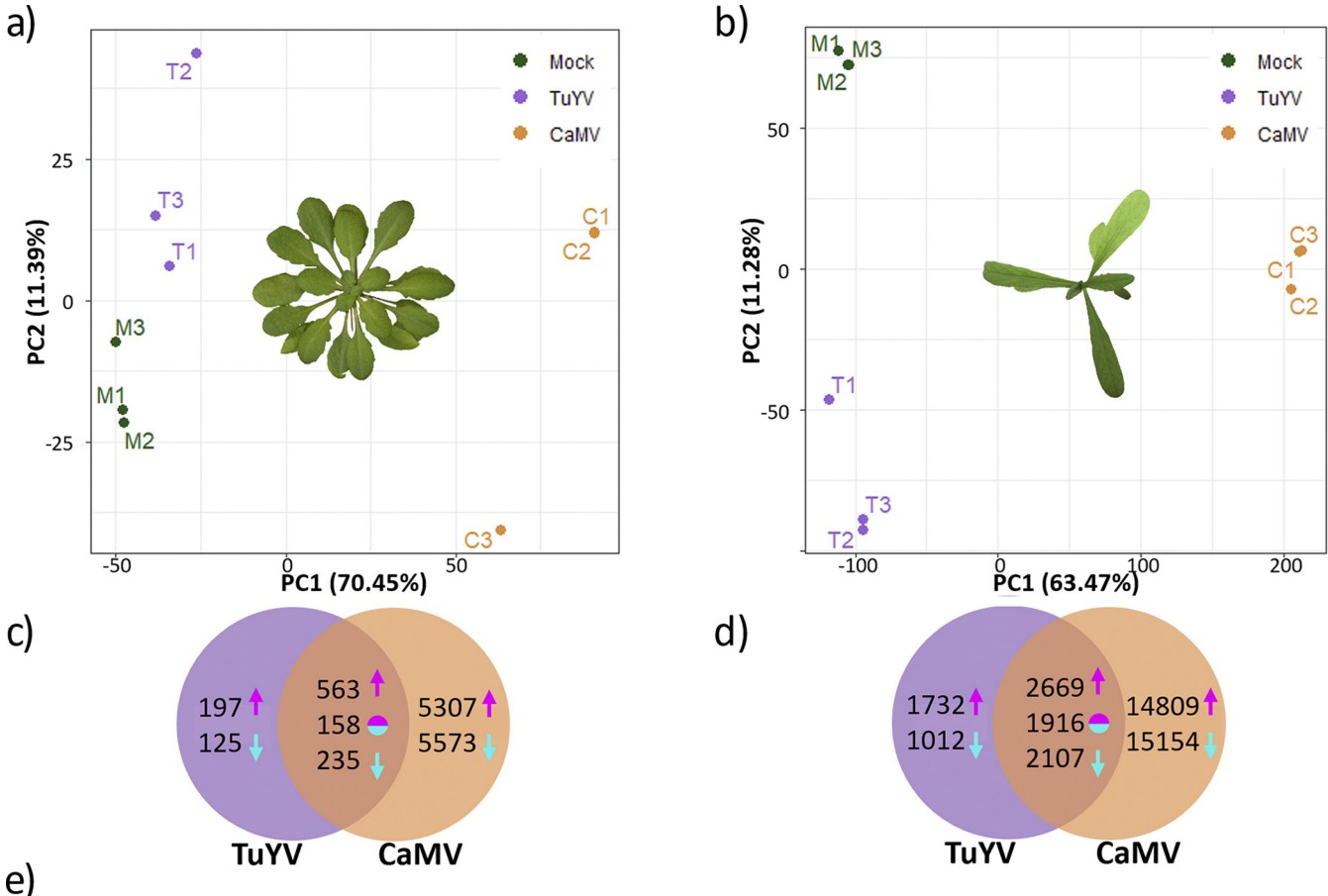

**FIG 4** Principal-component (PC) analysis of the transcriptome data sets on *A. thaliana* Col-0 (a) and *C. sativa* var. *Celine* (b). Three dots of the same color correspond to the three biological replicates. (c and d) Venn diagrams presenting the number of differentially expressed genes (DEGs) in TuYV and CaMV-infected *A. thaliana* Col-0 (c) and *C. sativa* var. *Celine* (d). Magenta arrows, number of upregulated genes; cyan arrows, number of downregulated genes; two-color circles, inversely regulated genes (upregulated genes in one virus-infected modality and downregulated in the other virus-infected modality). (e) Comparison of the number of DEGs and enriched GO terms in TuYV- and CaMV-infected *A. thaliana* Col-0 and *C. sativa* var. *Celine* plants.

are three orthologous *C. sativa* var. *Celine* genes. Also, the higher accumulation of both viruses in this host might contribute to the higher counts.

In *A. thaliana* Col-0, CaMV modified the expression of ~11,800 genes significantly ($P_{adj} < 0.05$), whereas TuYV modified the expression of ~1,300 genes, corresponding to 43% and 5% of the total genes, respectively (Fig. 4e). In CaMV-infected *C. sativa* var. *Celine*, we detected ~36,700 DEGs, and in TuYV-infected *C. sativa* var. *Celine*, we detected ~9,400 DEGs, corresponding to 41% and 11% of all genes, respectively. Thus, the impact of CaMV infection on gene deregulation was much more pronounced than TuYV infection, in accordance with the phenotype of infected plants (Fig. 1). The lower numbers of DEGs for TuYV in both hosts could be at least partially due to its restriction to phloem tissues, unlike CaMV, which infects all cell types.

CaMV modified expression of ~40% of the total genes independently of the host plant, whereas the proportion of TuYV-induced DEGs was host dependent and two times higher in infected *C. sativa* var. *Celine* than in *A. thaliana* Col-0 (11% versus 5%). This is in line with the relative loads of viral RNA (Data Set S1), indicating that *C. sativa* var. *Celine* is more susceptible to TuYV infection than *A. thaliana* Col-0 (3 times more TuYV RNA accumulation in *C. sativa* var. *Celine* than in *A. thaliana* Col-0), while CaMV accumulates in both hosts at comparable levels (only 1.5-fold difference in average viral RNA loads between *A. thaliana* Col-0 and *C. sativa* var. *Celine*).

There were 956 DEGs, corresponding to 3.4% of the genome, that were common for both viruses in *A. thaliana* Col-0. The proportion of common DEGs rose to 7.5% (~6,700 genes) in infected *C. sativa* var. *Celine*. Since CaMV and TuYV are viruses with entirely different replication mechanisms, mediated by, respectively, viral reverse transcriptase and viral RNA-dependent RNA polymerase, these common host genes might be involved in general stress responses and/or are constituents of the core defense mechanisms. Gene ontology (GO) analysis indicated that this was true for *A. thaliana* Col-0, with GO terms related to stress and transport in common for both infections, whereas for *C. sativa* var. *Celine*, rather, ribosome and replication-related genes were enriched (Fig. S2).

The proportions of up- and downregulated genes were similar for a given virus in the two hosts (Fig. 4e). However, when comparing the two viruses, the proportion of downregulated genes was higher in CaMV-infected plants (about 50% in CaMV-infected *A. thaliana* Col-0 and *C. sativa* var. *Celine*) than in TuYV-infected plants (34% in TuYV-infected *A. thaliana* Col-0 and 39% in TuYV-infected *C. sativa* var. *Celine*). Thus, there was a correlation between the proportion of downregulated genes and symptom severity. The milder disease symptoms of TuYV infection coincided, in both hosts, with the lower proportion of downregulated genes, while the ability of CaMV to downregulate a higher proportion of genes coincided with the more severe disease symptoms. The latter ability might reflect CaMV activities in both cytoplasm (viral mRNA translation and pgRNA reverse transcription) and nucleus (viral dsDNA repair followed by pgRNA transcription and export assisted by nuclear-imported viral proteins P4 and P5 and likely P6 [31, 32]).

**Impact of CaMV and TuYV infection on plant hosts: gene ontology analysis.** To identify the most prominent processes affected in aphid-infested CaMV and TuYV-infected *A. thaliana* Col-0 and *C. sativa* var. *Celine*, we carried out a GO analysis (Fig. 5). In general, TuYV-induced GO changes were much less pronounced (considering the percentage of DEGs in each category and the DEG counts) than CaMV, reflecting the low absolute numbers of DEGs in TuYV-infected plants and the weaker impact of TuYV on plant phenotype. Remarkably, in the top 25 categories, only about 25% of genes per GO were deregulated in TuYV-infected *A. thaliana* Col-0 (Fig. 5a), whereas this value increased to more than 50% in TuYV-infected *C. sativa* var. *Celine* (Fig. 5c). Again, this may indicate that TuYV infection had a stronger effect on gene regulation in *C. sativa* var. *Celine* than in *A. thaliana* Col-0. A different situation was found for CaMV, where the percentages of DEGs per GO were similar in both hosts and always above 50% of genes per GO, indicating similarly strong interactions with either host plant.

Then we looked closer at the different categories. Interestingly, in CaMV-infected *A. thaliana* Col-0 (Fig. 5b), most of the enriched GO terms were related to photosynthesis/chloroplast in both biological process (BP) and cellular component (CC) categories, which might explain leaf chlorosis (loss of chlorophyll and/or chloroplasts). The next most affected biological process was oxidation-reduction, which might be related to stress response. Also, GO terms related to microtubule-based movement appeared in the BP and CC lists, as well as apoplast, cell wall, kinesin complex, and nucleolus, which may be linked to virus or viral RNP intracellular trafficking. Taken together, CaMV infection mostly modified photosynthesis, oxidation-reduction processes, and microtubule-related processes.

GO analysis of CaMV-infected *C. sativa* var. *Celine* indicated a similar pattern (Fig. 5d). Again, several GO terms related to photosynthesis-chloroplast and oxidation-reduction were

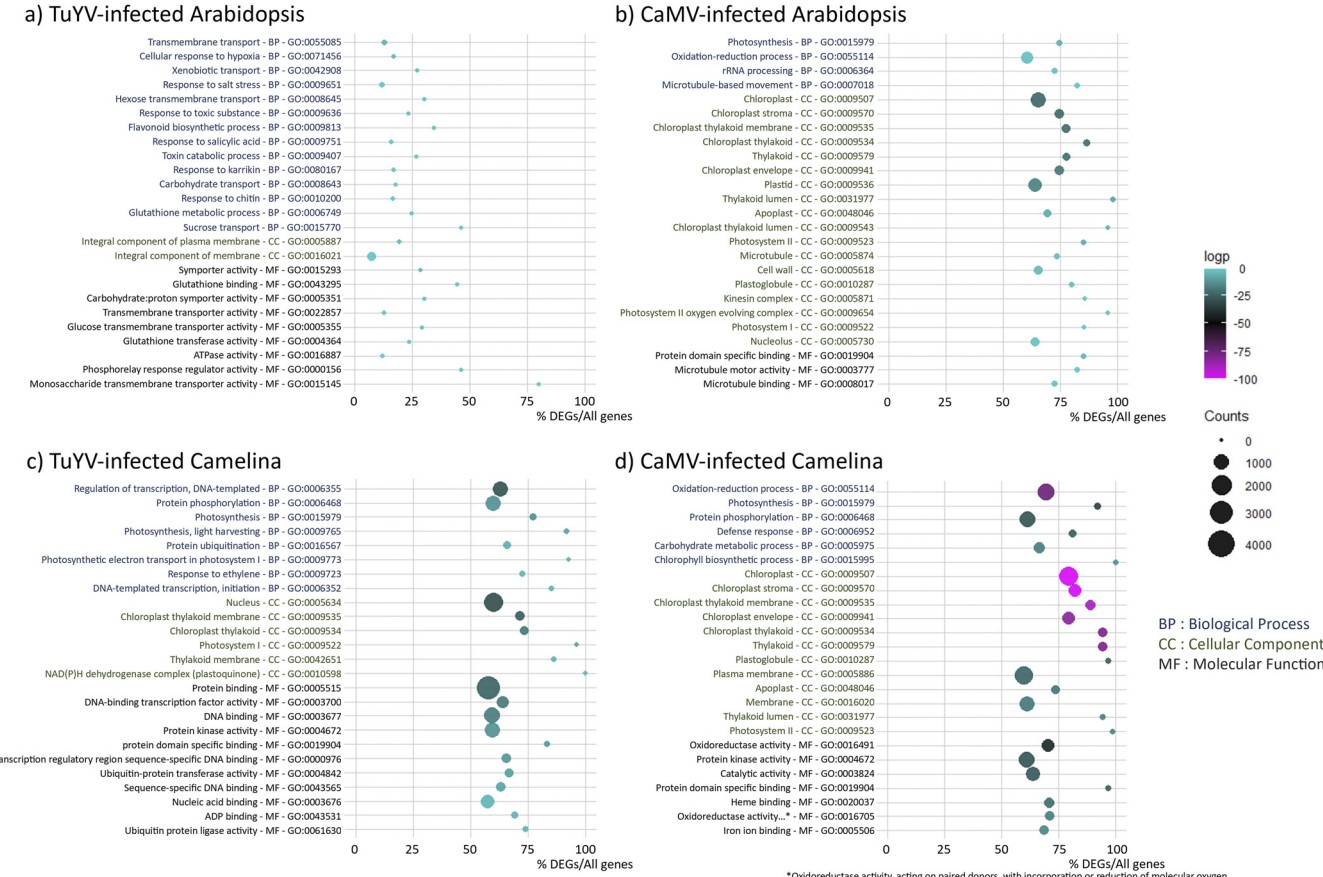

**FIG 5** Gene ontology (GO) analysis showing the top 25 GO terms of deregulated processes in TuYV- and CaMV-infected *A. thaliana* Col-0 and *C. sativa* var. *Celine*. (a) TuYV-infected versus mock-inoculated *A. thaliana* Col-0; (b) CaMV-infected versus mock-inoculated *A. thaliana* Col-0, (c) TuYV-infected versus mock-inoculated *C. sativa* var. *Celine*; (d) CaMV-infected versus mock-inoculated *C. sativa* var. *Celine*. GO IDs and corresponding GO terms are specified in the vertical axis. For each category (BP, biological process; CC, cellular component; MF, molecular function), GOs are sorted according to increasing $\log_2 P$ values, also indicated by the color of each spot (magenta representing the most significant *P* values; see color scale bar) to place the most significantly enriched GOs on top of the graph. The absolute number of DEGs that matched the GO term is indicated by the size of each spot, whereas the horizontal axis shows the ratio of DEGs to all genes belonging to the GO term.

enriched in both BP and CC categories. It is worth mentioning that for CaMV infection of *C. sativa* var. *Celine*, GO analysis showed enrichment of genes in the GO defense response (BP; GO:0006952), which was absent in the top 25 list of CaMV-infected *A. thaliana* Col-0. Other BP-related enriched GOs were protein phosphorylation and carbohydrate metabolism. As in *A. thaliana* Col-0, apoplast changes were significant. On the other side, neither cell wall nor microtubule processes were present among the top 25 deregulated processes in *C. sativa* var. *Celine*. Concerning the main categories of molecular functions, oxidation-reduction and protein domain-specific binding dominated this category.

In contrast to CaMV, TuYV infection of *A. thaliana* Col-0 had no impact on photosynthesis-related GO terms. In both BP and molecular function (MF), the most significantly enriched GO terms were found in transport, especially carbohydrate transport. In addition, some defense and stress responses (xenobiotics, chitin, salt) and glutathione metabolism —indicative of oxidative stress—were affected. Flavonoid synthesis was significantly deregulated, in line with the purple-colored leaves of TuYV-infected *A. thaliana* Col-0 (Fig. 1). In accordance with the modifications in sucrose transport, the most prominent category in CC comprised membranes. The top 25 GO terms in TuYV-infected *C. sativa* var. *Celine* were different from those in TuYV-infected *A. thaliana* Col-0. As in CaMV infection, DEGs in photosynthesis and related processes dominated the top 25 GO terms in TuYV-infected *C. sativa* var. *Celine* in BP and CC and were likely related to the mild yellowing symptoms appearing on old leaves. Next were DNA-related processes in both BP and MF

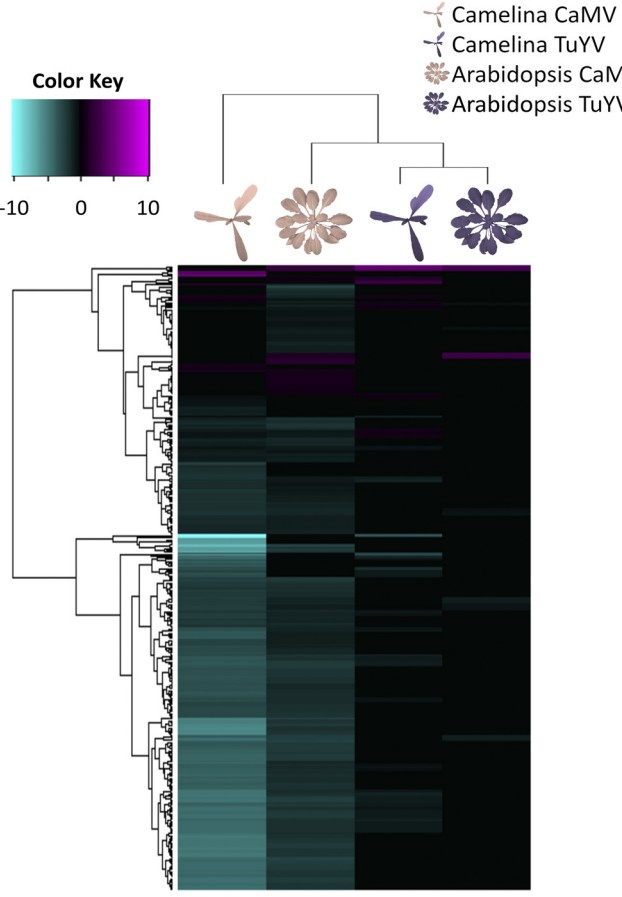

**FIG 6** Hierarchical clustering of differentially expressed genes (DEGs) related to photosynthesis (GO:0015979) in CaMV- and TuYV-infected *A. thaliana* Col-0 and *C. sativa* var. *Celine* compared to their mock-inoculated control plants (Data Set S2). The color key scale displays the $\log_2$ fold changes from −10 to +10 as a gradient from cyan to magenta.

categories, probably linked to transcriptional regulations of host genes in response to viral infection. In CC, the GO term nucleus was deregulated, again in favor of a strong effect of TuYV on transcriptional regulation in infected *C. sativa* var. *Celine*. Other deregulated processes included ubiquitination, which appeared in several categories in BP and MF. On the other hand, oxidation-reduction did not appear under the top 25 GO terms except as plastoquinone, which represents a significant difference between CaMV and TuYV infections of *A. thaliana* Col-0.

**Impact of CaMV and TuYV infection on plant hosts: heatmap analyses.** To better characterize the impact of viral infection on aphid-infested plants, we established the lists (Data Set S2) and corresponding heatmaps (Fig. 6 through 10) of DEGs for selected categories. Note that if not otherwise indicated, information on gene function is from the TAIR site (https://www.arabidopsis.org/). For mapping *A. thaliana* Col-0 and *C. sativa* var. *Celine* genes in the heatmaps, we used the syntenic orthologues matrix (24). This matrix assorts each *A. thaliana* Col-0 gene to the corresponding triplet of *C. sativa* var. *Celine* homologs. Out of 89,418 *C. sativa* var. *Celine* genes, 62,277 are syntelogs to *A. thaliana* Col-0 genes, among which some are considered "fractionated" (if one or two of the homologs were lost). This explains why for certain *A. thaliana* Col-0 genes, only one or two homologous *C. sativa* var. *Celine* genes are presented.

**Photosynthesis-related genes responsive to CaMV and TuYV.** CaMV and TuYV infection downregulated photosynthesis-related genes in infested *A. thaliana* Col-0 and *C. sativa* var. *Celine* (Fig. 6). Overall downregulation of photosynthesis-related genes was much more pronounced in CaMV-infected than in TuYV-infected plants. Interestingly, both viruses interacted more strongly with *C. sativa* var. *Celine* photosynthesis than with

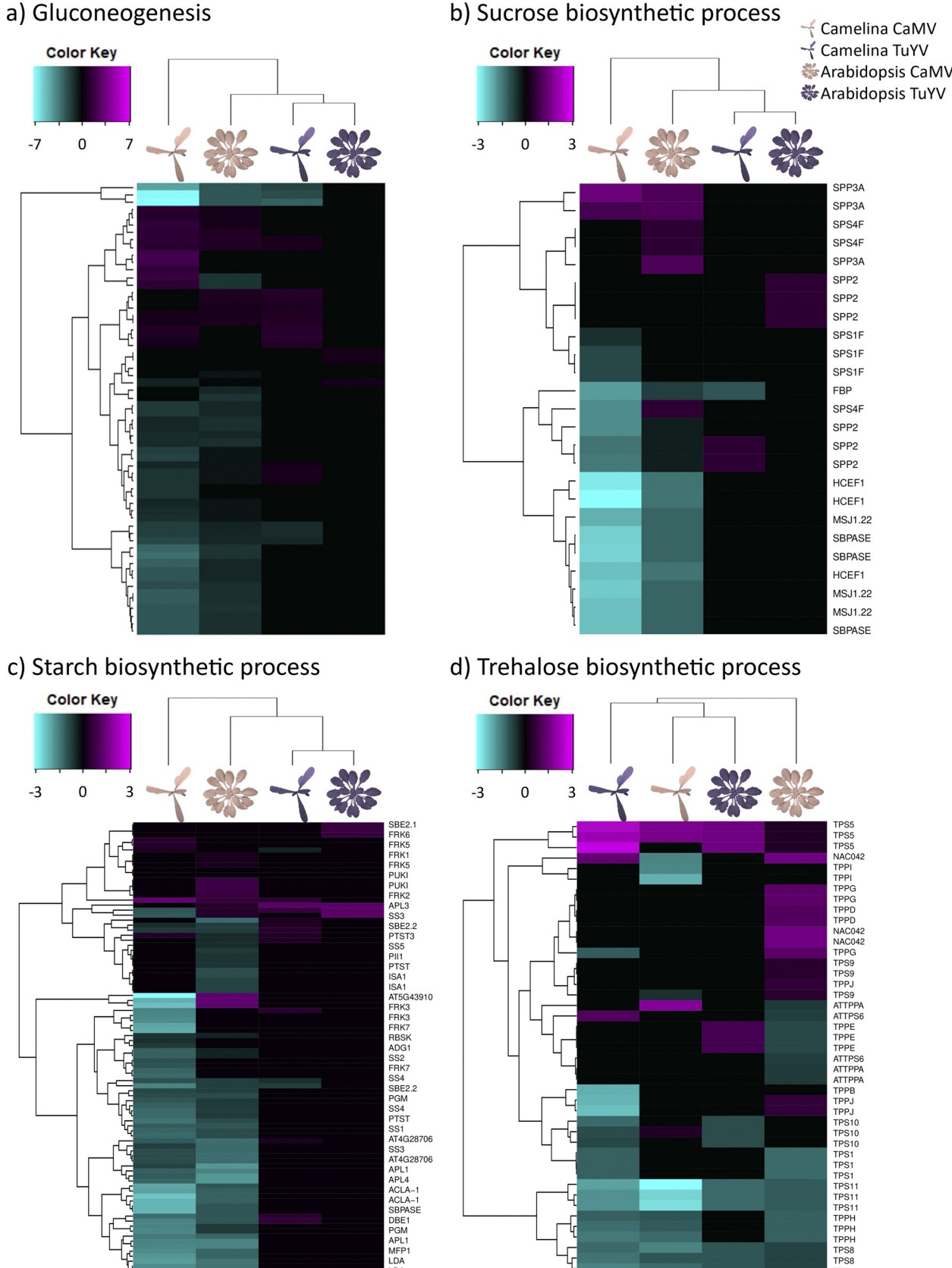

**FIG 7** Hierarchical clustering of differentially expressed genes (DEGs) related to gluconeogenesis (GO:0006094) (a), sucrose biosynthetic process (GO:0005986) (b), starch biosynthetic process (GO:0019252) (c), and trehalose biosynthetic process (GO:0005992) (d) in CaMV- and TuYV-infected *A. thaliana* Col-0 and *C. sativa* var. *Celine* compared to mock-inoculated controls (Data Set S2). The color key scales display the log$_2$ fold changes as color gradients from cyan to magenta.

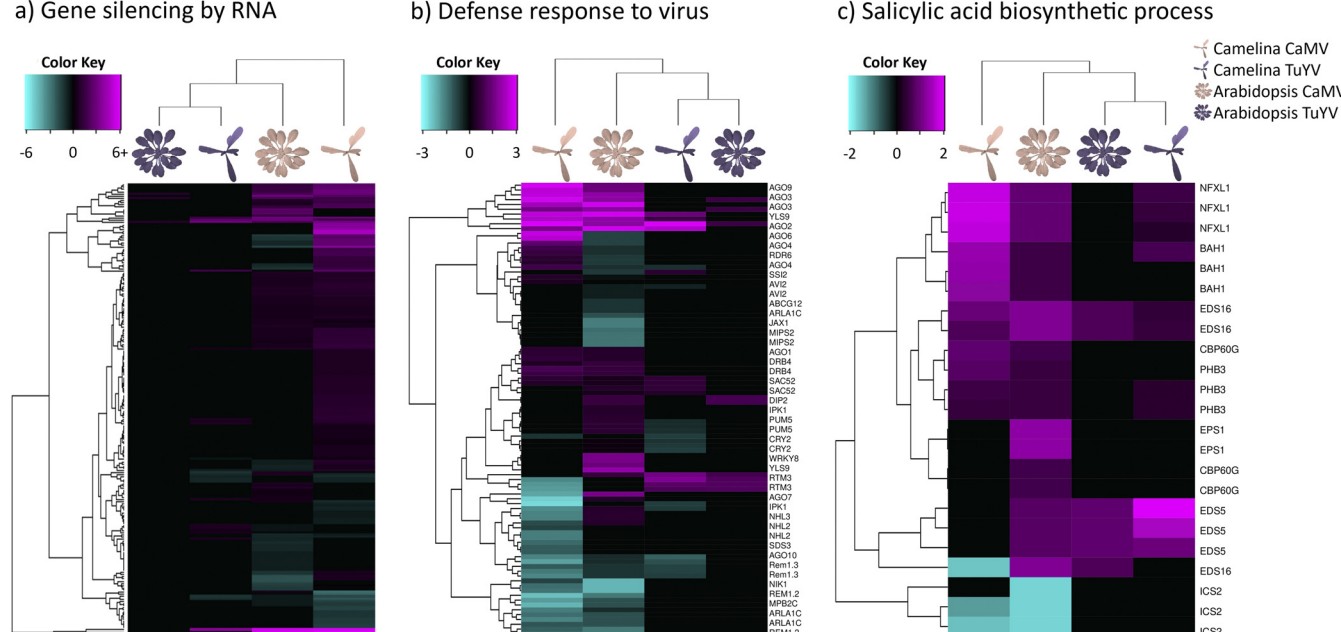

**FIG 8** Hierarchical clustering of differentially expressed genes (DEGs) related to production of siRNA involved in RNA interference and gene silencing by RNA (GO:0030422 and GO:0031047) (a), defense response to virus (GO:0051607) (b), and salicylic acid biosynthetic process (GO:0009697) (c) in CaMV- and TuYV-infected *A. thaliana* Col-0 and *C. sativa* var. *Celine* compared to their mock-inoculated controls (Data Set S2). The color key scales display the $\log_2$ fold changes as gradients from cyan to magenta.

*A. thaliana* Col-0 photosynthesis. The most downregulated photosynthesis gene in CaMV-infected *C. sativa* var. *Celine* was *PORA* {*C. sativa* var. *Celine* Csa02g051950 ($\log_2$ fold change [$\log_2$FC], −10.23), Csa11g086170 ($\log_2$FC = −5.53), and Csa18g025480 ($\log_2$FC = −3.48), corresponding to *A. thaliana* Col-0 AT5G54190), coding for a protein involved in chlorophyll biosynthesis, but also in response to ethylene. *PORA* expression was also inhibited in TuYV-infected *Camelina*. Expression of the *A. thaliana* Col-0 orthologue, however, was not modified by any of the two viruses. This might indicate that downregulation of *PORA* is a plant-specific, and not a virus-specific, response. The most downregulated gene in CaMV-infected *A. thaliana* Col-0, AT3G27690 ($\log_2$FC = −3.38), encoding the protein LHCb2.3, a component of the light-harvesting complex, was also strongly repressed in *C. sativa* var. *Celine* infected by CaMV. Expression of this gene was also affected in TuYV-infected *A. thaliana* Col-0, but to a lesser extent. Some photosynthesis genes were upregulated by infection. This was notably the case for the glucose-6-phosphate/phosphate transporter 2 (AT1G61800), which is involved in the regulation of photosynthesis and was upregulated in three of the four conditions (not in CaMV-infected *C. sativa* var. *Celine*). In CaMV-infected *C. sativa* var. *Celine*, a chloroplast ferritin (AT3G11050) was the most upregulated photosynthesis gene ($\log_2$FC > 3.7 for the three homologs). Ferritins are iron-binding proteins and are supposed to be involved in responses against oxidative stress in *A. thaliana* Col-0 (33), which could explain its overexpression.

Taken together, virus infection strongly interfered with photosynthesis. This might explain the leaf yellowing observed clearly on CaMV-infected plants and to a lesser extent on TuYV-infected *C. sativa* var. *Celine*. Leaf yellowing, probably due to reduced chlorophyll content in chloroplasts and/or a reduced number of photosynthesizing chloroplasts caused by the gene deregulations (34), can alter the settling preference of aphids (35, 36). However, this was not confirmed by previous observations on TuYV-infected and CaMV-infected *C. sativa* var. *Celine* plants (28). Indeed, although Chesnais and coworkers reported that *M. persicae* aphids preferred to settle on TuYV-infected *C. sativa* var. *Celine*, compared to healthy plants, no such aphid preference was observed

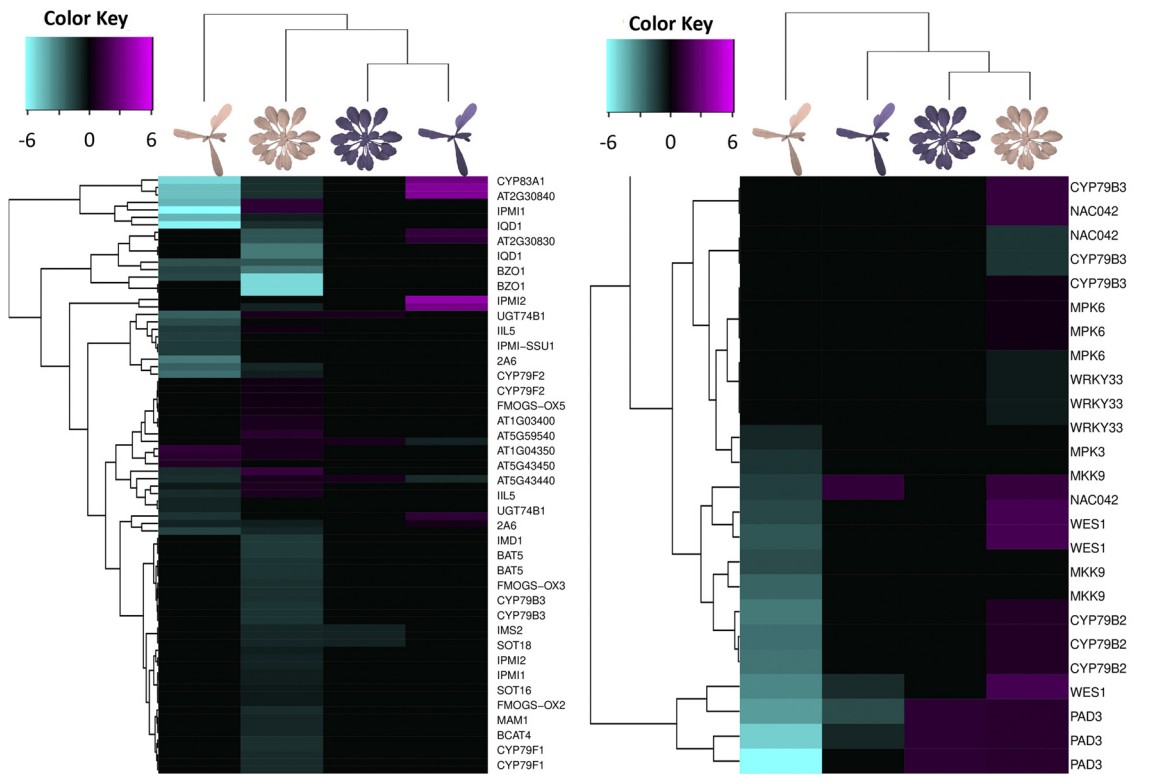

**FIG 9** Hierarchical clustering of differentially expressed genes (DEGs) related to jasmonic acid biosynthetic process (GO:0009695) (a), ethylene biosynthetic process (GO:0009693) (b), glucosinolate biosynthetic process (GO:0019761) (c), and camalexin biosynthetic process (GO:0010120) (d) in CaMV- and TuYV-infected *A. thaliana* Col-0 and *C. sativa* var. *Celine* compared to mock-inoculated controls (Data Set S2). The color key scales display the log$_2$ fold changes as gradients from cyan to magenta.

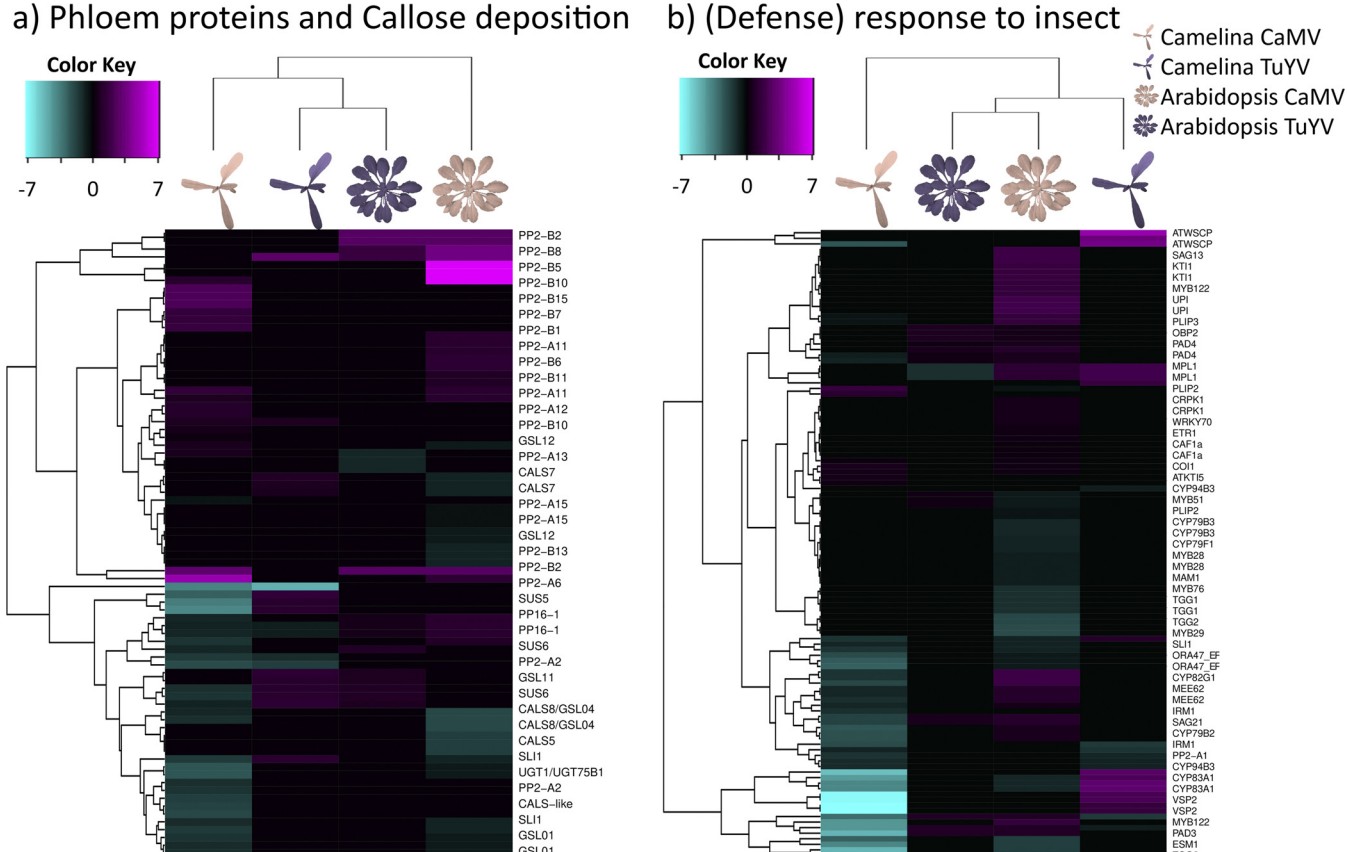

**FIG 10** Hierarchical clustering of differentially expressed genes (DEGs) related to phloem proteins (PP2 and PP1) and callose deposition in phloem sieve plates (GO:0080165) (a) and defense response to insect and response to insect (GO:0002213 and GO:0009625) (b) in CaMV- and TuYV-infected *A. thaliana* Col-0 and *C. sativa* var. *Celine* compared to mock-inoculated controls (Data Set S2). The color key scales display the log$_2$ fold changes as gradients from cyan to magenta.

for CaMV-infected *C. sativa* var. *Celine* despite the strong yellowing symptoms. This suggests that aphid preference for a plant is not only driven by visual aspects.

**Carbohydrate pathway genes responsive to CaMV and TuYV.** In line with the repression of photosynthesis, the expression of many sucrose synthesis and gluconeogenesis-related genes was reduced by CaMV infection (Fig. 7a and b). The effect of CaMV was stronger in *C. sativa* var. *Celine* than in *A. thaliana* Col-0. In TuYV-infected *C. sativa* var. *Celine*, the amplitude of the gene deregulation was smaller than in CaMV-infected *C. sativa* var. *Celine*, but the proportions of up- and downregulated genes were comparable. For both TuYV- and CaMV-infected plants, among the most downregulated genes were those coding for key enzymes in sucrose synthesis, in particular *HCEF1* (AT3G54050) and *FBP* (AT1G43670) (down to log$_2$FC values of −3.07 and −1.85, respectively). The most downregulated gene in gluconeogenesis was the aldolase *FBA5* (AT4G26530), and this was in all four conditions tested (down to log$_2$FC = −7.08 for CaMV-infected *C. sativa* var. *Celine*). In line with the stronger suppression of gluconeogenesis and sucrose synthesis-related genes by CaMV infection, also many starch synthesis-related genes were repressed by CaMV (but not TuYV) infection (Fig. 7c). An exception was *DBE1* (Csa17g005380, Csa14g004380, and Csa03g004400, syntelogs of AT1G03310) encoding a starch-branching enzyme upregulated in TuYV-infected *C. sativa* var. *Celine* (log$_2$FC = 0.6 to 0.8). This is consistent with a recent study showing that TuYV infection tends to increase the carbohydrate concentrations in *C. sativa* var. *Celine* leaves (21).

The effect of infection and infestation on trehalose metabolism was different from that on glucose, starch, and sucrose metabolism (Fig. 7d). Contrary to the

downregulation of the latter carbohydrate pathways only in CaMV-infected plants, trehalose-related genes were downregulated in both TuYV- and CaMV-infected *A. thaliana* Col-0 and *C. sativa* var. *Celine*. Upregulated genes in this pathway were also observed, in particular in CaMV-infected *A. thaliana* Col-0. Trehalose is induced by *M. persicae* infestation and has been shown to contribute to defenses against aphids (37–39). Trehalose phosphate synthase 11 (*TPS11*) especially has been implicated in mounting defenses against aphids (38) by promoting starch synthesis and also by activating the phytoalexin-deficient gene, *PAD4*. Starch is a feeding deterrent to aphids (40), and elevated starch levels are correlated with reduced aphid performance (38). Interestingly, *TPS11* and also *TPS8* were significantly downregulated in all virus-infected plants (log$_2$FC = −0.67 to −2.75), suggesting that viral infection might favor aphid infestation. On the other hand, other TPS isoforms were not modified in the same way; for example, the *TPS5* (AT4G17770) was upregulated in all four conditions tested (log$_2$FC = 0.35 to 2.45). This suggests a more complex regulation of this pathway by a viral infection and aphid infestation.

We also analyzed the expression of genes involved in amino acid metabolism because they represent the most important food source for aphids. Probably due to the multiple functions of these genes, no clear pattern was detected (Fig. S3).

**Virus defense-related genes responsive to CaMV and TuYV.** When looking at global virus defense-related genes, the effects on their regulation were more virus specific than host specific. In agreement with previous findings in CaMV-infected *A. thaliana* Col-0 (41), many RNA silencing-related genes were found to be upregulated by CaMV, not only in *A. thaliana* Col-0 but also in *C. sativa* var. *Celine* (Fig. 8a). Among them, most notable are components of the 21-nucleotide (nt) small interfering RNA (siRNA)-directed gene silencing pathways such as double-stranded RNA-binding protein 4 (DRB4; log$_2$FC = 0.44 to 1.09), a partner of the antiviral Dicer-like protein 4 (DCL4) generating 21-nt siRNAs, and siRNA-binding effector proteins Argonaute 1 (AGO1; AT1G48410), AGO2 (AT1G31280), and AGO3 (AT1G31290). Notably, AGO2, also known to be involved in defense against RNA viruses (42), was upregulated in TuYV-infected *C. sativa* var. *Celine* (log$_2$FC = 1.58 to 2.63) (but not in TuYV-infected *A. thaliana* Col-0), while AGO3, the Argonaute protein most closely related to AGO2 and also showing antiviral activity *in vitro* (43), was upregulated in response to TuYV infection in both hosts (log$_2$FC = 0.91 in *A. thaliana* Col-0 and log$_2$FC = 3.241 for Csa17g04886). This suggests redundant (and compensatory) roles of AGO2 and AGO3 in defenses against both RNA and DNA viruses. DCL4 itself (AT5G20320) and DCL2 (AT3G03300), generating 22-nt siRNAs and acting together with DCL4 in defenses against RNA and DNA viruses (44), were, respectively, weakly (one of three isoforms of *C. sativa* var. *Celine* DCL4; log2FC = 0.34 for Csa13g023280) and strongly upregulated (DCL2) in CaMV-infected *C. sativa* var. *Celine* (log$_2$FC = 3.85 to 4.83) but not in the other virus-host combinations where their levels were likely sufficiently high to cope with both viruses. Interestingly, AGO10, which counteracts AGO1 in the microRNA (miRNA)-directed silencing pathways regulating plant development and physiology (45), was downregulated by CaMV (but not TuYV) infection in both *A. thaliana* Col-0 (log$_2$FC = −0.47) and *C. sativa* var. *Celine* (log$_2$FC = −1.62 to −2.03). RNA-directed RNA polymerase 6 (RDR6), generating miRNA-dependent secondary 21-nt siRNAs, was upregulated by TuYV infection in *C. sativa* var. *Celine* (log$_2$FC = 0.38 to 0.59) and downregulated by CaMV infection in *A. thaliana* Col-0 (log$_2$FC = −0.69) while remaining nonresponsive in the other virus-host combinations. Components of the nuclear silencing and 24-nt siRNA-directed DNA methylation pathways such as AGO4 (AT2G27040), AGO6 (AT2G32940), and AGO9 (AT5G21150) were upregulated in *C. sativa* var. *Celine* by CaMV (log2FC > 0.8, up to 3.15) (but not TuYV), while AGO4 and AGO6 were downregulated (log$_2$FC = −0.56 and −0.83, respectively), and AGO9 was upregulated in CaMV-infected *A. thaliana* Col-0 (log$_2$FC = 1.62), denoting virus- and plant-specific gene deregulations. Interestingly, the most upregulated gene in the RNA silencing category was AT5G59390. It was strongly upregulated in CaMV-infected *C. sativa* var. *Celine* (log$_2$FC up to 12.35 for Csa18g032680) and *A. thaliana* Col-0 (log$_2$FC = 5.83), less strongly upregulated in TuYV-infected *C. sativa* var. *Celine* (log$_2$FC = 3.82 for Csa18g032680),

and not significantly deregulated in TuYV-infected *A. thaliana* Col-0. This gene codes for an XH/XS domain-containing protein, which probably functions in siRNA-directed DNA methylation and might contribute to methylation and transcriptional silencing of CaMV double-stranded DNA (dsDNA) in the nucleus (46). Taken together, CaMV infection strongly affected silencing-related genes in both hosts, but the deregulations were host specific, with downregulations dominating in *A. thaliana* Col-0 and upregulations in *C. sativa* var. *Celine*. Transcription of RNA silencing genes was much less affected in TuYV-infected plants.

Among components of other defense pathways (Fig. 8b), the hairpin-induced protein hin1 (AT2G35980, also referred to as YLS9 and reported to be induced by cucumber mosaic virus infection) was strongly induced during CaMV infection ($\log_2FC > 2$), while Rem 1.2 (AT3G61260, also referred to as remorin and known to negatively regulate the cell-to-cell movement of the potyvirus TuMV via competition with PCaP1 for binding to actin filaments [47]), was strongly downregulated during CaMV infection ($\log_2FC = -1.20$ to $-2.44$). None of these genes (*hin1* and *Rem1.2*) were deregulated by TuYV infection. However, another remorin, Rem1.3 (known to impair potato virus X movement [48]), was downregulated during TuYV infection in *C. sativa* var. *Celine* ($\log_2FC = -0.76$, up to $-0.82$) and during CaMV infection in both *A. thaliana* Col-0 and *C. sativa* var. *Celine* ($\log_2FC = -0.64$ in *A. thaliana* Col-0 and $\log_2FC = -1.15$, up to $-1.72$ in *C. sativa* var. *Celine*). On the other hand, the gene for myo-inositol-phosphate synthase 2 (*MIPS2*; AT2G22240) was downregulated under all conditions ($\log_2FC = -0.42$, up to $-3.28$), and *RTM3* (AT3G58350, known to block phloem movement of potyviruses [49]) was upregulated in TuYV- ($\log_2FC = 1.19$ to $1.93$) and downregulated in CaMV-infected hosts ($\log_2FC = -0.73$, up to $-2.00$). It is therefore conceivable that remorins and MIPS2 are factors controlling TuYV and CaMV movement. Curiously, the gene NSP-interacting kinase (*NIK1*; AT5G16000), which encodes a receptor-like kinase, involved in innate immunity-based defense response against a single-stranded DNA (ssDNA) geminivirus, was strongly downregulated in CaMV-infected ($\log_2FC = -0.79$, up to $-2.34$), but not in TuYV-infected, host plants. Considering that downregulation of *NIK1* should activate protein translation and could promote the accumulation of viral proteins (50), it could have a proviral effect during CaMV infection.

Next, we looked at salicylic acid (SA) synthesis, as this phytohormone is related to innate immunity-based defense responses against nonviral and viral pathogens, including CaMV (51, 52). Here, most genes were induced in both hosts and for both viruses (Fig. 8c), with the notable exception of *ICS2* (AT1G18870), which was downregulated in CaMV infection ($\log_2FC = -1.56$, up to $-2.09$) and slightly but nonsignificantly upregulated in TuYV infection, while its redundant orthologue *ICS1* (AT1G74710) was slightly but significantly downregulated only in CaMV-infected *A. thaliana* Col-0. Overall, genes involved in virus defense and SA biosynthesis were more strongly induced by CaMV infection than TuYV infection, whatever the host plant, indicating stronger pathogenicity of CaMV. The overexpression of SA-related genes in CaMV-infected plants could also reflect the ability of CaMV effector protein P6 to suppress SA-dependent autophagy, which might lead to compensatory feedback upregulation of SA genes (51). Deregulation of genes implicated in the SA pathway might have consequences on insect-plant interactions. In particular, increased SA could have a beneficial effect on aphid vector and possibly transmission because it can be concomitant with decreased JA levels and, consequently, with decreased plant defenses against aphids (53, 54).

**Insect defense-related genes responsive to CaMV and TuYV.** Next, we analyzed different metabolic pathways to determine if CaMV and TuYV infections modulate other hormones and secondary metabolites in ways that are more favorable for their aphid vector and, hence, for their transmission.

We first looked at jasmonic acid (JA) and its derivatives because they are plant-signaling molecules related to plant defense against herbivorous insects, microbial pathogens, and different abiotic stresses (Fig. 9a). We observed a strong virus-specific and host-independent effect for JA synthesis genes. CaMV downregulated many genes in the JA pathway, while TuYV upregulated some. Like for other pathways, the effect was

stronger in infected *C. sativa* var. *Celine* than in *A. thaliana* Col-0. Deregulated genes were, for example, *AOC1*, *AOC3*, and *AOC4* (3 out of 4 chloroplastic allene oxide cyclases involved in JA synthesis), *AOS* (AT5G42650, chloroplastic allene oxide synthase, involved in JA synthesis), and *LOX2* (AT3G45140, chloroplastic lipoxygenase required for wound-induced JA accumulation in *A. thaliana* Col-0). All of these genes were slightly upregulated in TuYV ($\log_2$FC = 0.32 to 0.68) and strongly repressed in CaMV-infected plants ($\log_2$FC up to $-6.82$ for Csa04g009860). This might imply that JA production is down in CaMV-infected plants and stable or slightly induced in TuYV-infected plants. JA is generally thought to decrease aphid growth and fecundity, so aphids on CaMV-infected plants might have greater fecundity. However, infection of *A. thaliana* Col-0 with CaMV lowered fecundity (Fig. 3a). JA-mediated signaling pathways are also known to increase proteins and secondary metabolites, which act as feeding deterrents (55). In this context, decreased JA production in CaMV-infected *A. thaliana* Col-0 could encourage longer/faster phloem sap ingestion, which we observed, indeed, in our experiments (Fig. 2a). Interestingly, phloem sap ingestion has been correlated with increased CaMV acquisition (30), which makes the JA pathway a major candidate for virus manipulation.

We also analyzed ethylene (ET) synthesis (Fig. 9b), as several studies have identified ET as a plant response against aphid infestation (56, 57). No specific gene expression patterns characteristic of a virus or a host were found, indicating that the ethylene response was unique for each virus-host pair. Notably, the ACC oxidase genes *ACO2* and *ACO5* (AT1G62380 and AT1G77330), involved in ethylene production, were strongly downregulated for CaMV-infected host plants ($\log_2$FC between $-0.97$ and $-2.69$). This is consistent with reduced accumulation of ethylene in CaMV-infected and P6-transgenic *A. thaliana* Col-0 in response to bacterial elicitors of innate immunity (51).

Glucosinolates (GLSs) are secondary metabolites that are produced by plants in the *Brassicaceae* family and set free in response to herbivore attacks (58). Some GLSs have been shown to be strong feeding deterrents for generalist aphids such as *M. persicae* (59) and might even have antibiosis effects on this aphid species (60, 61). CaMV infection downregulated genes involved in GLS synthesis (Fig. 9c), for example, the three *C. sativa* var. *Celine* orthologues encoding the cytochrome P450 monooxygenase CYP83A1 (AT4G13770) ($\log_2$FC between $-4.55$ and $-5.24$), whereas these three genes were upregulated in TuYV-infected *C. sativa* var. *Celine* ($\log_2$FC = 3.10 to 3.77). All in all, infection with CaMV predominantly downregulated transcription of GLS-related genes in *C. sativa* var. *Celine* and to a lesser extent in *A. thaliana* Col-0, whereas TuYV infection induced GLS synthesis in aphid-infested *C. sativa* var. *Celine* and had hardly any effect on *A. thaliana* Col-0. We therefore expected that *M. persicae* fitness and feeding behavior (i.e., phloem sap ingestion and ease to access phloem tissues) would be enhanced on CaMV-infected *C. sativa* var. *Celine* and *A. thaliana* Col-0 and be decreased on TuYV-infected *C. sativa* var. *Celine*. However, our fecundity experiments show that, on the contrary, *M. persicae* fecundity was decreased on CaMV-infected *A. thaliana* Col-0 and remained unchanged on TuYV-infected *A. thaliana* Col-0 compared to mock-inoculated plants, while no effects were observed on *C. sativa* var. *Celine* plants. Previous experiments indicated that *M. persicae* fecundity is even higher in TuYV-infected *C. sativa* var. *Celine* and lower in CaMV-infected *C. sativa* var. *Celine* (28). Note, however, that in the experiment of Chesnais et al. (28), a more severe CaMV strain was used, which might explain the discrepancies between both experiments. Overall, based on our results, deregulation of GLS-related genes after CaMV and TuYV infections does not seem to be the main factor controlling *M. persicae* fecundity. This is in line with another study that found no correlation between the GLS content of rapeseed and *M. persicae* fecundity (62).

On the other hand, our EPG experiments showed that aphids were able to reach phloem tissues and ingest phloem sap for a longer duration on CaMV-infected *A. thaliana* Col-0 and *C. sativa* var. *Celine* (see also reference 28). Therefore, downregulation of GLS genes might encourage aphid settling/feeding behavior on CaMV-infected plants and eventually promote CaMV acquisition by *M. persicae*. On TuYV-infected plants,

while some GLS-related genes were slightly upregulated, aphid feeding behavior was roughly equivalent to that on healthy plants, indicating that upregulations were not strong enough to induce feeding deterrence.

Camalexin is the major phytoalexin and has been shown to reduce the fecundity of aphids in *A. thaliana* Col-0 (63), although its effect on aphids might not be straightforward (64, 65). Phytoalexin-deficient 3 (PAD3) catalyzes the last step in its synthesis and CYP79B2, an important intermediate step. Here, we found contrasting effects of aphid infestation on virus-infected plants on camalexin-related genes (Fig. 9d). *PAD3* was downregulated in CaMV-infected (log$_2$FC less than −5.00), and to a lesser extent in TuYV-infected *C. sativa* var. *Celine* (log$_2$FC less than −0.75), and upregulated in *A. thaliana* Col-0 infected with CaMV or TuYV (log$_2$FC > 1.00). *CYP79B2* expression was unaffected in both *C. sativa* var. *Celine* and *A. thaliana* Col-0 infected with TuYV, but was substantially downregulated in CaMV-infected *C. sativa* var. *Celine* (log$_2$FC less than −2.5) and slightly upregulated in CaMV-infected *A. thaliana* Col-0 (log$_2$FC = 0.812). This indicates, for both genes, a strong host plant effect. Evidence indicates that PAD3 contributes more to camalexin synthesis than CYP79B2 (66–68). Thus, looking at *PAD3*, aphid fecundity should be higher on CaMV- and TuYV-infected *C. sativa* var. *Celine* and lower on infected *A. thaliana* Col-0. However, we observed a lower fecundity in CaMV-infected *A. thaliana* Col-0 and unchanged aphid fecundity in all other conditions, which suggests that aphid fecundity is not only linked to *PAD3* expression. On the other hand, phloem ingestion in both CaMV-infected *A. thaliana* Col-0 and *C. sativa* var. *Celine* increased, which is more in accordance with aphid plant acceptance. Overall, camalexin-related gene deregulations observed in both infected host plants did not seem to correlate with modified aphid fecundity or with aphid feeding behavior.

Callose is a polymer that is deposited by plants in between cells and in sieve tubes to restrict access of pathogens, including aphids, to tissues and phloem (69). We did not find any major DEG for this category except the stress-related plasma membrane respiratory burst oxidase RBOH F (70) and the pectin methylesterase inhibitor AT5G64640. No clear pattern of gene deregulation was observed, making interpretation difficult (Fig. S2). This might also be due to posttranslational modifications that majorly regulate RBOH F activity (71).

Since aphid lifestyle depends on compatible interactions with the phloem it feeds on, we looked at phloem protein expression (Fig. 10a). Under all conditions except aphid-infested, TuYV-infected *C. sativa* var. *Celine*, *PP2-B2* was the strongest induced gene (log$_2$FC > 3). *PP2-B2* codes for a phloem-specific lectin-like protein with an unknown function containing an F-box domain and a potential myristoylation site (72) that could control membrane localization. Specific for CaMV infection, the putative phloem lectin genes *PP2-B1* (AT2G02230) and *PP2-B5* (AT2G02300) were upregulated in *A. thaliana* Col-0 (log$_2$FC = 3.07 and 7.89, respectively), and one of their orthologues was upregulated in *C. sativa* var. *Celine* (log$_2$FC = 1.38 for Csa06g002480). The putative calcium-, lipid-, and RNA-binding phloem protein PP16-1 (AT3G55470) was, independent of the virus, upregulated in infected *A. thaliana* Col-0 and downregulated in *C. sativa* var. *Celine*. One *C. sativa* var. *Celine* orthologue of the *A. thaliana* Col-0 PP2-A1, known to repress aphid phloem feeding (73), was downregulated in *C. sativa* var. *Celine* infected with both CaMV and TuYV (log$_2$FC = −1.60 and −1.18, respectively), but in *A. thaliana* Col-0, this gene remained nonresponsive to viral infections. It is worth mentioning that PP2 proteins of cucurbits could potentially bind to viral particles of CABYV (genus *Polerovirus*, like TuYV) and increase virus stability in the aphid gut (74). Proteins of this type could therefore have double importance due to their role in vector aphids' feeding behavior and their possible involvement in virus transmission. Deregulation of most other phloem proteins did not follow a distinct pattern, and the unknown functions of most of these genes precluded any interpretation.

CalS7 (AT1G06490), a phloem-specific callose synthase responsible for wounding stress-induced callose deposition onto sieve tube plates and, hence, phloem plugging (75), was slightly upregulated in TuYV-infected *C. sativa* var. *Celine* (log$_2$FC = 0.9) and

downregulated in CaMV-infected *A. thaliana* Col-0 (log$_2$FC = −0.91). The same trend (upregulation in TuYV-infected *C. sativa* var. *Celine* and *A. thaliana* Col-0, downregulation in CaMV-infected *C. sativa* var. *Celine*) applied to the phloem-located sucrose synthases SUS5 and SUS6 (AT5G37180 and AT1G73370, respectively) that interact with CalS7 (76). Also, *SLI1* (AT3G10680), a gene coding for a phloem small heat shock-like protein known to be involved in resistance to *M. persicae* and other phloem feeders (77), was downregulated in both *A. thaliana* Col-0 and *C. sativa* var. *Celine* infected by CaMV (log$_2$FC = −0.88 in *A. thaliana* Col-0 and up to −1.62 in *C. sativa* var. *Celine*). This might indicate that CaMV infection, but not TuYV infection, favors phloem feeding of aphids by perturbing stress-related callose deposition on sieve plates. This is in line with the prolonged phloem ingestion observed for *M. persicae* on CaMV-infected plants (Fig. 2).

Next, we examined the expression of genes known to be involved in plant responses and defenses against insects (Fig. 10b), as their modulation could influence virus-insect interactions and, hence, transmission. General trends were suppression in CaMV-infected *C. sativa* var. *Celine* and activation in CaMV-infected *A. thaliana* Col-0 and TuYV-infected *C. sativa* var. *Celine* and *A. thaliana* Col-0, resulting in both host-specific and virus-specific responses. *ESM1* (AT3G14210) was strongly downregulated in both CaMV-infected hosts (log$_2$FC less than −1.9), but not in TuYV-infected hosts. Its gene product biases the production of glucosinolates, and its knockout mutant is more susceptible to herbivory by the caterpillar *Trichoplusia ni* (78). Thus, its downregulation in CaMV-infected plants might favor aphid colonization. The expression of ATWSCP (AT1G72290), a protease inhibitor and water-soluble chlorophyll-binding protein, was strongly upregulated in TuYV-infected (log$_2$FC > 3.4) and downregulated in CaMV-infected *C. sativa* var. *Celine* (log$_2$FC less than −2.7), whereas its expression was unchanged in *A. thaliana* Col-0. The apoplastic ATWSCP, together with the protease RD21, protects plants, especially greening plants, against herbivory (79). Whether it also acts against aphids is unknown. The *M. persicae*-induced lipase 1 (MPL1; AT5G14180) was upregulated in TuYV-infected *C. sativa* var. *Celine* (log$_2$FC > 2.4) and CaMV-infected *A. thaliana* Col-0 (log$_2$FC = 3.11) but downregulated in TuYV-infected *A. thaliana* Col-0 (log$_2$FC = −1.42) and unaffected in CaMV-infected *C. sativa* var. *Celine*. This gene is induced by aphid infestation and decreases aphid fecundity, but its enhanced expression does not change aphid behavior or plant choice (80). Whether the reduced fecundity of *M. persicae* on CaMV-infected *A. thaliana* Col-0 is partially due to the action of this gene remains an open question. Strong host plant-specific and virus-specific effects were found for VSP2 (AT5G24770, reported to have a role in defense against herbivorous insects [81]). Expression was upregulated in aphid-infested TuYV-infected *C. sativa* var. *Celine* (log$_2$FC > 2) and downregulated in aphid-infested CaMV-infected *C. sativa* var. *Celine* (log$_2$FC = −8) but was not affected in infested *A. thaliana* Col-0. All in all, plant defense responses against insects did not follow a clear pattern. This was probably due to the very divergent pathways and the heterogeneity of the plants' insect response genes.

**Concluding remarks.** In this work, we analyzed the effect of CaMV and TuYV infection of *M. persicae* aphid-infested *A. thaliana* Col-0 and *C. sativa* var. *Celine* on the plant hosts' transcriptomes as well as on the fecundity and feeding behavior of their vector *M. persicae*.

Our results show that CaMV infection caused more severe effects on the phenotype of both plant species than did TuYV infection (Fig. 1). The severity of symptoms correlated strongly with the proportion of DEGs (41 to 43% for CaMV, 5 to 11% for TuYV; Fig. 4e). CaMV infection affected the same percentage of genes in both plant hosts, whereas TuYV infection deregulated proportionally twice as many genes in *C. sativa* var. *Celine* than in *A. thaliana* Col-0. Again, this correlated with stronger visible symptoms on TuYV-infected *C. sativa* var. *Celine* than with TuYV-infected *A. thaliana* Col-0. Aphid performance changes were more pronounced on CaMV-infected hosts, whatever the plant species, than those caused by TuYV infection. Despite more DEGs in

TuYV-infected *C. sativa* var. *Celine* than in TuYV-infected *A. thaliana* Col-0, aphid behavior was slightly more impacted on TuYV-infected *A. thaliana* Col-0 (Fig. 2). This likely indicates modification of plant metabolites that cannot be identified by transcriptome profiling. A metabolomic analysis of virus-infected leaves or phloem sap should provide complementary data on the aphid-plant-virus interactions.

In this study, we did not compare the contribution of aphid infestation alone to the plant transcriptome. However, recent work (82) on the transcriptome changes of healthy *A. thaliana* Col-0 plants infested or not with *M. persicae* for 72 h identified a limited number of DEGs (265), suggesting that the contribution of aphid infestation to the transcriptome changes in healthy and probably also in virus-infected and infested plants is minor. Although it is difficult to directly compare their data with ours, we looked for *A. thaliana* Col-0 genes that were upregulated by aphid infestation alone but downregulated by aphid infestation plus virus infection. The rationale was that these genes might reflect viral effects to reduce the host plant's capability to recognize aphid infestation and establish defenses, thus favoring aphid infestation. For TuYV, only one gene that was upregulated by aphid infestation alone was downregulated by concomitant TuYV infection (the transcription factor DREB26; AT1G21910). In contrast, 36 genes were downregulated by CaMV while being upregulated by *Myzus* infestation alone (none inversely). GO analysis of these genes indicated an enrichment of genes related to "response to chitin," "response to salicylic acid," "response to salt stress," "response to wounding," "hormone-mediated signaling pathway," "defense response to fungus," "regulation of defense response," and "signal transduction" (see Table S3). This indicates that at least CaMV might dampen plant perception of aphid infestation and defenses against aphids, which might manifest itself in that aphids reach the phloem faster and feed longer on CaMV-infected plants. The fact that *Myzus* fecundity was lower on these plants might be explained by the profound changes in other GOs, especially photosynthesis and carbohydrate metabolism, which probably reduce the nutritional value of CaMV-infected plants. Some of these genes could merit further exploration.

The most pronounced effect of CaMV infection on plant hosts was a strong downregulation of photosynthesis genes (Fig. 6) and carbohydrate metabolism-related genes (Fig. 7). We observed significant changes in many other pathways, including categories that are likely affecting virus-vector interactions (i.e., defenses, silencing, hormones, secondary metabolites, etc.). However, the impact of these modifications on aphid fitness or feeding behavior was not easy to evaluate since these parameters are likely under the control of several, often overlapping, metabolic pathways. Trying to correlate the effect of specific genes on aphids as reported in the literature with our aphid behavior observations therefore often resulted in contrasting results. We offer the following explanations. The very strong alterations in photosynthesis might have drowned out otherwise visible effects of DEGs previously found to be involved in plant-aphid interactions. Another explanation is regulation by posttranscriptional and posttranslational modifications. While transcriptome profiling is a powerful tool, it can display only changes in transcript levels. In many cases, however, posttranslational modifications of proteins (such as phosphorylation, localization, complex formation, and many more) and even posttranscriptional RNA modifications (sequestering of RNAs in processing bodies [P-bodies] and others) will contribute to phenotype changes. Depending on the pathway, the contribution of the transcriptome and posttranscriptome to cellular processes and beyond will vary. This again indicates that complementary analyses such as metabolomics, proteomics, etc., might help to gain a more complete insight.

Nevertheless, we observed that virus infections have very distinct effects on the transcriptome of host plants and that, as expected, the non-phloem-limited virus (i.e., CaMV) has a significantly stronger impact on plant hosts than the phloem-limited virus (i.e., TuYV). Overall, viral infection with CaMV tends to have effects on metabolic pathways with strong potential implications for insect vector-plant host interactions, while TuYV only weakly alters

**TABLE 1** List of promising genes for functional validation[a]

| Gene locus | Gene | Trend of regulation | Function/pathway and potential effect on aphids | Reference |
|---|---|---|---|---|
| AT2G18700 | TPS11 | Down in all infected plants | Promotes synthesis of starch, which is an aphid antifeedant | 38 |
| AT3G25760 | AOC1 | Down in CaMV-infected plants, up in TuYV-infected plants | Involved in JA synthesis, central regulator of plant antiherbivore defenses | 93 |
| AT3G25780 | AOC3 | Down in CaMV-infected plants, up in TuYV-infected plants | Involved in JA synthesis, central regulator of plant antiherbivore defenses | 93 |
| AT1G13280 | AOC4 | Down in CaMV-infected plants, up in TuYV-infected plants | Involved in JA synthesis, central regulator of plant antiherbivore defenses | 93 |
| AT5G42650 | AOS | Down in CaMV-infected plants, up in TuYV-infected plants | Involved in JA synthesis, central regulator of plant antiherbivore defenses | 93 |
| AT3G45140 | LOX2 | Down in CaMV-infected plants, up in TuYV-infected plants | Involved in JA synthesis, central regulator of plant antiherbivore defenses | 93 |
| AT5G24770 | VSP2 | Down in CaMV-infected plants, up in TuYV-infected plants | JA-responsive gene | 94 |
| AT4G03560 | TPC1 | Down in all infected plants | Aphid recognition by calcium elevations | 95 |
| AT1G62380 | ACO2 | Down in CaMV-infected plants | Involved in production of ethylene, signaling of aphid feeding | 96 |
| AT1G77330 | ACO5 | Down in CaMV-infected plants | Involved in production of ethylene, signaling of aphid feeding | 96 |
| AT4G39030 | EDS5 | Up in all infected plants | Involved in SA synthesis/signaling, might reduce aphid resistance | 97 |
| AT4G13770 | CYP83A1 | Down in CaMV-infected plants, up in TuYV-infected plants | Involved in GLS synthesis, an aphid feeding deterrent | 98 |
| AT1G18570 | MYB51 | Up in TuYV-infected *A. thaliana* Col-0, down in CaMV-infected *A. thaliana* Col-0 | Involved in indole GLS synthesis, an aphid feeding deterrent. | 61 |
| AT5G14180 | MPL1 | Up in TuYV-infected *C. sativa* var. *Celine*, down in TuYV-infected *A. thaliana* Col-0 | Positive regulator of defense against aphids | 80 |
| AT3G10680 | SLI1 | Down in CaMV-infected plants | R gene that confers a broad-spectrum quantitative resistance to phloem-feeding insects | 77 |

[a]Genes were chosen because they could be involved in phenotypic modifications of plants and changes in behavior and performance of aphids.

these pathways. For example, the strong gene downregulations in the jasmonic acid, ethylene, and glucosinolate biosynthetic processes (Fig. 9a through c) in CaMV-infected plants could be responsible for the observed alterations in aphid feeding behavior and performance. The next steps could consist of functional validation of some candidate genes identified in our study for their role in viral manipulation and, consequently, potential impacts on viral transmission (Table 1).

## MATERIALS AND METHODS

**Aphids.** A Dutch green peach aphid clone (*Myzus persicae* Sulzer 1776) was used for the experiments. It was reared on Chinese cabbage (*Brassica rapa* L. *pekinensis* var. *Granaat*) in a growth chamber at 20 ± 1°C and a 16-h photoperiod. Only wingless forms were used in assays. For synchronization, adults were placed on detached Chinese cabbage leaves that were laid on 1% agarose in a petri dish. The adults were removed 24 h later, and the newborn larvae were used in experiments 5 days later (for transcriptomic experiments) or 8 days later (for feeding behavior and performances experiments).

**Viruses.** CaMV isolate Cm1841-Rev (22), which is a transmissible derivative of isolate Cm1841 (83), and TuYV isolate TuYV-FL1 (84) were maintained in *A. thaliana* Col-0 and propagated by aphid inoculation of 2-week-old plants. Growth conditions were as described below.

**Virus infection and aphid infestation.** Seeds of *Arabidopsis thaliana* Col-0 and *Camelina sativa* var. *Celine* were germinated in TS3 fine substrate (Klasmann-Deilmann) in 7- by 7-cm pots and watered with tap water. Growth conditions were 14-h day and 10-h night with LED illumination and a constant temperature of 21 ± 1°C. Two-week-old plants were inoculated with 3 to 5 wingless *Myzus persicae* aphids that had been allowed a 24-h acquisition access period on *A. thaliana* Col-0 infected with TuYV or CaMV or on healthy *A. thaliana* Col-0. Plants were individually wrapped in clear plastic vented bread bags to prevent cross-contamination. Aphids were manually removed after a 48-h inoculation period. Eighteen days postinoculation (dpi), 25 to 30 5-day-old nonviruliferous aphids were placed for infestation on the rosette (*A. thaliana* Col-0) or the apical leaves (*C. sativa* var. *Celine*) of CaMV- or TuYV-infected or mock-inoculated plants. After 72 h infestation (equal to 21 dpi), aphids were removed with a brush. The infested plants (virus infected or mock inoculated) were washed 3 times with deionized water and 3 times with Milli-Q water to remove any remaining aphid exuviae and honeydew. Then, rosettes (*A. thaliana* Col-0) or detached leaves (*C. sativa* var. *Celine*) were collected in 50-mL Falcon tubes. Three

biological replicates were used for analysis. For *A. thaliana* Col-0, 1 biological replicate consisted of 4 plants, and for *C. sativa* var. *Celine*, 1 replicate was 3 plants. Plant samples were conserved at −80°C until processing.

**RNA purification and Illumina sequencing.** Total RNA was extracted from 1 g of *A. thaliana* Col-0 (rosettes) and *C. sativa* var. *Celine* (leaves) frozen tissues using a cetyltrimethylammonium bromide (CTAB)-LiCl protocol (85) modified as described in detail by Golyaev et al. (86). Briefly, the plant material was ground in liquid nitrogen, homogenized in 10 mL CTAB buffer, and centrifuged for 10 min at 5,000 × *g* and 4°C. The supernatant was mixed with one volume of chloroform/isoamyl alcohol (24:1) followed by nucleic acid precipitation with 0.1 volume of 3 M sodium acetate (NaOAc, pH 5.2) and 0.6 volume of isopropanol, incubation at −20°C for 1 h, and centrifugation for 20 min at 20,000 × *g* and 4°C. The pellet was resuspended in 1 mL of RNase-free water followed by selective precipitation of RNA by addition of 0.3 volume of 10 M LiCl, overnight incubation at 4°C, and centrifugation for 30 min at 20,000 × *g* and 4°C. The RNA pellet was resuspended in 0.1 mL of RNase-free water, 0.1 volume of 3 M NaOAc (pH 5.2), and 2 volumes of cold absolute ethanol, centrifuged for 20 min at 20,000 × *g* and 4°C, washed with ice-cold 70% ethanol, air-dried, and dissolved in 50 $\mu$L RNase-free water.

Total RNA samples were subjected to quality control and Illumina sequencing at Fasteris (www .fasteris.com) using a standard stranded mRNA library preparation protocol and multiplexing the resulting 18 libraries (3 biological replicates per each of the 6 conditions) in one NovaSeq flow cell, SP-200, with 2× 75-nt paired-end customized run mode. The resulting 75-nt reads from each library were mapped with or without mismatches to the reference genomes of *A. thaliana* Col-0 (TAIR10.1 nuclear genome [5 chromosomes], chloroplast [Pltd], and mitochondrion [NT]; https://www.ncbi.nlm .nih.gov/genome/?term=txid3702[Organism:noexp]), *C. sativa* var. *Celine* (nuclear genome [20 chromosomes]; https://www.ncbi.nlm.nih.gov/genome/?term=txid71323[Organism:exp]), CaMV (strain CM1841rev [22]), and TuYV (GenBank accession no. NC_003743) (84). Note that the CaMV reference sequence was extended at the 3′ end by 74 nt from the 5′ terminus to account for its circular genome and allow for mapping reads containing the first and last nucleotides of the linear sequence. In the case of TuYV, some discrepancies with the reference sequence were detected when the reads were mapped to the viral reference sequence. Therefore, the reads were used to generate a new consensus master genome in the viral quasispecies population. For both viruses, the consensus genome sequences (see Sequence information S1 in the supplemental material) were used for (re)mapping and counting total viral reads as well as viral reads representing forward and reverse strands of the viral genomes (Data Set S1).

**RT-qPCR.** The expression of *A. thaliana* Col-0 genes was monitored by RT-qPCR analysis. cDNA was synthesized from 10 $\mu$g total RNA using AMV reverse transcriptase (Promega) and oligo(dT). Real-time qPCRs were completed in the LightCycler 480 instrument (Roche) using the Sybr Green master mix (Roche) and following the recommended protocol. Each reaction mixture (10 $\mu$L) included 3 $\mu$L of cDNA and 0.5 $\mu$L of 10 $\mu$M primers (Table S1). The thermocycler conditions were as follows: preincubation at 95°C for 5 min, followed by 40 cycles of 95°C for 10 s, 58 to 60°C for 20 s, and 72°C for 20 s. The expression was normalized to the *Arabidopsis* internal reference gene *PEX4* (AT5G25760) (Table S1).

**Raw data processing and quality control for transcriptome profiling.** Processing was carried out on the Galaxy France platform (https://usegalaxy.fr/) (87). Raw read quality was checked with FastQC (v0.11.8), and the results were then aggregated with MultiQC (v1.9). For *A. thaliana* Col-0, between 58.6 and 69.4 million 75-nt paired-end reads were sequenced with a mean phred score of >30 for all bases. For *C. sativa* var. *Celine*, between 56.4 and 77.6 million 75-nt paired-end reads were sequenced with a mean phred score >30 for all bases. In all samples, there were no overrepresented sequences and very few adapters (0.15% of adapter for the last bases of reads). Reads were aligned on the reference genome with STAR (v2.7.6a) using default parameters and quality checked again with MultiQC. Between 80% and 92.3% of reads were uniquely mapped for *A. thaliana* Col-0 samples, and between 60.8% and 70.5% of reads were uniquely mapped for *C. sativa* var. *Celine*. Between 17% to 20% of reads mapped to multiple loci in *C. sativa* var. *Celine* because of the triplication event of this genome. Reference genomes were *Camelina*_sativa.Cs.dna.release-49 and *Arabidopsis*_thaliana.TAIR10.49 from the EnsemblPlant portal. Gene counts were obtained with featureCounts (v2.0.1). This option allows reads to map to multiple features for *C. sativa* var. *Celine*. We assigned 92.2% to 93.3% of uniquely aligned reads to a gene for *A. thaliana* Col-0, and 80.7% to 88.6% aligned reads were assigned to a gene for *C. sativa* var. *Celine*. Differential gene expression was then analyzed with SARTools (v1.7.3) and the DESeq2 method (i.e., TuYV-infected plants versus mock inoculated, CaMV-infected plants versus mock inoculated). GO enrichment analysis was performed with GOseq (v1.36.0) on the DEGs.

To measure viral RNA loads in plants, the RNA-seq reads from each sample were mapped to the reference genome sequences of the host plant (*A. thaliana* Col-0 or *C. sativa* var. *Celine*) and the virus (CaMV or TuYV) with zero mismatches, and the mapped reads were sorted by polarity (forward, reverse, and total) and counted. Viral read counts were then normalized per million of total (viral plus nonviral) or plant reads (see Data Set S1).

**Western blotting.** Total protein extracts were prepared from leaves, separated by SDS-PAGE, and transferred to nitrocellulose as described previously (22). Western blots were performed using antisera raised against CaMV P4 (88) and TuYV CP (89). Secondary antibodies were horseradish peroxidase conjugates, and bound antibodies were revealed by enhanced chemiluminescence.

**Aphid feeding behavior.** To investigate the effects of TuYV and CaMV plant infections on the feeding behavior of *M. persicae*, we used the electrical penetration graph technique (EPG) (90). Eight adult aphids were connected to a Giga-8 dendritic cell (DC) system (EPG Systems; www.epgsystems.eu) and

placed on the leaf of an individual experimental host plant. To create electrical circuits that each included a plant and an aphid, we tethered each insect by attaching a 12.5-$\mu$m-diameter gold wire to the pronotum using conductive water-based silver glue. The whole system was set up inside a Faraday cage located in a climate-controlled room held at 21 $\pm$ 1°C and under constant LED illumination during recording. Plants were obtained as described in the previous section, but, unlike the plants used for the RNA-seq experiment, the plants used in EPG were not preinfested. We used the PROBE 3.5 software (EPG Systems) to acquire and analyze the EPG waveforms as described (91). Relevant EPG variables were calculated with EPG-Calc 6.1 software (92). We chose variables based on five different EPG waveforms corresponding to "probing duration," "stylet pathway phase," "phloem sap ingestion," "time to first phloem sap ingestion," and "salivation in phloem sap." For each aphid-plant-virus combination, we collected 8-h recordings from 20 to 23 individuals.

**Aphid fecundity.** To investigate the effects of TuYV and CaMV plant infections on the fecundity of *M. persicae*, we randomly selected synchronized wingless adults (8 $\pm$ 1 day old) and transferred them onto experimental host plants. For *A. thaliana* Col-0 experiments, we used one plant per aphid, and we covered the pots with vented bread bags. For *C. sativa* var. *Celine* experiments, to force aphids to settle on symptomatic leaves, we placed adults on detached leaves that were laid on 1% agarose in a petri dish. The number of nymphs produced per adult was recorded after 5 days. We discarded from the analysis the adult aphids that died before the end of the experiment. Data on both *Arabidopsis* and *C. sativa* var. *Celine* host plants were collected in three repetitions, comprising altogether 27 to 33 aphids per aphid-plant-virus combination.

**Statistical analyses of aphid behavior and fecundity.** Data on aphid feeding behavior were analyzed using generalized linear models (GLMs) with a likelihood ratio and the chi-square ($\chi^2$) test. Since duration parameters (i.e., probing duration, stylet pathway phase, phloem sap ingestion, and salivation) were not normally distributed, we carried out GLMs using a gamma (link, inverse) distribution. For the "time-to-first phloem" phase, we used the Cox proportional hazards model, and we treated cases where the given event did not occur as censored. The assumption of validity of proportional hazards was checked using the functions coxph and cox.zph, respectively (R package survival). For aphid fecundity, count data were not normally distributed. Accordingly, we carried out a GLM using a Poisson distribution, a quasilikelihood function was used to correct for overdispersion, and log was specified as the link function in the model. When a significant effect of one of the main factors was detected or when an interaction between factors was significant, a pairwise comparison using estimated marginal means (R package emmeans) (*P* value adjustment with Tukey method) at the 0.05 significance level was used to test for differences between treatments. The fit of all GLMs was controlled by inspecting residuals and quantile-quantile (QQ) plots. All statistical analyses were performed using R software v4.0.4 (https://www.r-project.org/).

**Data availability.** The raw RNA-seq data are available under BioProject number PRJEB49403 and at the European Nucleotide Archive (https://www.ebi.ac.uk/ena/browser/view/PRJEB49403).

## SUPPLEMENTAL MATERIAL

Supplemental material is available online only.
**SUPPLEMENTAL FILE 1**, PDF file, 1 MB.
**SUPPLEMENTAL FILE 2**, XLSX file, 0.04 MB.
**SUPPLEMENTAL FILE 3**, XLSX file, 0.2 MB.
**SUPPLEMENTAL FILE 4**, XLSX file, 0.01 MB.

## ACKNOWLEDGMENTS

We thank Claire Villeroy for aphid rearing and the experimental unit of INRAE Grand Est-Colmar (UEAV) for help with plant production.

This work was supported by a public grant overseen by the French National Research Agency (ANR) (reference ROME ANR-18-CE20-0017-01). Quentin Chesnais was supported by Région Grand Est (Soutien aux jeunes chercheurs, reference 18_GE5_013). The funding sources had no role in the study design; collection, analysis, and interpretation of data; writing of the report; and the decision to submit the article for publication.

We declare no conflict of interest.

Conceptualization, Q.C., V.B., M.M.P., and M.D.; methodology, Q.C., V.G. and M.D.; software, Q.C., A.V., and C.R.; validation, Q.C. and V.G.; formal analysis, Q.C., A.V., C.R., and M.D.; investigation, Q.C. and V.G.; data curation, Q.C., A.V., and C.R.; writing – original draft preparation, Q.C., M.M.P., and M.D.; writing – review & editing, Q.C., A.V., C.R., V.B., M.M.P., and M.D.; visualization, Q.C.; supervision, M.M.P. and M.D.; project administration, M.D.; and funding acquisition, M.M.P. and M.D.

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
