## [Reviewer comments · Microbiology Spectrum]

Microbiology Spectrum

Comparative plant transcriptome profiling of *Arabidopsis* and *Camelina* infested with *Myzus persicae* aphids acquiring circulative and non-circulative viruses reveals virus- and plant-specific alterations relevant to aphid feeding behavior and transmission

Quentin Chesnais, Victor Golyaev, Amandine Velt, Camille Rustenholz, Véronique Brault, Mikhail Pooggin, and Martin Drucker

Corresponding Author(s): Martin Drucker, SVQV INRAE Centre Grand Est Colmar

Review Timeline:

Submission Date:	January 12, 2022
Editorial Decision:	April 6, 2022
Revision Received:	June 17, 2022
Accepted:	June 19, 2022

Editor: Lindsey Burbank

Reviewer(s): Disclosure of reviewer identity is with reference to reviewer comments included in decision letter(s). The following individuals involved in review of your submission have agreed to reveal their identity: Kathleen Martin (Reviewer #2)

Transaction Report:

DOI: <https://doi.org/10.1128/spectrum.00136-22>

April 6, 2022

Dr. Martin Drucker
INRAE Centre Grand Est Colmar
Colmar
France

Re: Spectrum00136-22 (Comparative plant transcriptome profiling of Arabidopsis and Camelina infested with Myzus persicae aphids acquiring circulative and non-circulative viruses reveals virus- and plant-specific alterations relevant to aphid feeding behavior and transmission)

Dear Dr. Martin Drucker:

Thank you for submitting your manuscript to Microbiology Spectrum. The reviewers and myself appreciate the scope of your study and the interesting results presented. The reviewers have also provided a number of recommendations for improving clarity and making the manuscript easier to understand. Please address the reviewers recommendations outlined below and in the attached PDF file to submit a revised manuscript.

Link Not Available

Sincerely,

Lindsey Burbank

Journals Department
Reviewer comments:

Reviewer #1 (Comments for the Author):

This is a technically sound and interesting paper that presents data on vector behavior/fecundity in relation to infected and healthy hosts, contrasting two very different viruses each infecting two related, but distinct host species. This is followed up with a transcriptomic analysis of a different set of infected and healthy hosts of each species x virus combination in the same stage of growth/disease progression as that used for behavior/fecundity experiments. I have just a few suggestions for improvement of the manuscript.

1. It would be nice to know why virus replication levels were only quantified through the transcriptome data, and not using an RT-qPCR assay that avoids the issue of TuYV not having poly A tails. Although differences were still found, it would be good if these were verified with a more standard diagnostic method for evaluating virus replication in the host.
2. The results section is very long and dense. I would suggest breaking it up into more sub-sections, with each heading specifying the main finding of that sub-section. This is especially the case for the heatmap analyses. It is difficult to get the main picture from this section or to be able to revisit certain pathways discussed without some kind of headings as a roadmap.
3. The authors suggest that functional validation of candidate gene involvement in virus-induced phenotypes and effects on aphid behavior could be a future step. However, it is very hard to pick out what targets are most promising from this analysis because of the density of the text. The authors could provide a summary table relating behavior/fecundity effects to key changes recorded in the transcriptome analysis. This could highlight changes that could underlie the behavior/fecundity response and those that run counter to it, with relevant literature citations.

Reviewer #2 (Comments for the Author):

General Comments:

The main question of this paper is to address if RNA-Seq can tell if plant specific or virus-specific responses to insect feeding with the same vector insect can be detected. There are 6 different conditions including TuYV in Arabidopsis and Camelina, CaMV in Arabidopsis and Camelina and no virus in Arabidopsis and Camelina. There is a lot of data to go through and the authors present the majority of their RNA-Seq data as heat maps of only the relevant conditions, pathways and genes of interest between each of the plants and viruses. There is also inclusion of the general RNA-Seq data points of concern including total numbers of reads, total numbers of upregulated vs downregulated genes, etc. The authors also tested a number of aphid specific conditions such as feeding activity and fecundity. This is a very thorough study which is original in the terms of the breadth of coverage and the number of conditions tested to compare to each other. This reviewer is astounded and excited about the authors undertaking this enormous project and the sheer amount of data that was produced.

Paper is moderately well written, there are a number of small things that could clarify and make the paper more readable. Due to the sheer amount of data presented, it is easy to get lost and distracted in the paper on avenues which can be difficult to follow. Difficult in the sense that it required movement not only through the paper, but also through the extensive amount of supplementary data as well. Authors do not clarify the use of some terms used often in the results section and this could be much improved by simply putting the log2fold in parenthesis next to the transcript in question. For example: Line 481- "The most down-regulated gene in CaMV-infected Arabidopsis, AT3G27690, encoding a protein LHCb2.4, a component of the light harvesting complex, was also strongly repressed in Camelina infected by CaMV". In this example, the log2fold of the Arabidopsis gene could be put in the parenthesis next to that gene and the log2fold of the Camelina could also be put in parenthesis next to that reference. The author uses terms like "strongly downregulated", "strongly repressed", "strongly induced" and so on with no clear definition on what that means to them. Simply including the numbers in the writing when referencing the gene itself would clarify the writing substantially and not require the reader to constantly look back at the supplementary data which is exhausting and time consuming. Considering that the authors do not talk about all of the changes in gene expression- of which there were many- adding the numbers to the ones that were deemed important enough to talk about in the writings will make this paper much more understandable as to what the author means by the use of these general and subjective terms.

In the methods, please clarify why the use of 18 day time points were utilized for both viruses for the placement of insects. Was there additional tests done to determine at what stage the virus was in these two plants at this time point? Clarification as to what lead to this decision would lead to more information as to how this may impact the genes upregulated or downregulated later. Detail when symptoms first appear on plants of both types of both viruses to determine if the timing of infection is the same or if one may be later or earlier in infection status than the other.

There is still some disconnect between the EPG data and the RNA-Seq. There is also some disconnect in the fecundity and the RNA-Seq as well. This appears most prominent in the discussion and when these topics are not discussed in the concluding remarks in relation to the RNA-Seq.

Specific Comments:

Confirmation of the dataset through qRT-PCR resulted in only validation of the Arabidopsis results, no results presented for Camelina. Two of the transcripts for Arabidopsis in TuYV do not agree, however, the authors state it was due to weak expression changes. Did the authors explore alternative splicing transcripts at all? Why were these transcripts selected for validation if two could not pass due to issues? Why was Camelina ignored for validation? Was this due to the number of multiple gene forms in the genome?

In the concluding remarks- Line 796-the authors state that an earlier study looked at only the insect feeding and noticed the

response was much less than their current study. Given that the reviewer was expecting these conditions to be tested in this study as a further control, this reviewer feels that should be elaborated on to include any gene changes that were shared between the studies. Since it was in Arabidopsis, the Gene IDs should be the same and could be compared to look at it instead of just mentioning it offhand.

Introduction contains all relevant background required to determine the importance of the study, suggest no changes.

References seem well ordered and formatted, suggest no changes.

Scientific method and experiments are well conducted, suggest addressing why Camelina was not confirmed via qRT-PCR.

Also include the above mentioned comment about why 18 days was utilized as a time point.

Staff Comments:

Preparing Revision Guidelines

Please return the manuscript within 60 days; if you cannot complete the modification within this time period, please contact me. If you do not wish to modify the manuscript and prefer to submit it to another journal, please notify me of your decision immediately so that the manuscript may be formally withdrawn from consideration by Microbiology Spectrum.

Comparative plant transcriptome profiling of Arabidopsis and Camelina
infested with *Myzus persicae* aphids acquiring circulative and non-circulative
viruses reveals virus- and plant-specific alterations relevant to aphid feeding
behavior and transmission

Quentin Chesnais ^{1,a}, Victor Golyaev ^{2,a}, Amandine Velt ¹, Camille Rustenholz ¹, Véronique
Brault ¹, Mikhail Pooggin ^{2,b}, Martin Drucker ^{1,b}

¹ SVQV, UMR1131, INRAE Centre Grand Est – Colmar, Université Strasbourg, France

² PHIM, INRAE Centre Occitanie – Montpellier, CIRAD, IRD, Université Montpellier, Institut
Agro, France

14 ^a Contributed equally

15 ^b Correspondence: martin.drucker@inrae.fr, mikhail.pooggin@inrae.fr

16

17 **Abstract**

[revised manuscript text omitted]

The transmission mode of non-circulative viruses such as cucumber mosaic virus (CMV, genus
*Cucumovirus*), that are also transmitted by aphids and other hemipteran vectors, is entirely different
(for review Ng and Falk, 2006)). They are mostly acquired and inoculated during early probing
phases [i.e. intracellular penetrations in the epidermis and mesophyll tissues (Martin et al., 1997)].
They do not invade aphid cells but are retained externally in the mouthparts (stylets and/or
esophagus), from where they are also released into a new host. For this reason, non-circulative
viruses are acquired and inoculated within seconds to minutes, and vectors retain and transmit the
virus only for a limited time (minutes range). Therefore, this transmission mode is also named non-
persistent transmission. Some non-circulative viruses may be retained by the vectors during several
84 hours and are referred to as semi-persistent viruses. The aphid-transmitted cauliflower mosaic virus
(CaMV, genus *Caulimovirus*) belongs to this group (Kennedy et al., 1962; Moreno et al., 2012).

Available data indicate that many viruses modify host traits (i.e. color, volatiles,
primary/secondary metabolites etc.) in ways that are conducive for their transmission (for review
Dáder et al., 2017; Fereres and Moreno, 2009; Mauck et al., 2012). Theoretical considerations
suggest that these modifications depend on the virus species and in particular on the mode of virus
transmission by vectors (Dáder et al., 2017; Mauck et al., 2012). Circulative, persistent and phloem-
restricted viruses should benefit from faster access of vectors to the phloem and longer feeding,
which would promote both virus acquisition and inoculation. In addition, these viruses would tend
to improve nutrient quality of the host and consequently vector fitness and fecundity, concomitant
with an increased number of viruliferous vectors (Dáder et al., 2017; Fereres and Moreno, 2009;
Mauck et al., 2018). Both modifications have been reported for aphid-transmitted luteoviruses
(Bosque-Pérez and Eigenbrode, 2011). Non-persistent or semi-persistent, non-circulative viruses
with their fast transmission kinetics are expected to benefit from the attraction of vectors to infected
plants, followed by a rapid dispersion, before the virus is lost from the vector during subsequent
salivation events. The best-studied example is CMV, where altered volatiles incite aphids to alight
on infected plants and acquire CMV, before the poor taste and low nutritive value encourage the

aphids to leave and transmit the virus, attached to the stylets during this brief probing time, to other
(healthy) host plants (Carr et al., 2020; Mauck et al., 2010; Mauck et al., 2014).

While there is overwhelming evidence that some viruses do induce plant phenotype
manipulation in ways that are conducive for their own transmission, many significant knowledge
gaps remain (discussed by Mauck and Chesnais, 2020). In particular, the mechanisms and pathways
by which viruses alter aspects of the host phenotype, and the virus components that are responsible,
remain poorly understood (Mauck et al., 2019).

In the present study, we addressed these shortcomings and initiated analysis of the effects of two
viruses, TuYV and CaMV, belonging to two different transmission categories, on transcriptomic
profiles in two susceptible host plant species (*Arabidopsis thaliana* and *Camelina sativa*, both
family *Brassicaceae*), and on changes in insect vector feeding behavior and performances. We
selected the green peach aphid *Myzus persicae* as vector because it transmits both TuYV and CaMV
and infests both plant hosts. We chose two different plant species as virus hosts, as previous studies
have highlighted potential host-specific effects of viruses on host plant traits and vector
performance (Chesnais et al., 2019b; Chesnais et al., 2021). In addition, their phylogenetic
proximity allows rather easy gene-to-gene comparisons. In fact, the *C. sativa* genome is highly
similar to the *A. thaliana* genome and might have arisen from hybridization of three diploid
ancestors of *A. thaliana* (Malik et al., 2018). For this reason, its genome is allohexaploid. Over 70
119 % of *C. sativa* genes are syntenically orthologous to *A. thaliana* genes (Kagale et al., 2014),
facilitating genomic studies of *C. sativa*.

**Material and methods**

***Aphids***

A Dutch green peach aphid clone (*Myzus persicae* Sulzer, 1776) was used for the experiments. It
was reared on Chinese cabbage (*Brassica rapa* L. *pekinensis* var. Granaat) in a growth chamber at
20±1 °C and a 16 h photoperiod. Only wingless forms were used in assays. For synchronization,
adults were placed on detached Chinese cabbage leaves that were laid on 1 % agarose in a Petri
dish. The adults were removed 24 h later and the newborn larvae used in experiments 5 days later
(for transcriptomic experiments) or 8 days later (for feeding behavior and performances
experiments).

***Viruses***

CaMV isolate Cm1841-Rev (Chesnais et al., 2021), which is a transmissible derivative of isolate
Cm1841 (Tsuge et al., 1994), and TuYV isolate TuYV-FL1 (Veidt et al., 1988) were maintained in
*A. thaliana* Col-0 and propagated by aphid inoculation of 2-week-old plants. Growth conditions
were as described below.

***Virus infection and aphid infestation***

Seeds of *Arabidopsis thaliana* Col-0 (hereafter *Arabidopsis*) or *Camelina sativa* var. Celine
(hereafter *Camelina*) were germinated in TS 3 fine substrate (Klasmann-Deilmann) in 7*7 cm pots
and watered with tap water. Growth conditions were 14 h day 10 h night with LED illumination and
a constant temperature of 21±1 °C. Two-week-old plants were inoculated with 3-5 wingless *Myzus*
*persicae* aphids that had been allowed a 24 h acquisition access period on *Arabidopsis* infected with
TuYV or CaMV or on healthy *Arabidopsis*. Plants were individually wrapped in clear plastic vented
bread bags to prevent cross contamination. Aphids were manually removed after a 48 h inoculation
period. Eighteen days post-inoculation (dpi), 25 to 30 5-day-old non-viruliferous aphids were
placed for infestation on the rosette (*Arabidopsis*) or the apical leaves (*Camelina*) of CaMV- or
TuYV-infected or mock-inoculated plants. After 72 h infestation (= 21 dpi), aphids were removed
with a brush. The infested plants (virus-infected or mock-inoculated) were washed 3 times with
deionized water and 3 times with MilliQ water to remove any remaining aphid exuvia and
honeydew. Then rosettes (*Arabidopsis*) or detached leaves (*Camelina*) were collected in 50 ml
Falcon tubes. Three biological replicates were used for analysis. For *Arabidopsis*, one biological

replicate consisted of 4 plants, for Camelina one replicate was 3 plants. Plant samples were
conserved at -80 °C until processing.

**RNA purification and Illumina sequencing**

Total RNA was extracted from one gram of Arabidopsis (rosettes) and Camelina (leaves) frozen

[revised manuscript text omitted]

257 project.org/](http://www.r-project.org/)).
258

Figure 1: Phenotype of CaMV and TuYV-infected plants. a) Arabidopsis and b) Camelina plants at 21 days after inoculation with the indicated virus or after mock inoculation.

Plant phenotype

We used in this study 5-week-old Arabidopsis or Camelina plants that had been inoculated with
CaMV or TuYV three weeks before analysis. In both Arabidopsis and Camelina plants, CaMV
caused severe leaf curling, mosaic and vein chlorosis as well as dwarfism (Figure 1). TuYV-
infected Arabidopsis and Camelina plants were smaller compared to mock-inoculated plants, but
showed no leaf deformation or bleaching. Older TuYV-infected Arabidopsis leaves turned purple,
probably due to stress-induced anthocyanin accumulation as previously reported for infection of
Arabidopsis with another polerovirus, brassica yellows virus (Chen et al., 2018). The purple
coloring was first visible on the abaxial leaf surface and progressed slowly until covering the entire
leaf very late in infection. Old leaves of TuYV-infected Camelina displayed mild yellowing
symptoms, primarily on the leaf border.

Figure 2: Aphid feeding behavior parameters recorded by EPG on 5-week-old mock-inoculated, TuYV- or CaMV-
infected a) Arabidopsis and b) Camelina. Different letters indicate significant differences between plants as tested by
GLM followed by pairwise comparisons using “emmeans” ($p < 0.05$; method: Tukey, $n = 20-23$).

Figure 3: Aphid fecundity 5 days after deposit (one aphid per plant) on 5-week-old mock-inoculated, TuYV- or CaMV-
 infected a) Arabidopsis and b) Camelina. Different letters indicate significant differences between plants as tested by
 GLM followed by pairwise comparisons using “emmeans” ($p < 0.05$; method: Tukey, $n = 27-33$).

**Aphid feeding behavior and fecundity**

We used EPG to compare aphid probing and feeding behavior on Arabidopsis and Camelina
 infected or not with CaMV or TuYV (Figure 2a,b). The total probing time was similar for all six
 conditions and the aphids were active for approximately seven hours during the eight hour
 observation period. The pathway phase and the time until the first phloem ingestion were in general
 longer on Camelina for all three conditions, whereas the ingestion phase was longer on Arabidopsis.
 The most important difference was the salivation time, which was extended on Camelina (50-100 %
 longer than on Arabidopsis), independently of the infection status.

CaMV infection changed aphid behavior similarly on both plant hosts. The pathway phase and
 the time to first phloem ingestion were decreased, whereas phloem ingestion was increased.
 Salivation time was not affected by the infection status of the two-plant species. Infection with
 TuYV had no major effect. The only behavioral parameter significantly affected by TuYV was the
 time until first phloem ingestion, which was reduced by half on TuYV-infected Arabidopsis but not
 on TuYV-infected Camelina, compared to mock-inoculated plants. This is in contrast with CaMV
 infection for which the time to first phloem sap ingestion was reduced on both hosts. Previous EPG
 experiments on Arabidopsis (Bogaert et al., 2020) and Camelina (Chesnais et al., 2019a) have
 reported neutral to slightly positive effects of TuYV infection on aphid probing and feeding
 behavior, and highlighted also host-specific viral effects on plant quality and vector behavior
 (Chesnais et al., 2019b).

Taken together, the significantly reduced time until first phloem ingestion observed on infected
 Arabidopsis might contribute to a better acquisition of CaMV and TuYV. The other transmission-
 related feeding parameters were only marginally modified on TuYV-infected plants, whereas
 CaMV infection altered aphid feeding more strongly. The reduced pathway phase and the increased
 phloem ingestion might also facilitate CaMV acquisition from phloem tissues. These alterations are
 expected to be detrimental for non-circulative viruses (such as the non-persistent potyviruses) that
 are acquired during intracellular penetrations occurring in the pathway phase, but lost if the aphid
 stylets reach the phloem sap (Kloth and Kormelink, 2020). However, this does not apply to CaMV,
 acquired efficiently from phloem sap as well as mesophyll and epidermis cells (Palacios et al.,
 2002).

Infection with CaMV reduced aphid fecundity significantly on both plant host species (Figure
 3a,b), compared to mock-inoculated plants (GLM, $\chi^2 = 0.0007$ and $\chi^2 = 0.0409$ for Arabidopsis and
 Camelina, respectively) and correlated with the strong symptoms of infected plants. Fecundity was
 unchanged on TuYV-infected Arabidopsis and Camelina. This indicated that the severe (but less
 strong, compared to CaMV infection) phenotype of TuYV-infected Camelina did not interfere with
 aphid fecundity.

**Quality of RNA and sequence alignment data**

Roughly 29-35 million reads were obtained for Arabidopsis mRNA-seq datasets, of which >80 %
could be aligned for mock-inoculated and TuYV-infected samples and 80% for CaMV-infected
samples (Supplementary Table S2). For Camelina, 28-38 million reads were obtained and 61-71 %
of the reads could be aligned (Supplementary Table S2). Principal component analysis of RNA-seq
libraries (Figure 4a,b) for both plant species indicated that the three biological replicates per
condition clustered well together and that the different conditions (mock-inoculated or infected with
either virus) were well separated. Thus, the reads were of excellent quality and suited for a
transcriptome analysis.

For 8 selected Arabidopsis genes, the trends of gene deregulations detected in the transcriptome
data could be reproduced by an alternative analysis method, RT-qPCR (Supplementary Figure S1
RT-qPCR). All 8 genes followed the same trend using either method for CaMV-infected
Arabidopsis, and all except two (At_AOS and At_EDS5) for TuYV-infected Arabidopsis. The
discrepancies were probably due to the rather weak expression changes, which are sometimes
difficult to detect by RT-qPCR due to its intrinsic exponential amplification kinetics.

Quantification of viral RNA loads by counting viral reads normalized per million of total plant
reads in each sample revealed that CaMV pregenomic (pg)RNA and TuYV genomic (g)RNA (both
represented by forward reads; Supplementary Dataset S1) accumulated to comparable levels in each
of the three biological replicates, with the exception of one of the three TuYV-Arabidopsis
replicates which showed a lower number of normalized viral reads. The data confirmed further that
the mock-inoculated plants were not cross-contaminated. Note that, because TuYV gRNA is not
polyadenylated (unlike CaMV pgRNA) the poly(A) enrichment step of Illumina library preparation
protocol should have led to its depletion. This might explain the greater variation in relative
abundance of TuYV reads between biological replicates, compared to CaMV reads. Despite this
high variability, a lower virus load was observed in TuYV-infected Arabidopsis compared to
TuYV-infected Camelina plants (Supplementary Dataset S1). Notably, CaMV loads in Arabidopsis
were also lower than those in Camelina (ca. 1.5 times; Supplementary Dataset S1). This indicates
that despite drastic differences in disease symptoms between CaMV (severe symptoms) and TuYV
(mild symptoms) in both Arabidopsis and Camelina, Camelina appears to be more conducive for
replication of both viruses than Arabidopsis.

Figure 4: Principal component (PC) analysis of the transcriptome data sets on a) Arabidopsis and b) Camelina. The three dots of the same color correspond to the three biological replicates. (c and d) Venn diagrams presenting the number of differentially expressed genes (DEGs) in TuYV and CaMV-infected c) Arabidopsis and d) Camelina. Magenta arrows: number of up-regulated genes, cyan arrows: number of down-regulated genes and two-color circles: inversely regulated genes (up-regulated genes in one virus-infected modality and down-regulated in the other virus-infected modality). e) Comparison of the number of DEGs and enriched GO terms in TuYV and CaMV-infected Arabidopsis and Camelina plants.

CaMV modifies expression of far more genes than TuYV

We determined the number of differentially expressed genes (DEGs) in infected hosts (Figure 4c-e). Far more DEGs were detected in Camelina than in Arabidopsis. This was in part due to its allohexaploid genome consisting of three Arabidopsis-like genomes coding for almost 90,000 genes (Kagale et al., 2014), compared to Arabidopsis's diploid genome containing about 28,000 coding genes (http://ensembl.gramene.org/Arabidopsis_thaliana/Info/Annotation/#assembly, last accessed 17 December 2021). This means that for many Arabidopsis genes there are three orthologous Camelina genes. Also, the higher accumulation of both viruses in this host might contribute to the higher counts.

In Arabidopsis, CaMV modified expression of ~11,800 genes significantly ($P_{(adj)} < 0.05$), whereas TuYV modified expression of ~1,300 genes, corresponding to 43 % and 5 % of the total genes, respectively (Figure 4e). In CaMV-infected Camelina, we detected ~36,700 DEGs, and in TuYV-infected Camelina ~9,400 DEGs, corresponding to 41 % and 11 % of all genes, respectively. Thus, the impact of CaMV infection on gene deregulation was much more pronounced when compared to TuYV infection, in accordance with the phenotype of infected plants (Figure 1). The lower numbers of DEGs for TuYV in both hosts could be at least partially due to its restriction to phloem tissues, unlike CaMV which infects all cell types.

CaMV modified expression of ~40 % of the total genes independently of the host plant, whereas the proportion of TuYV-induced DEGs was host-dependent and two times higher in infected Camelina compared to Arabidopsis (11 % vs. 5 %). This is in line with the relative loads of viral

RNA (Supplementary Dataset S1), indicating that Camelina is more susceptible to TuYV infection
 than Arabidopsis (3 times more TuYV RNA accumulation in Camelina compared to Arabidopsis),
 while CaMV accumulates in both hosts at comparable levels (only 1.5-fold difference in average
 viral RNA loads between Arabidopsis and Camelina).

956 DEGs, corresponding to 3.4 % of the genome, were common for both viruses in
 Arabidopsis. The proportion of common DEGs rose to 7.5 % (~6,700 genes) in infected Camelina.
 Since CaMV and TuYV are viruses with entirely different replication mechanisms, mediated
 respectively by viral reverse transcriptase and viral RNA-dependent RNA polymerase, these
 common host genes might be involved in general stress responses and/or are constituents of the core
 defense mechanisms. A GO analysis indicated that this was true for Arabidopsis with GO terms
 related to stress and transport in common for both infections, whereas for Camelina, ribosome and
 replication-related genes were enriched (Supplementary Figure S2).

The proportions of up- and down-regulated genes were similar for a given virus in the two hosts
 (Figure 4e). However, when comparing the two viruses, the proportion of down-regulated genes
 was higher in CaMV-infected plants (about 50 % in CaMV-infected Arabidopsis and Camelina)
 than in TuYV-infected plants (34 % in TuYV-infected Arabidopsis 39 % in TuYV-infected
 Camelina). Thus, there was a correlation between the proportion of down-regulated genes and
 symptom severity. The milder disease symptoms of TuYV infection coincided in both hosts with
 the lower proportion of down-regulated genes, while the ability of CaMV to downregulate a higher
 proportion of genes coincided with the more severe disease symptoms. The latter ability might
 reflect CaMV activities in both cytoplasm (viral mRNA translation and pgRNA reverse
 transcription) and nucleus [viral dsDNA repair followed by pgRNA transcription and export
 assisted by nuclear-imported viral proteins P4, P5 and likely P6 (Haas et al., 2008; Kubina et al.,
 2021)].

**Impact of CaMV and TuYV infection on plant hosts: Gene ontology analysis**

 Figure 5: Gene ontology (GO) analysis showing the Top 25 GO of deregulated processes in TuYV- and CaMV-infected
 Arabidopsis and Camelina. a) TuYV-infected vs. mock-inoculated Arabidopsis, b) CaMV-infected vs. mock-inoculated

Arabidopsis, c) TuYV-infected vs. mock-inoculated Camelina and d) CaMV-infected vs. mock-inoculated Camelina.
GO IDs and corresponding GO terms are specified in the vertical axis. For each category (BP: Biological Process, CC:
Cellular Component and MF: Molecular Function), GOs are sorted according to increasing log₂ p-values, also indicated
by the color of each spot (magenta representing the most significant p-values, see color scale bar), in order to place the
most significantly enriched GOs on top of the graph. The absolute number of DEGs that matched the GO term is
indicated by the size of each spot, whereas the horizontal axis shows the ratio of DEGs vs. all genes belonging to the
GO term.

To identify the most prominent processes affected in aphid-infested CaMV and TuYV-infected
Arabidopsis and Camelina, we carried out a Gene Ontology (GO) analysis (Figure 5). In general,
TuYV-induced GO changes were much less pronounced (considering the percentage of DEGs in
each category and the DEG counts) compared to CaMV, reflecting the low absolute numbers of
DEGs in TuYV-infected plants and the weaker impact of TuYV on plant phenotype. Remarkably,
in the Top 25 categories, only about 25 % of genes per GO were deregulated in TuYV-infected
Arabidopsis (Figure 5a), whereas this value increased to more than 50 % in TuYV-infected
Camelina (Figure 5c). Again, this may indicate that TuYV infection had a stronger effect on gene
regulation in Camelina than in Arabidopsis. A different situation was found for CaMV, where the
percentages of DEGs per GO were similar in both hosts, and always above 50 % of genes per GO,
indicating similarly strong interactions with either host plant.

Then we looked closer at the different categories. Interestingly, in CaMV-infected Arabidopsis
(Figure 5b), most of the enriched GO-terms were related to photosynthesis/chloroplast in both BP
and CC categories, which might explain leaf chlorosis (loss of chlorophyll and/or chloroplasts). The
next most affected biological process was oxidation-reduction that might be related to stress
response. Also, GO-terms related to microtubule-based movement appeared in the BP and CC lists,
as well as apoplast, cell wall, kinesin complex and nucleolus, which may be linked to virus or viral
RNP intracellular trafficking. Taken together, CaMV infection mostly modified photosynthesis,
oxidation-reduction processes and microtubule-related processes.

GO analysis of CaMV-infected Camelina indicated a similar pattern (Figure 5d). Again, several
GO-terms related to photosynthesis/chloroplast and oxidation/reduction were enriched in both BP
and CC categories. It is worth mentioning that for CaMV infection of Camelina, GO analysis
showed enrichment of genes in the GO Defense Response (BP – GO:0006952), which was absent
in the Top 25 list of CaMV-infected Arabidopsis. Other BP-related enriched GOs were protein
phosphorylation and carbohydrate metabolism. As in Arabidopsis, apoplast changes were
significant. On the other side, neither cell wall nor microtubule processes were present among the
Top 25 deregulated processes in Camelina. Concerning the main categories of molecular functions,
oxidation-reduction and protein domain specific binding dominated this category.

In contrast to CaMV, TuYV infection of Arabidopsis had no impact on photosynthesis-related
GO terms. In both BP and MF, the most significantly enriched GO-terms were found in transport,
especially carbohydrate transport. In addition, some defense and stress responses (xenobiotics,
chitin, salt) and glutathione metabolism – indicative of oxidative stress – were affected. Flavonoid
synthesis was significantly deregulated, in line with the purple-colored leaves of TuYV-infected
Arabidopsis (Figure 1). In accordance with the modifications in sucrose transport, the most
prominent category in CC comprised membranes. The Top 25 GO-terms in TuYV-infected
Camelina were different from those in TuYV-infected Arabidopsis. As in CaMV infection, DEGs in
photosynthesis and related processes dominated the Top 25 GO in TuYV-infected Camelina in BP
and CC and were likely related to the mild yellowing symptoms appearing on old leaves. Next were
DNA-related processes in both BP and MF categories, probably linked to transcriptional regulations
of host genes in response to viral infection. In CC, the GO-term nucleus was deregulated, again in
favor of a strong effect of TuYV on transcriptional regulation in infected Camelina. Other
deregulated processes included ubiquitination, which appeared in several categories in BP and MF.
On the other hand, oxidation-reduction did not appear under the top 25 GO except as plastoquinone,
which represents a significant difference between CaMV and TuYV infections of Arabidopsis.

**Impact of CaMV and TuYV infection on plant hosts: Heatmap analyses**

To better characterize the impact of viral infection on aphid-infested plants, we established the lists
(Supplementary Dataset S2) and corresponding heatmaps (Figures 6-10) of DEGs for selected
categories. Note that if not otherwise indicated, information on gene function is from the TAIR site
(<https://www.arabidopsis.org/>). For mapping of Arabidopsis and Camelina genes in the heatmaps,
we used the syntelog matrix (Kagale et al., 2014). This matrix assorts each individual Arabidopsis
gene to the corresponding triplet of Camelina homologues. 62,277 Camelina genes out of 89,418
are syntenically orthologues (syntelogs) to Arabidopsis genes, among which some are considered
‘fractionated’ (if one or two of the homologues were lost). This explains why for certain
Arabidopsis genes, only one or two homologous Camelina genes are presented.

Figure 6. Hierarchical clustering of differentially expressed genes (DEGs) related to photosynthesis (GO:0015979) in
CaMV- and TuYV-infected Arabidopsis and Camelina compared to their mock-inoculated control plants
(Supplementary Dataset S2). The color key scale displays the log₂fold changes from -10 to +10 as a gradient from cyan
to magenta.

CaMV and TuYV infection downregulated photosynthesis-related genes in infested Arabidopsis
and Camelina (Figure 6). Overall downregulation of photosynthesis-related genes was much more
pronounced in CaMV-infected than in TuYV-infected plants. Interestingly, both viruses interacted
more strongly with Camelina photosynthesis than with Arabidopsis photosynthesis. No
photosynthesis-related gene was similarly deregulated in all four conditions. The most
downregulated photosynthesis gene in CaMV-infected Camelina was *PORA* (Camelina
Csa02g051950, Csa11g086170 and Csa18g025480, corresponding to Arabidopsis AT5G54190),
coding for a protein involved in chlorophyll biosynthesis, but also in response to ethylene. *PORA*
expression was also inhibited in TuYV-infected Camelina. Expression of the Arabidopsis
orthologue, however, was not modified by any of the two viruses. This might indicate that
downregulation of *PORA* is a plant-specific and not a virus-specific response. The most
downregulated gene in CaMV-infected Arabidopsis, AT3G27690, encoding the protein LHCb2.4, a
component of the light harvesting complex, was also strongly repressed in Camelina infected by
CaMV. Expression of this gene was also affected in TuYV-infected Arabidopsis but to a lesser
extent and expression was not modified in TuYV-infected Camelina. Some genes were upregulated
by infection. This was notably the case for the glucose-6-phosphate/phosphate transporter 2
(AT1G61800), which is involved in regulation of photosynthesis and which was upregulated in
three of the four conditions (no gene expression modification in CaMV-infected Camelina). In

CaMV-infected Camelina, a chloroplastic ferritin (AT3G11050) was the most upregulated
 photosynthesis gene. Ferritins are iron-binding proteins and are supposed to be involved in
 responses against oxidative stress in Arabidopsis (Briat et al., 2010), which could explain its
 overexpression.

Taken together, virus infection strongly interfered with photosynthesis. This might explain the
 leaf yellowing observed clearly on CaMV-infected plants and to a lesser extent on TuYV-infected
 Camelina. Leaf yellowing, probably due to reduced chlorophyll content in chloroplasts and/or to a
 reduced number of photosynthesizing chloroplasts caused by the gene deregulations (Chen et al.,
 2002), can alter settling preference of aphids (A'Brook, 1973; Fennell et al., 2020). However, this
 was not confirmed by previous observations on TuYV-infected and CaMV infected Camelina plants
 (Chesnais et al., 2019a). Indeed, although Chesnais and coworkers reported that *M. persicae* aphids
 preferred to settle on TuYV-infected Camelina, compared to healthy plants, no such aphid
 preference was observed for CaMV-infected Camelina, despite the strong yellowing symptoms.
 This suggests that aphid preference for a plant is not only driven by visual aspects.

 Figure 7. Hierarchical clustering of differentially expressed genes (DEGs) related to a) gluconeogenesis (GO:0006094),
 b) sucrose biosynthetic process (GO:0005986), c) starch biosynthetic process (GO:0019252) and d) trehalose
 biosynthetic process (GO:0005992) in CaMV- and TuYV-infected Arabidopsis and Camelina compared to mock-
 inoculated controls (Supplementary Dataset S2). The color key scales display the log2fold changes as color gradients
 from cyan to magenta.

In line with the repression of photosynthesis, expression of many sucrose synthesis and
 gluconeogenesis-related genes was reduced by CaMV infection (Figure 7a,b). The effect of CaMV

was stronger in Camelina than in Arabidopsis. In TuYV-infected Camelina, the amplitude of the
 gene deregulation was smaller, compared to CaMV-infected Camelina, but the proportions of up-
 and down-regulated genes were comparable. In TuYV-infected Arabidopsis, expression changes
 were even smaller than in TuYV-infected Camelina. For both TuYV- and CaMV-infected plants,
 among the most down-regulated genes were those coding for key enzymes in sucrose synthesis, in
 particular *HCEF1* (AT3G54050) and *FBP* (AT1G43670). Interestingly, the sucrose phosphatase
 *SPP1* (AT1G51420) was strongly upregulated by CaMV infection, but downregulated in TuYV-
 infected plants. The most down-regulated gene in gluconeogenesis was the aldolase *FBA5*
 (AT4G26530) and this in all four conditions tested. In line with the stronger suppression of
 gluconeogenesis and sucrose synthesis-related genes by CaMV infection, many starch synthesis-
 related genes were also repressed by CaMV (but not TuYV) infection (Figure 7c). An exception
 was *DBE1* (Csa17g005380, Csa14g004380 and Csa03g004400, syntelogs of AT1G03310)
 encoding a starch branching enzyme was upregulated in TuYV-infected Camelina, which is
 consistent with a recent study showing that TuYV-infection tends to increase the carbohydrate
 concentrations in Camelina leaves (Chesnais et al., 2019b).

The effect of infection and infestation on trehalose metabolism was different from that on
 glucose, starch and sucrose metabolism (Figure 7d). Contrary to downregulation of the latter
 carbohydrate pathways only in CaMV-infected plants, trehalose-related genes were downregulated
 in both TuYV- and CaMV-infected Arabidopsis and Camelina. Upregulated genes in this pathway
 were also observed, in particular in CaMV-infected Arabidopsis. Trehalose is induced by *M.*
 *persicae* infestation and has been shown to contribute to defenses against aphids (Hodge et al.,
 2013; Singh et al., 2011; Singh and Shah, 2012). Especially *TPS11* (Trehalose Phosphate Synthase
 11) has been implicated in mounting defenses against aphids (Singh et al., 2011), by promoting
 starch synthesis, but also by activating the phytoalexin-deficient gene, *PAD4*. Starch is a feeding-
 deterrent to aphids (Campbell et al., 1986) and elevated starch levels are correlated with reduced
 aphid performance (Singh et al., 2011). Interestingly, *TPS11* but also *TPS8* were significantly
 down-regulated in all virus-infected plants, suggesting that viral infection might favor aphid
 infestation. On the other hand, other TPS isoforms were not modified in the same way, for example
 the *TPS5* (AT4G17770) was upregulated in all four conditions tested. This suggests a more
 complex regulation of this pathway by viral infection and aphid infestation.

We also analyzed expression of genes involved in amino acid metabolism, because they represent
 the most important nutrient for aphids. Probably due to the multiple functions of these genes, no
 clear pattern was detected (Supplementary Figure S3).

 Figure 8. Hierarchical clustering of differentially expressed genes (DEGs) related to a) production of siRNA involved in
 RNA interference and Gene silencing by RNA (GO:0030422 and GO:0031047), b) defense response to virus

(GO:0051607) and c) salicylic acid biosynthetic process (GO:0009697) in CaMV- and TuYV-infected Arabidopsis and
Camelina compared to their mock-inoculated controls (Supplementary Dataset S2). The color key scales display the
log₂fold changes as gradients from cyan to magenta.

When looking at global virus defense-related genes, the effects on their regulation were more
virus-specific than host-specific. In agreement with previous findings in CaMV-infected
Arabidopsis (Shivaprasad et al., 2008), many RNA silencing-related genes were found to be
upregulated by CaMV not only in Arabidopsis but also in Camelina (Figure 8a). Among them, most
notable are components of the 21 nt siRNA-directed gene silencing pathways such as Double-
stranded RNA-binding protein 4 (DRB4), a partner of the antiviral Dicer-like protein 4 (DCL4)
generating 21 nt siRNAs, and siRNA-binding effector proteins Argonaute 1 (AGO1; AT1G48410),
AGO2 (AT1G31280) and AGO3 (AT1G31290). Notably, AGO2, also known to be involved in
defense against RNA viruses (Carbonell and Carrington, 2015), was up-regulated in TuYV-infected
Camelina (but not in TuYV-infected Arabidopsis), while AGO3 – the Argonaute protein most
closely related to AGO2 and also showing antiviral activity *in vitro* (Schuck et al., 2013) – was up-
regulated in response to TuYV infection in both hosts. This suggests redundant (and compensatory)
roles of AGO2 and AGO3 in defenses against both RNA and DNA viruses. DCL4 itself
(AT5G20320) and DCL2 (AT3G03300), generating 22 nt siRNAs and acting together with DCL4
in defenses against RNA and DNA viruses (Blevins et al., 2006), were respectively weakly (one of
three isoforms of Camelina DCL4) and strongly upregulated (DCL2) in CaMV-infected Camelina
but not in the other virus-host combinations where their levels were likely sufficiently high to cope
with both viruses. Interestingly, AGO10 that counteracts AGO1 in the miRNA-directed silencing
pathways regulating plant development and physiology (Yu et al., 2017) was downregulated by
CaMV (but not TuYV) infection in both Arabidopsis and Camelina. RNA-directed RNA
polymerase 6 (RDR6) generating miRNA-dependent secondary 21-nt siRNAs was upregulated by
TuYV infection in Camelina and down-regulated by CaMV infection in Arabidopsis, while
remaining non-responsive in the other virus-host combinations. Components of the nuclear
silencing and 24 nt siRNA-directed DNA methylation pathways such as AGO4 (AT2G27040),
AGO6 (AT2G32940) and AGO9 (AT5G21150) were upregulated in Camelina by CaMV (but not
TuYV), while AGO4 and AGO6 were down-regulated and AGO9 up-regulated in CaMV-infected
Arabidopsis, denoting virus- and plant-specific gene deregulations. Interestingly, the most
upregulated gene in the RNA silencing category was AT5G59390. It was strongly upregulated in
CaMV-infected Camelina and Arabidopsis, less strongly upregulated in TuYV-infected Camelina
and not significantly deregulated in TuYV-infected Arabidopsis. This gene codes for a XH/XS
domain-containing protein, which probably functions in siRNA-directed DNA methylation and
might contribute to methylation and transcriptional silencing of CaMV dsDNA in the nucleus
(Omae et al., 2020). Taken together, CaMV infection strongly affected silencing-related genes in
both hosts, but the deregulations were host-specific, with down-regulations dominating in
Arabidopsis, and upregulations in Camelina. Transcription of RNA silencing genes was much less
affected in TuYV-infected plants.

Among components of other defense pathways (Figure 8b), the hairpin-induced protein *hin1*
(AT2G35980, also referred to as YLS9 and reported to be induced by cucumber mosaic virus
infection) was strongly induced during CaMV infection, while *Rem 1.2* (AT3G61260, also referred
to as REMORIN and known to negatively regulate cell-to-cell movement of the potyvirus TuMV
via competition with PCaP1 for binding actin filaments (Cheng et al., 2020) was strongly down-
regulated during CaMV infection. None of these genes (*hin1* and *Rem1.2*) were deregulated by
TuYV-infection. However, another REMORIN, *Rem1.3* (known to impair potato virus X
movement (Raffaële et al., 2009)) was down-regulated during TuYV infection in Camelina and
during CaMV infection in both Arabidopsis and Camelina. On the other hand, the gene for myo-
inositol-phosphate synthase 2 (*MIPS2*, AT2G22240) was downregulated in all conditions and
*RTM3* [AT3G58350, known to block phloem movement of potyviruses, (Cosson et al., 2010)] was
upregulated in TuYV- and downregulated in CaMV-infected hosts. It is therefore conceivable that

remorins and MIPS2 are factors controlling TuYV and CaMV movement. Curiously, the gene *NIK1*
(AT5G16000, NSP-interacting kinase), which encodes a receptor-like kinase, involved in innate
immunity-based defense response against a ssDNA geminivirus, was strongly down-regulated in
CaMV-infected but not in TuYV-infected host plants. Considering that downregulation of *NIK1*
should activate protein translation and could promote accumulation of viral proteins (Zorzatto et al.,
2015), it could have a proviral effect during CaMV infection.

Next, we looked at salicylic acid (SA) synthesis as this phytohormone is related to innate
immunity-based defense responses against non-viral and viral pathogens including CaMV (Zvereva
et al., 2016; Zvereva and Pooggin, 2012). Here, most genes were induced in both hosts and for both
viruses (Figure 8c), with the notable exception of *ICS2* (AT1G18870), which was downregulated in
CaMV infection and slightly upregulated in TuYV infection, while its redundant orthologue *ICS1*
(AT1G74710) was slightly but significantly downregulated only in CaMV-infected Arabidopsis.
Overall, genes involved in virus defense and SA biosynthesis were more strongly induced by
CaMV-infection than TuYV-infection, whatever the host-plant, indicating a stronger pathogenicity
of CaMV. The overexpression of SA-related genes in CaMV-infected plants could also reflect the
ability of CaMV effector protein P6 to suppress SA-dependent autophagy, which might lead to
compensatory feedback up-regulation of SA genes (Zvereva et al., 2016). Deregulation of genes
implicated in the SA pathway might have consequences on insect-plant interactions. In particular
increased SA could have a beneficial effect on aphid vector and possibly transmission, because it
can be concomitant with decreased JA levels and consequently with decreased plant defenses
against aphids (Kloth et al., 2016; Lu et al., 2020).

 Figure 9. Hierarchical clustering of differentially expressed genes (DEGs) related to a) jasmonic acid biosynthetic
 process (GO:0009695), b) ethylene biosynthetic process (GO:0009693), c) glucosinolate biosynthetic process
 (GO:0019761) and d) camalexin biosynthetic process (GO:0010120) in CaMV- and TuYV-infected Arabidopsis and
 Camelina compared to mock-inoculated controls (Supplementary Dataset S2). The color key scales display the log₂fold
 changes as gradients from cyan to magenta.

Next, we analyzed different metabolic pathways to determine if CaMV and TuYV infections
 modulate other hormones and secondary metabolites in ways that are more favorable for their aphid
 vector, and hence, for their own transmission.

We first looked at jasmonic acid (JA) and its derivatives because they are plant signaling
 molecules related to plant defense against herbivorous insects, microbial pathogens and different
 abiotic stresses (Figure 9a). We observed a strong virus-specific and host-independent effect for JA
 synthesis genes. CaMV downregulated many genes in the JA pathway, while TuYV upregulated
 some. Like for other pathways, the effect was stronger in infected Camelina than in Arabidopsis.
 Deregulated genes were for example *AOC1/3/4* (3 out of for 4 chloroplastic allene oxide cyclases,
 involved in JA synthesis), *AOS* (AT5G42650, chloroplastic allene oxide synthase, involved in JA
 synthesis) and *LOX2* (AT3G45140, chloroplastic lipoxygenase required for wound-induced JA
 accumulation in Arabidopsis). All these genes were slightly upregulated in TuYV, and strongly
 repressed in CaMV-infected plants. This might imply that JA production is down in CaMV-infected
 plants and stable or slightly induced in TuYV-infected plants. JA is generally thought to decrease
 aphid growth and fecundity, so aphids on CaMV-infected plants might have greater fecundity.
 However, infection of Arabidopsis with CaMV lowered fecundity (Figure 3a). JA-mediated

signaling pathways are also known to increase proteins and secondary metabolites, which act as
feeding deterrents (Howe and Jander, 2008). In this context, decreased JA production in CaMV-
infected Arabidopsis could encourage longer/faster phloem sap ingestion, which we observed
indeed in our experiments (Figure 2a). Interestingly, phloem sap ingestion has been correlated with
increased CaMV acquisition (Palacios et al., 2002), which makes JA pathway a major candidate for
virus manipulation.

We also analyzed ethylene (ET) synthesis (Figure 9b) as several studies have identified ET as a
plant response against aphid infestation (Anstead et al., 2010; Mewis et al., 2005). No specific gene
expression patterns characteristic for a virus or a host were found, indicating that the ethylene
response was unique for each virus-host pair. Notably, the ACC oxidase genes *ACO2* and *ACO5*
(AT1G62380 and AT1G77330), involved in ethylene production, were strongly down-regulated for
CaMV-infected host-plants. This is consistent with reduced accumulation of ethylene in CaMV-
infected and P6-transgenic Arabidopsis in response to bacterial elicitors of innate immunity
(Zvereva et al., 2016).

Glucosinolates (GLSs) are secondary metabolites that are produced by plants in the *Brassicaceae*
family and set free in response to herbivore attacks (Kim et al., 2008). Some GLSs have been
shown to be strong feeding deterrents for generalist aphids such as *M. persicae* (Kim and Jander,
2007) and might even have antibiosis effects on this aphid species (Cole, 1997; Westwood et al.,
2013). CaMV infection down-regulated genes involved in GLS synthesis (Figure 9c), for example,
the three Camelina orthologues encoding the cytochrome P450 monooxygenase CYP83A1
(AT4G13770), whereas these three genes were upregulated in TuYV-infected Camelina. The effect
of CaMV infection on CYP83A1 was less pronounced in Arabidopsis, where another gene,
AT1G65880, involved in benzoyloxyglucosinolate 1 synthesis, was strongly repressed. Other genes
implicated in GLS synthesis, for example *IMDI* (AT5G14200), an isopropylmalate dehydrogenase
(He et al., 2011) were upregulated in TuYV-infected Camelina. The transcription factor *MYB51*
(AT1G18570), involved in indole glucosinolate synthesis (Barco and Clay, 2020), was
downregulated in CaMV-infected Camelina, but not in the other conditions. All in all, infection
with CaMV predominantly down-regulated transcription of GLS-related genes in Camelina and to a
lesser extent in Arabidopsis, whereas TuYV infection induced GLS synthesis in aphid-infested
Camelina, and had hardly any effect on Arabidopsis. We therefore expected that *M. persicae* fitness
and feeding behavior (i.e. phloem sap ingestion and ease to access phloem tissues) would be
enhanced on CaMV-infected Camelina and Arabidopsis, and be decreased on TuYV-infected
Camelina. However, our fecundity experiments show that, on the contrary, *M. persicae* fecundity
was decreased on CaMV-infected Arabidopsis and remained unchanged on TuYV-infected
Arabidopsis compared to mock-inoculated plants, while no effects were observed on Camelina
plants. Previous experiments indicated that *M. persicae* fecundity is even higher in TuYV-infected
Camelina and lower in CaMV-infected Camelina (Chesnais et al., 2019a). Note, however, that in
the experiment of Chesnais et al. (2019a), a more severe CaMV strain was used, which might
explain the discrepancies between both experiments. Overall, based on our results, deregulation of
GLS-related genes after CaMV and TuYV infections do not seem to be the main factors controlling
*M. persicae* fecundity. This is in line with another study that found no correlation between the GLS
content of rapeseed and *M. persicae* fecundity (Weber et al., 1986).

On the other hand, our EPG experiments showed that aphids were able to reach phloem tissues
and ingest phloem sap for a longer duration on CaMV-infected Arabidopsis and Camelina (see also
Chesnais et al., 2019a; Chesnais et al., 2021). Therefore, down-regulation of GLS genes might
encourage aphid settling/feeding behavior on CaMV-infected plants, and eventually promote CaMV
acquisition by *M. persicae*. On TuYV-infected plants, while some GLS-related genes were slightly
up-regulated, aphid feeding behavior was roughly equivalent to that on healthy plants, indicating
that up-regulations were not strong enough to induce feeding deterrence.

Camalexin is the major phytoalexin and has been shown to reduce fecundity of aphids in
Arabidopsis (Kettles et al., 2013), although its effect on aphids might not be straight-forward (Kloth

et al., 2019; Pegadaraju et al., 2005). *PAD3* (phytoalexin deficient 3) catalyzes the last step in its
 synthesis and *CYP79B2* an important intermediate step. Here we found contrasting effects of aphid
 infestation on virus-infected plants on camalexin-related genes (Figure 9d). *PAD3* was
 downregulated in CaMV-infected and to a lesser extent in TuYV-infected *Camelina*, and slightly
 upregulated in *Arabidopsis* infected with CaMV or TuYV. *CYP79B2* expression was unaffected in
 both *Camelina* and *Arabidopsis* infected with TuYV, but substantially downregulated in CaMV-
 infected *Camelina* and slightly upregulated in CaMV-infected *Arabidopsis*. This indicates for both
 genes a strong host-plant effect. Evidence indicates that *PAD3* contributes more to camalexin
 synthesis than *CYP79B2* (Kim et al., 2015; Zang et al., 2008; Zhang et al., 2020). Thus, looking at
 *PAD3*, aphid fecundity should be higher on CaMV- and TuYV-infected *Camelina*, and lower on
 infected *Arabidopsis*. However, we observed a lower fecundity in CaMV-infected *Arabidopsis* and
 unchanged aphid fecundity in all other conditions, which suggests that aphid fecundity is not only
 linked to *PAD3* expression. On the other hand, phloem ingestion on both CaMV-infected
 *Arabidopsis* and *Camelina* increased which is more in accordance with aphid plant acceptance.
 Overall, camalexin-related gene deregulations observed in both infected host plants did not seem to
 correlate with modified aphid fecundity, nor with aphid feeding behavior.

Callose is a polymer that is deposited by plants in between cells and in sieve tubes to restrict
 access of pathogens, including aphids, to tissues and phloem (Kuśnierczyk et al., 2008). We did not
 find any major DEG for this category except the stress-related plasma membrane respiratory burst
 oxidase Rboh F (Suzuki et al., 2011) and the pectin methylesterase inhibitor AT5G64640. No clear
 pattern of gene deregulation was observed, making interpretation difficult (Supplementary Figure
 S2). This might also be due posttranslational modifications that majorly regulate Rboh F activity
 (Kadota et al., 2015).

Figure 10. Hierarchical clustering of differentially expressed genes (DEGs) related to a) phloem proteins (PP2 and PP1)
 and callose deposition in phloem sieve plates (GO:0080165) and b) defense response to insect and response to insect
 (GO:0002213 and GO:0009625) in CaMV- and TuYV-infected *Arabidopsis* and *Camelina* compared to mock-
 inoculated controls (Supplementary Dataset S2). The color key scales display the log₂fold changes as gradients from
 cyan to magenta.

 Since aphid lifestyle depends on compatible interactions with the phloem they feed on, we
 looked at phloem protein expression (Figure 10a). In all conditions except aphid-infested TuYV-

infected *Camelina*, *PP2-B2* was the strongest induced gene. *PP2-B2* codes for a phloem-specific
lectin-like protein with unknown function containing an F-box domain and a potential
myristoylation site (Boisson et al., 2003) that could control membrane localization. Specific for
CaMV infection, the putative phloem lectin genes *PP2-B1* (AT2G02230) and *PP2-B5*
(AT2G02300) were upregulated in *Arabidopsis* and one of their orthologues was upregulated in
*Camelina*. The putative calcium-, lipid- and RNA-binding phloem protein PP16-1 (AT3G55470)
was, independent of the virus, upregulated in infected *Arabidopsis* and downregulated in *Camelina*.
One *Camelina* orthologue of the *Arabidopsis* *PP2-A1*, known to repress aphid phloem feeding
(Zhang et al., 2011), was down-regulated in *Camelina* infected with both CaMV and TuYV, but in
*Arabidopsis* this gene remained non-responsive to viral infections. It is worth mentioning that PP2
proteins of cucurbits could potentially bind to viral particles of CABYV (genus *Polerovirus* like
TuYV) and increase virus stability in the aphid gut (Bencharki et al., 2010). Proteins of this type
could therefore have a double importance due to their role on vector aphids' feeding behavior and
their possible involvement in virus transmission. Deregulation of most other phloem proteins did
not follow a distinct pattern and the unknown functions of most of these genes precluded any
interpretation.

CalS7 (AT1G06490), a phloem-specific callose synthase responsible for wounding stress-
induced callose deposition onto sieve tube plates and hence phloem plugging (Xie et al., 2011), was
slightly upregulated in TuYV-infected *Camelina* and downregulated in CaMV-infected
*Arabidopsis*. The same trend (upregulation in TuYV-infected *Camelina* and *Arabidopsis*,
downregulation in CaMV-infected *Camelina*) applied to the phloem-located sucrose synthases
SUS5 and SUS6 (AT5G37180 and AT1G73370) that interact with CalS7 (Barratt et al., 2011). Also
*SLII* (AT3G10680), a gene coding for a phloem small heat shock-like protein known to be involved
in resistance to *M. persicae* and other phloem feeders (Kloth et al., 2021), was downregulated in
both *Arabidopsis* and *Camelina* infected by CaMV. This might indicate that CaMV infection but
not TuYV infection favors phloem feeding of aphids by perturbing stress-related callose deposition
on sieve plates. This is in line with the prolonged phloem ingestion observed for *M. persicae* on
CaMV-infected plants (Figure 2).

Next, we examined expression of genes known to be involved in plant responses and defenses
against insects (Figure 10b), as their modulation could influence virus-insect interactions and hence
transmission. General trends were suppression in CaMV-infected *Camelina* and activation in
CaMV-infected *Arabidopsis* and in TuYV-infected *Camelina* and *Arabidopsis*, resulting in both
host-specific and virus-specific responses. *ESMI* (AT3G14210) was strongly downregulated in both
CaMV-infected hosts, but not in TuYV-infected hosts. Its gene product biases production of
glucosinolates, and its knockout mutant is more susceptible to herbivory by the caterpillar
*Trichoplusia ni* (Zhang et al., 2006). Thus, its downregulation in CaMV-infected plants might favor
aphid colonization. Expression of ATWSCP (AT1G72290), a protease inhibitor and water-soluble
chlorophyll-binding protein, was strongly upregulated in TuYV-infected and downregulated in
CaMV-infected *Camelina*, whereas its expression was unchanged in *Arabidopsis*. The apoplastic
ATWSCP, together with the protease RD21, protects plants, especially greening plants, against
herbivory (Boex-Fontvieille et al., 2015). Whether it also acts against aphids is unknown. The *M.*
*persicae*-induced lipase 1 (MPL1, AT5G14180) was upregulated in TuYV-infected *Camelina* and
CaMV-infected *Arabidopsis*, but downregulated in TuYV-infected *Arabidopsis* and unaffected in
CaMV-infected *Camelina*. This gene is induced by aphid infestation and decreases aphid fecundity,
but its enhanced expression does not change aphid behavior or plant choice (Louis et al., 2010).
Whether the reduced fecundity of *M. persicae* on CaMV-infected *Arabidopsis* is partially due to the
action of this gene, remains an open question. Strong host plant-specific and virus-specific effects
were found for VSP2 [AT5G24770, reported to have a role in defense against herbivorous insects
(Liu et al., 2005)]. Expression was up-regulated in aphid-infested TuYV-infected *Camelina* and
down-regulated in aphid-infested CaMV-infected *Camelina* but not affected in infested
*Arabidopsis*. All in all, plant defense responses against insects did not follow a clear pattern. This

was probably due to the very divergent pathways and the heterogeneity of the plant insect response
genes.

**Concluding remarks**

In this work we analyzed the effect of CaMV and TuYV infection of *M. persicae* aphid-infested
Arabidopsis and Camelina on the plant hosts' transcriptomes as well as on the fecundity and
feeding behavior of their vector *M. persicae*.

Our results show that CaMV infection caused more severe effects on phenotype of both plant
species than did TuYV infection (Figure 1). The severity of symptoms correlated strongly with the
proportion of DEGs (41-43 % for CaMV, 5-11 % for TuYV, Figure 4e). CaMV infection affected
the same percentage of genes in both plant hosts, whereas TuYV infection deregulated
proportionally twice as many genes in Camelina than in Arabidopsis. Again, this correlated with
stronger visible symptoms on TuYV-infected Camelina in comparison with TuYV-infected
Arabidopsis. Aphid performance changes were more pronounced on CaMV-infected hosts,
whatever the plant species, compared to those caused by TuYV infection. In spite of more DEGs in
TuYV-infected Camelina than in TuYV-infected Arabidopsis, aphid behavior was slightly more
impacted on TuYV-infected Arabidopsis (Figure 2). This likely indicates modification of plant
metabolites that cannot be identified by transcriptome profiling. A metabolomic analysis of virus-
infected leaves or phloem sap should provide complementary data on the aphid-plant-virus
interactions.

In this study, we did not compare the contribution of aphid infestation alone on the plant
transcriptome. However, recent work (Annacondia et al., 2021) on the transcriptome changes of
healthy Arabidopsis plants infested or not with *M. persicae* for 72 h, identified a limited number of
DEGs (265) suggesting that the contribution of aphid infestation to the transcriptome in healthy and
probably also in virus-infected and infested plants could be minor.

The most pronounced effect of CaMV infection on plant hosts was a strong downregulation of
photosynthesis genes (Figure 6) and carbohydrate metabolism-related genes (Figure 7). We
observed significant changes in many other pathways, including categories that are likely affecting
virus-vector interactions (*i.e.* defenses, silencing, hormones, secondary metabolites etc.). However,
the impact of these modifications on aphid fitness or feeding behavior was not easy to evaluate
since these parameters are likely under the control of several, often overlapping metabolic
pathways. Trying to correlate the effect of specific genes on aphids as reported in the literature with
our aphid behavior observations therefore often resulted in contrasting results. We offer the
following explanations. The very strong alterations in photosynthesis might have drowned
otherwise visible effects of DEGs previously found to be involved in plant-aphid interactions.
Another explanation is regulation by posttranscriptional and posttranslational modifications. While
transcriptome profiling is a powerful tool, it can display only changes of transcript levels. In many
cases, however, posttranslational modifications of proteins (such as phosphorylation, localization,
complex formation and many more) and even posttranscriptional RNA modifications (sequestering
of RNAs in p-bodies and others) will contribute to phenotype changes. Depending on the pathway,
the contribution of the transcriptome and posttranscriptome on cellular processes and beyond will
vary. This again indicates that complementary analyses such as metabolomics, proteomics etc.
might help to gain a more complete insight.

Nevertheless, we observed that virus infections have very distinct effects on the transcriptome of
host plants, and that, as expected, the non-phloem-limited virus (*i.e.* CaMV) has a significantly
stronger impact on plant hosts than the phloem-limited virus (*i.e.* TuYV). Overall, viral infection
with CaMV tends to have effects on metabolic pathways with strong potential implications for
insect-vector / plant-host interactions, while TuYV only weakly alters these pathways. For example,
the strong gene downregulations in the jasmonic acid, ethylene and glucosinolate biosynthetic
processes (Figure 9a-c) in CaMV-infected plants could be responsible for the observed alterations
of aphid feeding behavior and performance. Next steps could consist of functional validation of

some candidate genes identified in our study for their role in viral manipulation and consequently
potential impacts on viral transmission.

**Data availability**

The raw RNA-seq data are available under project number PRJEB49403 at the European
Nucleotide Archive (<https://www.ebi.ac.uk/ena/browser/view/PRJEB49403>).

**Author contributions**

Conceptualization, Q.C., V.B., M.P. and M.D.; methodology, Q.C., V.G. and M.D.; software, Q.C.,
836 A.V. and C.R.; validation, Q.C. and V.G.; formal analysis, Q.C., A.V., C.R. and M.D.;
investigation, Q.C. and V.G.; Data curation, Q.C., A.V. and C.R.; Writing – Original Draft
Preparation, Q.C., M.P. and M.D.; Writing – Review & Editing, Q.C., A.V., C.R., V.B., M.P. and
839 M.D.; Visualization, Q.C.; supervision, M.P. and M.D.; project administration, M.D.; funding
acquisition, M.P. and M.D.

**Acknowledgments**

We thank Claire Villeroy for aphid rearing and the experimental unit of INRAE Grand Est –
Colmar (UEAV) for help with plant production. This work was supported by a public grant
overseen by the French National Research Agency (ANR) (reference: ROME ANR-18-CE20-0017-
01). Dr. Quentin Chesnais was supported by Région Grand Est (Soutien aux jeunes chercheurs,
reference: 18_GE5_013). The funding sources had no role in the study design; in the collection,
analysis, and interpretation of data; in the writing of the report; and in the decision to submit the
article for publication. The authors declare no conflict of interest.

**References**

- A'Brook, J. (1973) Observations on different methods of aphid trapping. *Annals of Applied Biology*,
74, 263–277. <https://doi.org/10.1111/j.1744-7348.1973.tb07747.x>.
- Afgan, E., Baker, D., van den Beek, M., Blankenberg, D., Bouvier, D., Čech, M., et al. (2016) The
Galaxy platform for accessible, reproducible and collaborative biomedical analyses: 2016 update.
*Nucleic Acids Research*, 44, W3–W10. <https://doi.org/10.1093/nar/gkw343>.
- Annacondia, M.L., Markovic, D., Reig-Valiente, J.L., Scaltsoyiannes, V., Pieterse, C.M.J.,
Ninkovic, V., et al. (2021) Aphid feeding induces the relaxation of epigenetic control and the
associated regulation of the defense response in *Arabidopsis*. *New Phytologist*, 230, 1185–1200.
<https://doi.org/10.1111/nph.17226>.
- Anstead, J., Samuel, P., Song, N., Wu, C., Thompson, G.A. & Goggin, F. (2010) Activation of
ethylene-related genes in response to aphid feeding on resistant and susceptible melon and tomato
plants. *Entomologia Experimentalis et Applicata*, 134, 170–181. <https://doi.org/10.1111/j.1570-7458.2009.00945.x>.
- Barco, B. & Clay, N.K. (2020) Hierarchical and Dynamic Regulation of Defense-Responsive
Specialized Metabolism by WRKY and MYB Transcription Factors. *Frontiers in Plant Science*, 10,
1775. <https://doi.org/10.3389/fpls.2019.01775>.
- Barratt, D.H.P., Kölling, K., Graf, A., Pike, M., Calder, G., Findlay, K., et al. (2011) Callose
Synthase GSL7 Is Necessary for Normal Phloem Transport and Inflorescence Growth in
*Arabidopsis*. *Plant Physiology*, 155, 328–341. <https://doi.org/10.1104/pp.110.166330>.
- Bencharki, B., Boissinot, S., Revollon, S., Ziegler-Graff, V., Erdinger, M., Wiss, L., et al. (2010)
Phloem Protein Partners of *Cucurbit aphid borne yellows virus*: Possible Involvement of Phloem
Proteins in Virus Transmission by Aphids. *Molecular Plant-Microbe Interactions*®, 23, 799–810.
<https://doi.org/10.1094/MPMI-23-6-0799>.
- Blanc, S., Drucker, M. & Uzest-Bonhomme, M. (2014) Localizing viruses in their insect vectors.
*Annual Review of Phytopathology*, 52, 403–425. <https://doi.org/10.1146/annurev-phyto-102313-045920>.
- Blevins, T., Rajeswaran, R., Shivaprasad, P.V., Beknazariants, D., Si-Ammour, A., Park, H.-S., et
al. (2006) Four plant Dicers mediate viral small RNA biogenesis and DNA virus induced silencing.
*Nucleic Acids Research*, 34, 6233–6246. <https://doi.org/10.1093/nar/gkl886>.
- Boex-Fontvieille, E., Rustgi, S., Wettstein, D. von, Reinbothe, S. & Reinbothe, C. (2015) Water-
soluble chlorophyll protein is involved in herbivore resistance activation during greening of
*Arabidopsis thaliana*. *Proceedings of the National Academy of Sciences*, 112, 7303–7308.
<https://doi.org/10.1073/pnas.1507714112>.
- Bogaert, F., Marmonier, A., Pichon, E., Boissinot, S., Ziegler-Graff, V., Chesnais, Q., et al. (2020)
Impact of Mutations in *Arabidopsis thaliana* Metabolic Pathways on Polerovirus Accumulation,
Aphid Performance, and Feeding Behavior. *Viruses*, 12, 146. <https://doi.org/10.3390/v12020146>.
- Boisson, B., Giglione, C. & Meinnel, T. (2003) Unexpected Protein Families Including Cell
Defense Components Feature in the N-Myristoylome of a Higher Eukaryote. *Journal of Biological*
*Chemistry*, 278, 43418–43429. <https://doi.org/10.1074/jbc.M307321200>.
- Bosque-Pérez, N.A. & Eigenbrode, S.D. (2011) The influence of virus-induced changes in plants on
aphid vectors: insights from luteovirus pathosystems. *Virus Research*, 159, 201–205.
<https://doi.org/10.1016/j.virusres.2011.04.020>.
- Brault, V., Herrbach, E. & Reinbold, C. (2007) Electron microscopy studies on luteovirid
transmission by aphids. *Micron (Oxford, England: 1993)*, 38, 302–312.
<https://doi.org/10.1016/j.micron.2006.04.005>.
- Briat, J.-F., Ravet, K., Arnaud, N., Duc, C., Boucherez, J., Touraine, B., et al. (2010) New insights
into ferritin synthesis and function highlight a link between iron homeostasis and oxidative stress in
plants. *Annals of Botany*, 105, 811–822. <https://doi.org/10.1093/aob/mcp128>.

Campbell, B.C., Jones, K.C. & Dreyer, D.L. (1986) Discriminative behavioral responses by aphids
to various plant matrix Polysaccharides. *Entomologia Experimentalis et Applicata*, 41, 17–24.
<https://doi.org/10.1111/j.1570-7458.1986.tb02166.x>.

Carbonell, A. & Carrington, J.C. (2015) Antiviral roles of plant ARGONAUTES. *Current Opinion*
*in Plant Biology*, 27, 111–117. <https://doi.org/10.1016/j.pbi.2015.06.013>.

Carr, J.P., Tungadi, T., Donnelly, R., Bravo-Cazar, A., Rhee, S.-J., Watt, L.G., et al. (2020)
Modelling and manipulation of aphid-mediated spread of non-persistently transmitted viruses. *Virus*
*Research*, 277, 197845. <https://doi.org/10.1016/j.virusres.2019.197845>.

Chen, X., Smith, M.D., Fitzpatrick, L. & Schnell, D.J. (2002) In Vivo Analysis of the Role of
atTic20 in Protein Import into Chloroplasts. *The Plant Cell*, 14, 641–654.
<https://doi.org/10.1105/tpc.010336>.

Chen, X.-R., Wang, Y., Zhao, H.-H., Zhang, X.-Y., Wang, X.-B., Li, D.-W., et al. (2018) Brassica
yellows virus' movement protein upregulates anthocyanin accumulation, leading to the
development of purple leaf symptoms on *Arabidopsis thaliana*. *Scientific Reports*, 8, 16273.
<https://doi.org/10.1038/s41598-018-34591-5>.

Cheng, G., Yang, Z., Zhang, H., Zhang, J. & Xu, J. (2020) Remorin interacting with PCaP1 impairs
Turnip mosaic virus intercellular movement but is antagonised by VPg. *New Phytologist*, 225,
2122–2139. <https://doi.org/10.1111/nph.16285>.

Chesnais, Q., Couty, A., Uzest, M., Brault, V. & Ameline, A. (2019a) Plant infection by two
different viruses induce contrasting changes of vectors fitness and behavior. *Insect Science*, 26, 86–
96. <https://doi.org/10.1111/1744-7917.12508>.

Chesnais, Q., Mauck, K.E., Bogaert, F., Bamière, A., Catterou, M., Spicher, F., et al. (2019b) Virus
effects on plant quality and vector behavior are species specific and do not depend on host
physiological phenotype. *Journal of Pest Science*, 92, 791–804. <https://doi.org/10.1007/s10340-019-01082-z>.

Chesnais, Q., Verdier, M., Burckbuchler, M., Brault, V., Pooggin, M. & Drucker, M. (2021)
Cauliflower mosaic virus protein P6-TAV plays a major role in alteration of aphid vector feeding
behaviour but not performance on infected *Arabidopsis*. *Molecular Plant Pathology*, 22, 911–920.
<https://doi.org/10.1111/mpp.13069>.

Cole, R.A. (1997) The relative importance of glucosinolates and amino acids to the development of
two aphid pests *Brevicoryne brassicae* and *Myzus persicae* on wild and cultivated brassica species.
*Entomologia Experimentalis et Applicata*, 85, 121–133. <https://doi.org/10.1046/j.1570-7458.1997.00242.x>.

Cosson, P., Sofer, L., Le, Q.H., Léger, V., Schurdi-Levraud, V., Whitham, S.A., et al. (2010)
RTM3, which controls long-distance movement of potyviruses, is a member of a new plant gene
family encoding a meprin and TRAF homology domain-containing protein. *Plant Physiology*, 154,
222–232. <https://doi.org/10.1104/pp.110.155754>.

Dáder, B., Then, C., Berthelot, E., Ducouso, M., Ng, J.C.K. & Drucker, M. (2017) Insect
transmission of plant viruses: Multilayered interactions optimize viral propagation. *Insect Science*,
24, 929–946. <https://doi.org/10.1111/1744-7917.12470>.

Dietzgen, R.G., Mann, K.S. & Johnson, K.N. (2016) Plant Virus-Insect Vector Interactions: Current
and Potential Future Research Directions. *Viruses*, 8. <https://doi.org/10.3390/v8110303>.

Fennell, J.T., Wilby, A., Sobeih, W. & Paul, N.D. (2020) New understanding of the direct effects of
spectral balance on behaviour in *Myzus persicae*. *Journal of Insect Physiology*, 126, 104096.
<https://doi.org/10.1016/j.jinsphys.2020.104096>.

Fereres, A. & Moreno, A. (2009) Behavioural aspects influencing plant virus transmission by
homopteran insects. *Virus Research*, 141, 158–168. <https://doi.org/10.1016/j.virusres.2008.10.020>.

Fereres, A. & Raccach, B. (2015) Plant Virus Transmission by Insects. In: John Wiley & Sons Ltd
(Ed.) eLS. Chichester, UK: John Wiley & Sons, Ltd, pp. 1–12.

- Giordanengo, P. (2014) EPG-Calc: a PHP-based script to calculate electrical penetration graph
(EPG) parameters. *Arthropod-Plant Interactions*, 8, 163–169. [https://doi.org/10.1007/s11829-014-](https://doi.org/10.1007/s11829-014-9298-z)
9298-z.
- Golyaev, V., Candresse, T., Rabenstein, F. & Pooggin, M.M. (2019) Plant virome reconstruction
and antiviral RNAi characterization by deep sequencing of small RNAs from dried leaves. *Scientific*
*Reports*, 9, 19268. <https://doi.org/10.1038/s41598-019-55547-3>.
- Haas, G., Azevedo, J., Moissiard, G., Geldreich, A., Himber, C., Bureau, M., et al. (2008) Nuclear
import of CaMV P6 is required for infection and suppression of the RNA silencing factor DRB4.
*The EMBO Journal*, 27, 2102–2112. <https://doi.org/10.1038/emboj.2008.129>.
- He, Y., Galant, A., Pang, Q., Strul, J.M., Balogun, S.F., Jez, J.M., et al. (2011) Structural and
functional evolution of isopropylmalate dehydrogenases in the leucine and glucosinolate pathways
of *Arabidopsis thaliana*. *The Journal of Biological Chemistry*, 286, 28794–28801.
<https://doi.org/10.1074/jbc.M111.262519>.
- Hodge, S., Ward, J.L., Beale, M.H., Bennett, M., Mansfield, J.W. & Powell, G. (2013) Aphid-
induced accumulation of trehalose in *Arabidopsis thaliana* is systemic and dependent upon aphid
density. *Planta*, 237, 1057–1064. <https://doi.org/10.1007/s00425-012-1826-4>.
- Howe, G.A. & Jander, G. (2008) Plant Immunity to Insect Herbivores. *Annual Review of Plant*
*Biology*, 59, 41–66. <https://doi.org/10.1146/annurev.arplant.59.032607.092825>.
- Kadota, Y., Shirasu, K. & Zipfel, C. (2015) Regulation of the NADPH Oxidase RBOHD During
Plant Immunity. *Plant and Cell Physiology*, 56, 1472–1480. <https://doi.org/10.1093/pcp/pcv063>.
- Kagale, S., Koh, C., Nixon, J., Bollina, V., Clarke, W.E., Tuteja, R., et al. (2014) The emerging
biofuel crop *Camelina sativa* retains a highly undifferentiated hexaploid genome structure. *Nature*
*Communications*, 5, 3706. <https://doi.org/10.1038/ncomms4706>.
- Kennedy, J.S., Day, M.F. & Eastop, V.F. (1962) A conspectus of aphids as vectors of plant viruses.
Eastern Press Ltd., London and Reading. London: Commonwealth Inst. Entomol.
- Kettles, G.J., Drurey, C., Schoonbeek, H., Maule, A.J. & Hogenhout, S.A. (2013) Resistance of *A*
*rabidopsis thaliana* to the green peach aphid, *Myzus persicae*, involves camalexin and is
regulated by micro RNA s. *New Phytologist*, 198, 1178–1190. <https://doi.org/10.1111/nph.12218>.
- Kim, J.H. & Jander, G. (2007) *Myzus persicae* (green peach aphid) feeding on *Arabidopsis* induces
the formation of a deterrent indole glucosinolate: Aphid-induced glucosinolate changes. *The Plant*
*Journal*, 49, 1008–1019. <https://doi.org/10.1111/j.1365-313X.2006.03019.x>.
- Kim, J.H., Lee, B.W., Schroeder, F.C. & Jander, G. (2008) Identification of indole glucosinolate
breakdown products with antifeedant effects on *Myzus persicae* (green peach aphid). *The Plant*
*Journal: For Cell and Molecular Biology*, 54, 1015–1026. [https://doi.org/10.1111/j.1365-](https://doi.org/10.1111/j.1365-313X.2008.03476.x)
313X.2008.03476.x.
- Kim, J.I., Dolan, W.L., Anderson, N.A. & Chapple, C. (2015) Indole Glucosinolate Biosynthesis
Limits Phenylpropanoid Accumulation in *Arabidopsis thaliana*. *The Plant Cell*, 27, 1529–1546.
<https://doi.org/10.1105/tpc.15.00127>.
- Kloth, K.J., Abreu, I.N., Delhomme, N., Petřík, I., Villard, C., Ström, C., et al. (2019) PECTIN
ACETYLESTERASE9 Affects the Transcriptome and Metabolome and Delays Aphid Feeding.
*Plant Physiology*, 181, 1704–1720. <https://doi.org/10.1104/pp.19.00635>.
- Kloth, K.J. & Kormelink, R. (2020) Defenses against Virus and Vector: A Phloem-Biological
Perspective on RTM- and SLI1-Mediated Resistance to Potyviruses and Aphids. *Viruses*, 12, 129.
<https://doi.org/10.3390/v12020129>.
- Kloth, K.J., Shah, P., Broekgaarden, C., Ström, C., Albrechtsen, B.R. & Dicke, M. (2021) SLI1
confers broad-spectrum resistance to phloem-feeding insects. *Plant, Cell & Environment*, 44, 2765–
2776. <https://doi.org/10.1111/pce.14064>.
- Kloth, K.J., Wiegiers, G.L., Busscher-Lange, J., Haarst, J.C. van, Kruijer, W., Bouwmeester, H.J., et
al. (2016) AtWRKY22 promotes susceptibility to aphids and modulates salicylic acid and jasmonic

acid signalling. *Journal of Experimental Botany*, 67, 3383–3396.
<https://doi.org/10.1093/jxb/erw159>.

Kubina, J., Geldreich, A., Gales, J.P., Baumberger, N., Bouton, C., Ryabova, L.A., et al. (2021)
Nuclear export of plant pararetrovirus mRNAs involves the TREX complex, two viral proteins and
the highly structured 5' leader region. *Nucleic Acids Research*, 49, 8900–8922.
<https://doi.org/10.1093/nar/gkab653>.

Kuśnierczyk, A., Winge, P., Jørstad, T.S., TrocZYńska, J., Rossiter, J.T. & Bones, A.M. (2008)
Towards global understanding of plant defence against aphids--timing and dynamics of early
Arabidopsis defence responses to cabbage aphid (*Brevicoryne brassicae*) attack. *Plant, Cell &*
*Environment*, 31, 1097–1115. <https://doi.org/10.1111/j.1365-3040.2008.01823.x>.

Liu, Y., Ahn, J.-E., Datta, S., Salzman, R.A., Moon, J., Huyghues-Despointes, B., et al. (2005)
Arabidopsis Vegetative Storage Protein Is an Anti-Insect Acid Phosphatase. *Plant Physiology*, 139,
1545–1556. <https://doi.org/10.1104/pp.105.066837>.

Louis, J., Lorenc-Kukula, K., Singh, V., Reese, J., Jander, G. & Shah, J. (2010) Antibiosis against
the green peach aphid requires the Arabidopsis thaliana MYZUS PERSICAE-INDUCED LIPASE1
gene: Plant-aphid interaction. *The Plant Journal*, 64, 800–811. <https://doi.org/10.1111/j.1365-313X.2010.04378.x>.

Lu, H., Zhu, J., Yu, J., Chen, X., Kang, L. & Cui, F. (2020) A Symbiotic Virus Facilitates Aphid
Adaptation to Host Plants by Suppressing Jasmonic Acid Responses. *Molecular plant-microbe*
*interactions: MPMI*, 33, 55–65. <https://doi.org/10.1094/MPMI-01-19-0016-R>.

Malik, M.R., Tang, J., Sharma, N., Burkitt, C., Ji, Y., Mykytyshyn, M., et al. (2018) Camelina
sativa, an oilseed at the nexus between model system and commercial crop. *Plant Cell Reports*, 37,
1367–1381. <https://doi.org/10.1007/s00299-018-2308-3>.

Martin, B., Collar, J.L., Tjallingii, W.F. & Fereres, A. (1997) Intracellular ingestion and salivation
by aphids may cause the acquisition and inoculation of non-persistently transmitted plant viruses. *J.*
*Gen. Virol.*, 78, 2701–2705.

Mauck, K., Bosque-Pérez, N.A., Eigenbrode, S.D., De Moraes, C.M. & Mescher, M.C. (2012)
Transmission mechanisms shape pathogen effects on host–vector interactions: evidence from plant
viruses. *Functional Ecology*, 26, 1162–1175. <https://doi.org/10.1111/j.1365-2435.2012.02026.x>.

Mauck, K.E. & Chesnais, Q. (2020) A synthesis of virus-vector associations reveals important
deficiencies in studies on host and vector manipulation by plant viruses. *Virus Research*, 285,
197957. <https://doi.org/10.1016/j.virusres.2020.197957>.

Mauck, K.E., Chesnais, Q. & Shapiro, L.R. (2018) Evolutionary Determinants of Host and Vector
Manipulation by Plant Viruses. *Advances in Virus Research*, 101, 189–250.
<https://doi.org/10.1016/bs.aivir.2018.02.007>.

Mauck, K.E., De Moraes, C.M. & Mescher, M.C. (2014) Biochemical and physiological
mechanisms underlying effects of Cucumber mosaic virus on host-plant traits that mediate
transmission by aphid vectors. *Plant, Cell & Environment*, 37, 1427–1439.
<https://doi.org/10.1111/pce.12249>.

Mauck, K.E., De Moraes, C.M. & Mescher, M.C. (2010) Deceptive chemical signals induced by a
plant virus attract insect vectors to inferior hosts. *Proceedings of the National Academy of Sciences*
*of the United States of America*, 107, 3600–3605. <https://doi.org/10.1073/pnas.0907191107>.

Mauck, K.E., Kenney, J. & Chesnais, Q. (2019) Progress and challenges in identifying molecular
mechanisms underlying host and vector manipulation by plant viruses. *Current Opinion in Insect*
*Science*, 33, 7–18. <https://doi.org/10.1016/j.cois.2019.01.001>.

Mewis, I., Appel, H.M., Hom, A., Raina, R. & Schultz, J.C. (2005) Major signaling pathways
modulate Arabidopsis glucosinolate accumulation and response to both phloem-feeding and
chewing insects. *Plant Physiology*, 138, 1149–1162. <https://doi.org/10.1104/pp.104.053389>.

- Morante-Carriel, J., Sellés-Marchart, S., Martínez-Márquez, A., Martínez-Esteso, M.J., Luque, I. &
Bru-Martínez, R. (2014) RNA isolation from loquat and other recalcitrant woody plants with high
quality and yield. *Analytical Biochemistry*, 452, 46–53. <https://doi.org/10.1016/j.ab.2014.02.010>.
- Moreno, A., Tjallingii, W.F., Fernandez-Mata, G. & Fereres, A. (2012) Differences in the
mechanism of inoculation between a semi-persistent and a non-persistent aphid-transmitted plant
virus. *Journal of General Virology*, 93, 662–667. <https://doi.org/10.1099/vir.0.037887-0>.
- Mulot, M., Monsion, B., Boissinot, S., Rastegar, M., Meyer, S., Bochet, N., et al. (2018)
Transmission of Turnip yellows virus by *Myzus persicae* Is Reduced by Feeding Aphids on
Double-Stranded RNA Targeting the Ephrin Receptor Protein. *Frontiers in Microbiology*, 9, 457.
<https://doi.org/10.3389/fmicb.2018.00457>.
- Ng, J.C.K. & Falk, B.W. (2006) Virus-vector interactions mediating nonpersistent and
semipersistent transmission of plant viruses. *Annual Review of Phytopathology*, 44, 183–212.
<https://doi.org/10.1146/annurev.phyto.44.070505.143325>.
- Omae, N., Suzuki, M. & Ugaki, M. (2020) The genome of the Cauliflower mosaic virus, a plant
pararetrovirus, is highly methylated in the nucleus. *FEBS Letters*, 594, 1974–1988.
<https://doi.org/10.1002/1873-3468.13852>.
- Palacios, I., Drucker, M., Blanc, S., Leite, S., Moreno, A. & Fereres, A. (2002) Cauliflower mosaic
virus is preferentially acquired from the phloem by its aphid vectors. *The Journal of General*
*Virology*, 83, 3163–3171.
- Pegadaraju, V., Knepper, C., Reese, J. & Shah, J. (2005) Premature leaf senescence modulated by
the Arabidopsis PHYTOALEXIN DEFICIENT4 gene is associated with defense against the
phloem-feeding green peach aphid. *Plant Physiology*, 139, 1927–1934.
<https://doi.org/10.1104/pp.105.070433>.
- Raffaele, S., Bayer, E., Lafarge, D., Cluzet, S., German Retana, S., Boubekur, T., et al. (2009)
Remorin, a Solanaceae Protein Resident in Membrane Rafts and Plasmodesmata, Impairs *Potato*
*virus X* Movement. *The Plant Cell*, 21, 1541–1555. <https://doi.org/10.1105/tpc.108.064279>.
- Rodriguez, P.A. & Bos, J.I.B. (2013) Toward understanding the role of aphid effectors in plant
infestation. *Molecular plant-microbe interactions: MPMI*, 26, 25–30.
<https://doi.org/10.1094/MPMI-05-12-0119-FI>.
- Schuck, J., Gursinsky, T., Pantaleo, V., Burguán, J. & Behrens, S.-E. (2013) AGO/RISC-mediated
antiviral RNA silencing in a plant in vitro system. *Nucleic Acids Research*, 41, 5090–5103.
<https://doi.org/10.1093/nar/gkt193>.
- Shivaprasad, P.V., Rajeswaran, R., Blevins, T., Schoelz, J., Meins, F., Jr, Hohn, T., et al. (2008)
The CaMV transactivator/viroplasm interferes with RDR6-dependent trans-acting and secondary
siRNA pathways in Arabidopsis. *Nucleic Acids Research*, 36, 5896–5909.
<https://doi.org/10.1093/nar/gkn590>.
- Singh, V., Louis, J., Ayre, B.G., Reese, J.C., Pegadaraju, V. & Shah, J. (2011) TREHALOSE
PHOSPHATE SYNTHASE11-dependent trehalose metabolism promotes Arabidopsis thaliana
defense against the phloem-feeding insect *Myzus persicae*. *The Plant Journal: For Cell and*
*Molecular Biology*, 67, 94–104. <https://doi.org/10.1111/j.1365-313X.2011.04583.x>.
- Singh, V. & Shah, J. (2012) Tomato responds to green peach aphid infestation with the activation of
trehalose metabolism and starch accumulation. *Plant Signaling & Behavior*, 7, 605–607.
<https://doi.org/10.4161/psb.20066>.
- Suzuki, N., Miller, G., Morales, J., Shulaev, V., Torres, M.A. & Mittler, R. (2011) Respiratory burst
oxidases: the engines of ROS signaling. *Current Opinion in Plant Biology*, 14, 691–699.
<https://doi.org/10.1016/j.pbi.2011.07.014>.
- Tjallingii, W.F. (1988) Electrical recording of stylet penetration activities. Aphids, their biology,
natural enemies and control. Amsterdam: Elsevier Science Publishers, pp. 95–108.

Tjallingii, W.F. & Hogen Esch, T. (1993) Fine structure of aphid stylet routes in plant tissues in
correlation with EPG signals. *Physiol. Entomol.*, 18, 317–328. [https://doi.org/10.1111/j.1365-](https://doi.org/10.1111/j.1365-3032.1993.tb00604.x)
[3032.1993.tb00604.x](https://doi.org/10.1111/j.1365-3032.1993.tb00604.x).

Tsuge, S., Kobayashi, K., Nakayashiki, H., Okuno, T. & Furusawa, I. (1994) Replication of
cauliflower mosaic virus ORF I mutants in turnip protoplasts. *Ann Phytopath Soc Japan*, 60, 27–35.

Veidt, I., Lot, H., Leiser, M., Scheidecker, D., Guilley, H., Richards, K., et al. (1988) Nucleotide
sequence of beet western yellows virus RNA. *Nucleic Acids Research*, 16, 9917–9932.
<https://doi.org/10.1093/nar/16.21.9917>.

Weber, G., Oswald, S. & Zollner, U. (1986) Die Wirtseignung von Rapsorten unterschiedlichen
Glucosinolatgehalts fuer *Brevicoryne brassicae* (L.) und *Myzus persicae* (Sulzer) (Hemiptera,
*Aphididae*). *Zeitschrift für Pflanzenkrankheiten und Pflanzenschutz*, 93, 113–124.

Westwood, J.H., Groen, S.C., Du, Z., Murphy, A.M., Anggoro, D.T., Tungadi, T., et al. (2013) A
Trio of Viral Proteins Tunes Aphid-Plant Interactions in *Arabidopsis thaliana* Rao, A.L.N. (Ed.).
*PLoS ONE*, 8, e83066. <https://doi.org/10.1371/journal.pone.0083066>.

Xie, B., Wang, X., Zhu, M., Zhang, Z. & Hong, Z. (2011) *Cals7* encodes a callose synthase
responsible for callose deposition in the phloem: A phloem-specific callose synthase. *The Plant*
*Journal*, 65, 1–14. <https://doi.org/10.1111/j.1365-313X.2010.04399.x>.

Yu, Y., Ji, L., Le, B.H., Zhai, J., Chen, J., Luscher, E., et al. (2017) ARGONAUTE10 promotes the
degradation of miR165/6 through the SDN1 and SDN2 exonucleases in *Arabidopsis* Zamore, P.
(Ed.). *PLOS Biology*, 15, e2001272. <https://doi.org/10.1371/journal.pbio.2001272>.

Zang, Y.-X., Lim, M.-H., Park, B.-S., Hong, S.-B. & Kim, D.H. (2008) Metabolic engineering of
indole glucosinolates in Chinese cabbage plants by expression of *Arabidopsis* CYP79B2,
CYP79B3, and CYP83B1. *Molecules and Cells*, 25, 231–241.

Zhang, C., Shi, H., Chen, L., Wang, X., Lü, B., Zhang, S., et al. (2011) Harpin-induced expression
and transgenic overexpression of the phloem protein gene *AtPP2-A1* in *Arabidopsis* repress phloem
feeding of the green peach aphid *Myzus persicae*. *BMC Plant Biology*, 11, 11.
<https://doi.org/10.1186/1471-2229-11-11>.

Zhang, D., Song, Y.H., Dai, R., Lee, T.G. & Kim, J. (2020) Aldoxime Metabolism Is Linked to
Phenylpropanoid Production in *Camelina sativa*. *Frontiers in Plant Science*, 11, 17.
<https://doi.org/10.3389/fpls.2020.00017>.

Zhang, Z., Ober, J.A. & Kliebenstein, D.J. (2006) The Gene Controlling the Quantitative Trait
Locus *EPITHIOSPECIFIER MODIFIER1* Alters Glucosinolate Hydrolysis and Insect Resistance in
*Arabidopsis*. *The Plant Cell*, 18, 1524–1536. <https://doi.org/10.1105/tpc.105.039602>.

Zorzatto, C., Machado, J.P.B., Lopes, K.V.G., Nascimento, K.J.T., Pereira, W.A., Brustolini,
O.J.B., et al. (2015) NIK1-mediated translation suppression functions as a plant antiviral immunity
mechanism. *Nature*, 520, 679–682. <https://doi.org/10.1038/nature14171>.

Zvereva, A.S., Golyaev, V., Turco, S., Gubaeva, E.G., Rajeswaran, R., Schepetilnikov, M.V., et al.
(2016) Viral protein suppresses oxidative burst and salicylic acid-dependent autophagy and
facilitates bacterial growth on virus-infected plants. *The New Phytologist*, 211, 1020–1034.
<https://doi.org/10.1111/nph.13967>.

Zvereva, A.S. & Pooggin, M.M. (2012) Silencing and innate immunity in plant defense against
viral and non-viral pathogens. *Viruses*, 4, 2578–2597. <https://doi.org/10.3390/v4112578>.

**Supplemental data**

Provided in this file:

Table S1: Oligonucleotides used for RT-qPCR

Table S2: Aligned reads for transcriptome profiling

Figure S1. Validation of Illumina RNA-seq expression data by quantitative reverse-transcription
PCR (RT-qPCR)

Figure S2: Gene ontology analysis showing the Top 25 GO of deregulated processes

Figure S3: Supplementary heatmaps

Provided as extra files:

Supplementary Dataset S1 Plant mRNA-seq.xlsx

Supplementary Dataset S2 Heatmaps DEGs List.xlsx

Supplementary Sequence Information S1 on CaMV and TuYV.docx

Table S1: Oligonucleotides used for RT-qPCR

Gene	Organism	Primers
AT5G25760 (PEX4)	A. thaliana	Forward primer TGCAACCTCCTCAAGTTCGA Reverse primer GCAGGACTCCAAGCATTCTT
AT2G18700 (TPS11)	A. thaliana	Forward primer AAGTTTTGGGCGATGGGTCA Reverse primer CGAGAACCACTTTCCCACGA
AT1G61800 (GPT2)	A. thaliana	Forward primer AGTGTCAATTTCTTGATCAGACCATC Reverse primer CCAGTAGCGACACACCTCAAT
AT4G39030 (EDS5)	A. thaliana	Forward primer ACCCTAGCGACAAATGACAGC Reverse primer TCACTTGCTCCATTATTAACCTGC
AT1G31280 (AGO2)	A. thaliana	Forward primer ATGCTGACAAGGCTGCTTCT Reverse primer CAGAAGACGAAGACGCTCCA
AT4G26530 (FBA5)	A. thaliana	Forward primer TTGGTTGCCATTTGGTTGTGT Reverse primer CTGAAGAGGACGAGGATGCC
AT3G26830 (PAD3)	A. thaliana	Forward primer AGGGCAAGGAAAATGTCGGT Reverse primer CAGGGGTAAGAGGACGAGGA
AT5G42650 (AOS)	A. thaliana	Forward primer TCACGATGGGAGCGATTGAG Reverse primer ACCGTATTGAGCCGTAACCG
AT4G13770 (CYP83A1)	A. thaliana	Forward primer AGGAACAACGGTCAACGTCA Reverse primer CGGTCCCCATTCTTTCTCGT

Table S2: Aligned reads for transcriptome profiling

Sample name	Aligned reads	Assigned Reads	Mapped Ratio
Ara_M2	34,670,703	31,998,776	92,3%
Ara_M3	31,050,325	28,078,404	90,4%
Ara_M4	29,883,629	26,930,049	90,1%
Ara_C1	33,833,716	27,677,748	81,8%
Ara_C2	30,911,854	24,741,132	80,0%
Ara_C3	30,653,084	24,528,34	80,0%
Ara_T1	29,290,278	25,109,740	85,7%
Ara_T2	32,381,425	28,898,406	89,2%
Ara_T3	32,197,738	28,344,993	88,0%

Sample name	Aligned reads	Assigned Reads	Mapped Ratio
Cam_M1	33,109,309	22,574,696	68,2%
Cam_M2	32,233,737	22,145,697	68,7%
Cam_M3	33,890,114	22,984,537	67,8%
Cam_C1	38,817,320	24,936,094	64,2%
Cam_C2	31,267,901	20,165,623	64,5%
Cam_C3	28,166,913	17,126,400	60,8%
Cam_T1	29,673,896	20,168,532	68,0%
Cam_T2	29,764,440	20,762,349	69,8%
Cam_T3	34,740,771	24,509,728	70,6%

Figure S1. Validation of Illumina RNA-seq expression data by quantitative reverse-transcription PCR (RT-qPCR). a) CaMV-infected Arabidopsis. b) TuYV-infected Arabidopsis. The y-axis presents the normalized log₂ fold change of expression derived from Illumina RNA-seq read counts and PCR $\Delta\Delta C_T$, respectively. The TAIR gene loci of the tested mRNAs are: At_AGO2, AT1G31280; At_AOS, AT5G42650; At_CYP83A1, AT4G13770; At_EDS5, AT4G39030; At_FBA5, AT4G26530; At_GPT2, AT1G61800; At_PAD3, AT3G26830; At_TPS11, AT2G18700.

a) Common DEGS between CaMV- and TuYV-infected Arabidopsis

b) Common DEGS between CaMV- and TuYV-infected Camelina

Figure S2: Gene ontology analysis showing the Top 25 GO of deregulated processes. a) common DEGS (n=956)
between CaMV-infected and TuYV-infected Arabidopsis and b) common DEGS (n=6,692) between CaMV-infected
and TuYV-infected Camelina. GO IDs and corresponding GO terms are specified in the vertical axis. For each category
(BP: Biological Process, CC: Cellular Component and MF: Molecular Function), GOs are sorted according to
decreasing log₂ (1/p-value), also indicated by the color of each spot, in order to place most significantly enriched GOs
on top of the graph. The absolute number of DEGs that matched the GO term is indicated by the size of each spot,
whereas the horizontal axis shows the ratio of DEGs vs. all genes belonging to the GO term.

**Supplementary heatmaps**

Callose deposition is induced via pathogen molecular pattern- and pathogen effector-triggered
 immunity pathways. We found that expression of genes related to callose deposition was only
 slightly deregulated in aphid-infested infected plants except for a strong upregulation of *SWEET13*
 in *Camelina* infected with CaMV or TuYV, and a slight downregulation of *BHLH89* in CaMV-
 infected *Arabidopsis* and *Camelina*. *SWEET13* is a sugar transporter and there is no direct link with
 callose (at least I did not find any, even if there is callose deposition in the *Arabidopsis* GO
 annotation). *BHLH89* is a transcription factor that is upstream of callose deposition; (no more
 information). *UGPI* was virus-specifically upregulated in TuYV-infected *Camelina* and
 *Arabidopsis* and downregulated in CaMV-infected plants. *UGPI* (AT3G03250) is a UDP-glucose
 pyrophosphorylase and involved in the first synthesis steps of cellulose and other sugar polymers
 (PMID 29569779) such as callose.

Figure S3. Hierarchical clustering of differentially expressed genes (DEGs) related to a) Defense response by callose
 deposition (GO:0052542), b) Callose deposition in cell wall (GO:0052543) and c) Cellular amino acid biosynthetic
 process (GO:0008652) in CaMV- and TuYV-infected *Arabidopsis thaliana* and *Camelina sativa* compared to their
 mock-inoculated relatives (Supplementary Dataset S2). The color keys show log₂fold changes as indicated below the
 keys in gradients from the minimal value in cyan to the maximal value in magenta.

Reviewer comments:

Reviewer #1 (Comments for the Author):

This is a technically sound and interesting paper that presents data on vector behavior/fecundity in relation to infected and healthy hosts, contrasting two very different viruses each infecting two related, but distinct host species. This is followed up with a transcriptomic analysis of a different set of infected and healthy hosts of each species x virus combination in the same stage of growth/disease progression as that used for behavior/fecundity experiments. I have just a few suggestions for improvement of the manuscript.

1. It would be nice to know why virus replication levels were only quantified through the transcriptome data, and not using an RT-qPCR assay that avoids the issue of TuYV not having poly A tails. Although differences were still found, it would be good if these were verified with a more standard diagnostic method for evaluating virus replication in the host.

As an alternative approach to assess virus accumulation we added to Figure 1 Western blots targeting coat proteins of both viruses in both hosts. They show that TuYV accumulates to similar levels in Arabidopsis and Camelina and that CaMV accumulates more strongly in Camelina than in Arabidopsis. The TuYV data are also supported by previous work using RT-qPCR (Claudel et al. 2018, doi 10.3390/ijms19082316).

The methods section, the results sections “Plant Phenotype” and “Quality of RNA and sequence alignment data” and the legend to Figure 1 were accordingly adapted:

Method section:

“Western blotting

Total protein extracts were prepared from leaves, separated by SDS-PAGE and transferred to nitrocellulose as described previously (Chesnais et al., 2021). Western blots were performed using antisera raised against CaMV P4 (Champagne et al., 2004) and TuYV CP (Bruyère et al., 1997). Secondary antibodies were horseradish peroxidase conjugates, and bound antibodies were revealed by enhanced chemiluminescence.”

Plant phenotype section

“Western blot analysis showed that TuYV accumulated, as previously reported (Claudel et al., 2018), to similar levels in Arabidopsis and Camelina. Thus there was no obvious link between TuYV accumulation and severity of symptoms, since despite comparable TuYV loads in Arabidopsis and Camelina, disease symptoms were stronger in Camelina than in Arabidopsis (Figure 1; compare the stunted phenotype of TuYV-infected Camelina with the weak phenotype of TuYV-infected Arabidopsis). CaMV loads were higher in Camelina than in Arabidopsis; whether this correlated with symptom expression, was difficult to access because of the severe phenotype in both hosts.”

Quality of RNA and sequence alignment data section

“Therefore, TuYV accumulation as judged by our RNA-seq data might not be accurate and indeed, we found a difference between Western blot results and RNA-seq data (Supplementary Dataset S1). Concerning CaMV, its loads were lower in Arabidopsis than in Camelina (ca. 1.5 times; Supplementary Dataset S1), in line with the Western blot results (Figure 1d).”

Legend to Figure 1

“c-d) Western blot analysis of TuYV CP coat protein (c) and CaMV coat protein P4 (d). On each lane, a total extract from a different plant was loaded. Ponceau staining of the small RuBisCO subunit is shown as a loading control.”

2. The results section is very long and dense. I would suggest breaking it up into more sub-sections, with each heading specifying the main finding of that sub-section. This is especially the case for the heatmap analyses. It is difficult to get the main picture from this section or to be able to revisit certain pathways discussed without some kind of headings as a roadmap.

We shortened the text and divided the section on heatmaps in four subsections, each headed by its own subtitle:

“Photosynthesis-related genes responsive to CaMV and TuYV”

“Carbohydrate pathways genes responsive to CaMV and TuYV”

“Virus defense-related genes responsive to CaMV and TuYV”

“Insect defense-related genes responsive to CaMV and TuYV”

We hope that the section is now easier to read.

3. The authors suggest that functional validation of candidate gene involvement in virus-induced phenotypes and effects on aphid behavior could be a future step. However, it is very hard to pick out what targets are most promising from this analysis because of the density of the text. The authors could provide a summary table relating behavior/fecundity effects to key changes recorded in the transcriptome analysis. This could highlight changes that could underlie the behavior/fecundity response and those that run counter to it, with relevant literature citations.

We agree with the reviewer and added a summary table as suggested (Table 1 in the “Concluding remarks” section). In this table, we have listed some candidate genes which seem to us the most promising for functional analyses. These genes were selected according to potential behavioral/performance effects on aphids by referring to the existing literature.

Reviewer #2 (Comments for the Author):

General Comments:

The main question of this paper is to address if RNA-Seq can tell if plant specific or virus-specific responses to insect feeding with the same vector insect can be detected. There are 6 different conditions including TuYV in Arabidopsis and Camelina, CaMV in Arabidopsis and Camelina and no virus in Arabidopsis and Camelina. There is a lot of data to go through and the authors present the majority of their RNA-Seq data as heat maps of only the relevant conditions, pathways and genes of interest between each of the plants and viruses. There is also inclusion of the general RNA-Seq data points of concern including total numbers of reads, total numbers of upregulated vs downregulated genes, etc. The authors also tested a number of aphid specific conditions such as feeding activity and fecundity. This is a very thorough study which is original in the terms of the breadth of coverage and the number of conditions tested to compare to each other. This reviewer is astounded and excited about the authors undertaking this enormous project and the sheer amount of data that was produced.

Thank you.

Paper is moderately well written, there are a number of small things that could clarify and make the paper more readable. Due to the sheer amount of data presented, it is easy to get lost and distracted in the paper on avenues which can be difficult to follow. Difficult in the sense that it required movement not only through the paper, but also through the extensive amount of supplementary data as well. Authors do not clarify the use of some terms used often in the results section and this could be much improved by simply putting the log2fold in parenthesis next to the transcript in question. For example: Line 481- "The most down-regulated gene in CaMV-infected Arabidopsis, AT3G27690, encoding a protein LHCb2.4, a component of the light harvesting complex, was also strongly repressed in Camelina infected by CaMV". In this example, the log2fold of the Arabidopsis gene could be put in the parenthesis next to that gene and the log2fold of the Camelina could also be put in parenthesis next to that reference. The author uses terms like "strongly downregulated", "strongly repressed", "strongly induced" and so on with no clear definition on what that means to them. Simply including the numbers in the writing when referencing the gene itself would clarify the writing substantially and not require the reader to constantly look back at the supplementary data which is exhausting and time consuming. Considering that the authors do not talk about all of the changes in gene expression- of which there were many- adding the numbers to the ones that were deemed important enough to talk about in the writings will make this paper much more understandable as to what the author means by the use of these general and subjective terms.

We agree that for better visibility it is pertinent to add the log-fold changes to the expression statements directly into the text. Accordingly, we added the numbers to the cited genes.

In the methods, please clarify why the use of 18 day time points were utilized for both viruses for the placement of insects.

We infested plants with aphids at 18 dpi, because at this time point and three days later (when plants were used for RNA-seq, EPG analysis and fecundity experiments) symptoms were visible but not yet necrotic (especially CaMV). We chose to use for both viruses the same time point to exclude aging effects from analysis. A sentence was added to the results and discussion, "Plant phenotype" section: "We used plants at this age and infection state because they displayed symptoms but not yet necrosis."

Was there additional tests done to determine at what stage the virus was in these two plants at this time point? Clarification as what lead to this decision would lead to more information as to how this may impact the genes upregulated or downregulated later. Detail when symptoms first appear on plants of both types of both viruses to determine if the timing of infection is the same or if one may be later or earlier in infection status than the other.

We followed the experimental protocol used by others and us for experiments on CaMV and TuYV infected plants, where 21 dpi are routinely used. See for example Bogaert et al. for TuYV on Arabidopsis (<https://doi.org/10.1007/s10340-019-01082-z>), Chesnais et al. for TuYV and CaMV on Camelina (<https://doi.org/10.1111/1744-7917.12508>), Cecchini et al. (<https://doi.org/10.1093/jxb/49.321.731>) or Dader et al. (<https://doi.org/10.1371/journal.pone.0213087>) for CaMV on Arabidopsis.

The infection status at 21 dpi of the Camelina and Arabidopsis plants infected by CaMV, and of Camelina and Arabidopsis plants infected by TuYV, respectively, is similar. Compared to mock-inoculated controls, CaMV-infected Arabidopsis and Camelina display leaf both mosaics, leaf curling and dwarfism. TuYV-infected Arabidopsis and Camelina show little to moderate dwarfism (especially Camelina), purple leaf coloring (more pronounced on Arabidopsis) and some leaf yellowing (more strongly on older Camelina leaves than on Arabidopsis leaves). All in all, this shows that the plants

were at comparable stages of infection. For more clarity, we added arrows in Figure 1 to show leaf purpling on TuYV-infected Arabidopsis and yellowing on TuYV-infected Camelina.

There is still some disconnect between the EPG data and the RNA-Seq. There is also some disconnect in the fecundity and the RNA-Seq as well. This appears most prominent in the discussion and when these topics are not discussed in the concluding remarks in relation to the RNA-Seq.

We believe it is extremely difficult to connect RNA-seq data directly with EPG or fecundity data, because both aphid parameters do only present a very restricted facet of the whole aphid-plant interaction sphere. Further, transcriptome changes may effect aphid behavior and fecundity indirectly, for example by changed metabolites etc. that are complicated to deduce from RNA-seq data. Therefore, we prefer to leave the manuscript in the present state.

Specific Comments:

Confirmation of the dataset through qRT-PCR resulted in only validation of the Arabidopsis results, no results presented for Camelina. Two of the transcripts for Arabidopsis in TuYV do not agree, however, the authors state it was due to weak expression changes. Did the authors explore alternative splicing transcripts at all? Why were these transcripts selected for validation if two could not pass due to issues? Why was Camelina ignored for validation? Was this due to the number of multiple gene forms in the genome?

Weak expression changes are difficult to measure with qPCR, due to its exponential amplification characteristics.

We selected both high expression and low expression genes to validate the results of bioinformatics analysis of the RNA-seq data. We did not specifically explore any genes with alternatively spliced transcripts.

Indeed, we did not analyze Camelina genes because we experienced difficulties with the multiple gene forms.

In the concluding remarks- Line 796-the authors state that an earlier study looked at only the insect feeding and noticed the response was much less then their current study. Given that the reviewer was expecting these conditions to be tested in this study as a further control, this reviewer feels that should be elaborated on to include any gene changes that were shared between the studies. Since it was in Arabidopsis, the Gene IDs should be the same and could be compared to look at it instead of just mentioning it offhand.

This is a very good point. However, our objective was to distill changes caused by TuYV infection and CaMV infection on infested plants and therefore reflecting DEGs involved in virus-aphid interactions and less so in aphid-plant interactions. Technically, it might be possible to use the raw data from German Martinez's group (Annacondia et al.) to get a list of genes deregulated by aphids only (compare their control with our control), but biologically it is somewhat questionable because the experiments were not done under exactly the same conditions in the two labs. Nonetheless, we compiled a list of oppositely regulated genes in the condition +aphid (Annacondia et al.) vs +aphid+virus (our work) as Supplemental Table S3 and discuss them in the 'Concluding remarks' section. The interpretation is very interesting:

“In this study, we did not compare the contribution of aphid infestation alone to the plant transcriptome. However, recent work (Annacondia et al., 2021) on the transcriptome changes of healthy Arabidopsis plants infested or not with M. persicae for 72 h, identified a limited number of DEGs (265) suggesting that the contribution of aphid infestation to the transcriptome changes in

healthy (and probably also in virus-infected and aphid-infested plants) is minor. Although it is difficult to directly compare their data with ours, we looked for Arabidopsis genes that were upregulated by aphid infestation alone but downregulated by aphid infestation plus virus infection. The rationale was that these genes might reflect viral effects to reduce the host plant's capability to recognize aphid infestation and establish defenses, thus favoring aphid infestation. For TuYV, only one gene that was upregulated by aphid infestation was downregulated by concomitant TuYV-infection (the transcription factor DREB26, AT1G21910). But 36 genes were downregulated by CaMV while upregulated by Myzus infestation alone (none inversely). GO analysis of these genes indicated an enrichment of genes related to 'response to chitin', 'response to salicylic acid', 'response to salt stress', 'response to wounding', 'hormone-mediated signaling pathway', 'defense response to fungus', 'regulation of defense response' and 'signal transduction' (see Supplementary Table S3). This indicates that at least CaMV might dampen plant perception of aphid infestation and defenses against aphids, which might manifest itself in that aphids reach the phloem faster and feed longer on CaMV-infected plants. The fact that Myzus fecundity was lower on these plants, might be explained by the profound changes in other GOs, especially photosynthesis and carbohydrate metabolism, which probably reduce the nutritional value of CaMV-infected plants. Some of these genes could merit further exploration."

Supplementary Table S3: List of plant genes upregulated by aphid infestation alone (extracted from the RNA-seq data by Annacondia et al. (2021)) but downregulated in plants concomitantly aphid-infested and infected by TuYV or CaMV.

Introduction contains all relevant background required to determine the importance of the study, suggest no changes.

Thank you.

References seem well ordered and formatted, suggest no changes.

Thank you.

Scientific method and experiments are well conducted, suggest addressing why Camelina was not confirmed via qRT-PCR. Also include the above mentioned comment about why 18 days was utilized as a time point.

Please see above.

Reviewer comments:

Reviewer #1 (Comments for the Author):

This is a technically sound and interesting paper that presents data on vector behavior/fecundity in relation to infected and healthy hosts, contrasting two very different viruses each infecting two related, but distinct host species. This is followed up with a transcriptomic analysis of a different set of infected and healthy hosts of each species x virus combination in the same stage of growth/disease progression as that used for behavior/fecundity experiments. I have just a few suggestions for improvement of the manuscript.

1. It would be nice to know why virus replication levels were only quantified through the transcriptome data, and not using an RT-qPCR assay that avoids the issue of TuYV not having poly A tails. Although differences were still found, it would be good if these were verified with a more standard diagnostic method for evaluating virus replication in the host.

As an alternative approach to assess virus accumulation we added to Figure 1 Western blots targeting coat proteins of both viruses in both hosts. They show that TuYV accumulates to similar levels in Arabidopsis and Camelina and that CaMV accumulates more strongly in Camelina than in Arabidopsis. The TuYV data are also supported by previous work using RT-qPCR (Claudel et al. 2018, doi 10.3390/ijms19082316).

The methods section, the results sections “Plant Phenotype” and “Quality of RNA and sequence alignment data” and the legend to Figure 1 were accordingly adapted:

Method section:

“Western blotting

Total protein extracts were prepared from leaves, separated by SDS-PAGE and transferred to nitrocellulose as described previously (Chesnais et al., 2021). Western blots were performed using antisera raised against CaMV P4 (Champagne et al., 2004) and TuYV CP (Bruyère et al., 1997). Secondary antibodies were horseradish peroxidase conjugates, and bound antibodies were revealed by enhanced chemiluminescence.”

Plant phenotype section:

“Western blot analysis showed that TuYV accumulated, as previously reported (Claudel et al., 2018), to similar levels in Arabidopsis and Camelina. Thus there was no obvious link between TuYV accumulation and severity of symptoms, since despite comparable TuYV loads in Arabidopsis and Camelina, disease symptoms were stronger in Camelina than in Arabidopsis (Figure 1; compare the stunted phenotype of TuYV-infected Camelina with the weak phenotype of TuYV-infected Arabidopsis). CaMV loads were higher in Camelina than in Arabidopsis; whether this correlated with symptom expression, was difficult to access because of the severe phenotype in both hosts.”

Quality of RNA and sequence alignment data section:

“Therefore, TuYV accumulation as judged by our RNA-seq data might not be accurate and indeed, we found a difference between Western blot results and RNA-seq data (Supplementary Dataset S1). Concerning CaMV, its loads were lower in Arabidopsis than in Camelina (ca. 1.5 times; Supplementary Dataset S1), in line with the Western blot results (Figure 1d).”

Legend to Figure 1:

“c-d) Western blot analysis of TuYV CP coat protein (c) and CaMV coat protein P4 (d). On each lane, a total extract from a different plant was loaded. Ponceau staining of the small RuBisCO subunit is shown as a loading control.”

2. The results section is very long and dense. I would suggest breaking it up into more sub-sections, with each heading specifying the main finding of that sub-section. This is especially the case for the heatmap analyses. It is difficult to get the main picture from this section or to be able to revisit certain pathways discussed without some kind of headings as a roadmap.

We shortened the text and divided the section on heatmaps in four subsections, each headed by its own subtitle:

“Photosynthesis-related genes responsive to CaMV and TuYV”

“Carbohydrate pathways genes responsive to CaMV and TuYV”

“Virus defense-related genes responsive to CaMV and TuYV”

“Insect defense-related genes responsive to CaMV and TuYV”

We hope that the section is now easier to read.

3. The authors suggest that functional validation of candidate gene involvement in virus-induced phenotypes and effects on aphid behavior could be a future step. However, it is very hard to pick out what targets are most promising from this analysis because of the density of the text. The authors could provide a summary table relating behavior/fecundity effects to key changes recorded in the transcriptome analysis. This could highlight changes that could underlie the behavior/fecundity response and those that run counter to it, with relevant literature citations.

We agree with the reviewer and added a summary table as suggested (Table 1 in the “Concluding remarks” section). In this table, we have listed some candidate genes which seem to us the most promising for functional analyses. These genes were selected according to potential behavioral/performance effects on aphids by referring to the existing literature.

Reviewer #2 (Comments for the Author):

General Comments:

The main question of this paper is to address if RNA-Seq can tell if plant specific or virus-specific responses to insect feeding with the same vector insect can be detected. There are 6 different conditions including TuYV in Arabidopsis and Camelina, CaMV in Arabidopsis and Camelina and no virus in Arabidopsis and Camelina. There is a lot of data to go through and the authors present the majority of their RNA-Seq data as heat maps of only the relevant conditions, pathways and genes of interest between each of the plants and viruses. There is also inclusion of the general RNA-Seq data points of concern including total numbers of reads, total numbers of upregulated vs downregulated genes, etc. The authors also tested a number of aphid specific conditions such as feeding activity and fecundity. This is a very thorough study which is original in the terms of the breadth of coverage and the number of conditions tested to compare to each other. This reviewer is astounded and excited about the authors undertaking this enormous project and the sheer amount of data that was produced.

Thank you.

Paper is moderately well written, there are a number of small things that could clarify and make the paper more readable. Due to the sheer amount of data presented, it is easy to get lost and distracted in the paper on avenues which can be difficult to follow. Difficult in the sense that it required movement not only through the paper, but also through the extensive amount of supplementary data as well. Authors do not clarify the use of some terms used often in the results section and this could be much improved by simply putting the log₂fold in parenthesis next to the transcript in question. For example: Line 481- "The most down-regulated gene in CaMV-infected Arabidopsis, AT3G27690, encoding a protein LHCb2.4, a component of the light harvesting complex, was also strongly repressed in Camelina infected by CaMV". In this example, the log₂fold of the Arabidopsis gene could be put in the parenthesis next to that gene and the log₂fold of the Camelina could also be put in parenthesis next to that reference. The author uses terms like "strongly downregulated", "strongly repressed", "strongly induced" and so on with no clear definition on what that means to them. Simply including the numbers in the writing when referencing the gene itself would clarify the writing substantially and not require the reader to constantly look back at the supplementary data which is exhausting and time consuming. Considering that the authors do not talk about all of the changes in gene expression- of which there were many- adding the numbers to the ones that were deemed important enough to talk about in the writings will make this paper much more understandable as to what the author means by the use of these general and subjective terms.

We agree that for better visibility it is pertinent to add the log-fold changes to the expression statements directly into the text. Accordingly, we added the numbers to the cited genes.

In the methods, please clarify why the use of 18 day time points were utilized for both viruses for the placement of insects.

We infested plants with aphids at 18 dpi, because at this time point and three days later (when plants were used for RNA-seq, EPG analysis and fecundity experiments) symptoms were visible but not yet necrotic (especially CaMV). We chose to use for both viruses the same time point to exclude aging effects from analysis. A sentence was added to the results and discussion, "Plant phenotype" section: "We used plants at this age and infection state because they displayed symptoms but not yet necrosis."

Was there additional tests done to determine at what stage the virus was in these two plants at this time point? Clarification as what lead to this decision would lead to more information as to how this may impact the genes upregulated or downregulated later. Detail when symptoms first appear on plants of both types of both viruses to determine if the timing of infection is the same or if one may be later or earlier in infection status than the other.

We followed the experimental protocol used by others and us for experiments on CaMV and TuYV infected plants, where 21 dpi are routinely used. See for example Bogaert et al. for TuYV on Arabidopsis (<https://doi.org/10.1007/s10340-019-01082-z>), Chesnais et al. for TuYV and CaMV on Camelina (<https://doi.org/10.1111/1744-7917.12508>), Cecchini et al. (<https://doi.org/10.1093/jxb/49.321.731>) or Dader et al. (<https://doi.org/10.1371/journal.pone.0213087>) for CaMV on Arabidopsis.

The infection status at 21 dpi of the Camelina and Arabidopsis plants infected by CaMV, and of Camelina and Arabidopsis plants infected by TuYV, respectively, is similar. Compared to mock-inoculated controls, CaMV-infected Arabidopsis and Camelina display leaf both mosaics, leaf curling and dwarfism. TuYV-infected Arabidopsis and Camelina show little to moderate dwarfism (especially Camelina), purple leaf coloring (more pronounced on Arabidopsis) and some leaf yellowing (more strongly on older Camelina leaves than on Arabidopsis leaves). All in all, this shows that the plants

were at comparable stages of infection. For more clarity, we added arrows in Figure 1 to show leaf purpling on TuYV-infected Arabidopsis and yellowing on TuYV-infected Camelina.

There is still some disconnect between the EPG data and the RNA-Seq. There is also some disconnect in the fecundity and the RNA-Seq as well. This appears most prominent in the discussion and when these topics are not discussed in the concluding remarks in relation to the RNA-Seq.

We believe it is extremely difficult to connect RNA-seq data directly with EPG or fecundity data, because both aphid parameters do only present a very restricted facet of the whole aphid-plant interaction sphere. Further, transcriptome changes may effect aphid behavior and fecundity indirectly, for example by changed metabolites etc. that are complicated to deduce from RNA-seq data. Therefore, we prefer to leave the manuscript in the present state.

Specific Comments:

Confirmation of the dataset through qRT-PCR resulted in only validation of the Arabidopsis results, no results presented for Camelina. Two of the transcripts for Arabidopsis in TuYV do not agree, however, the authors state it was due to weak expression changes. Did the authors explore alternative splicing transcripts at all? Why were these transcripts selected for validation if two could not pass due to issues? Why was Camelina ignored for validation? Was this due to the number of multiple gene forms in the genome?

Weak expression changes are difficult to measure with qPCR, due to its exponential amplification characteristics.

We selected both high expression and low expression genes to validate the results of bioinformatics analysis of the RNA-seq data. We did not specifically explore any genes with alternatively spliced transcripts.

Indeed, we did not analyze Camelina genes because we experienced difficulties with the multiple gene forms.

In the concluding remarks- Line 796-the authors state that an earlier study looked at only the insect feeding and noticed the response was much less then their current study. Given that the reviewer was expecting these conditions to be tested in this study as a further control, this reviewer feels that should be elaborated on to include any gene changes that were shared between the studies. Since it was in Arabidopsis, the Gene IDs should be the same and could be compared to look at it instead of just mentioning it offhand.

This is a very good point. However, our objective was to distill changes caused by TuYV infection and CaMV infection on infested plants and therefore reflecting DEGs involved in virus-aphid interactions and less so in aphid-plant interactions. Technically, it might be possible to use the raw data from German Martinez's group (Annacondia et al.) to get a list of genes deregulated by aphids only (compare their control with our control), but biologically it is somewhat questionable because the experiments were not done under exactly the same conditions in the two labs. Nonetheless, we compiled a list of oppositely regulated genes in the condition +aphid (Annacondia et al.) vs +aphid+virus (our work) as Supplemental Table S3 and discuss them in the 'Concluding remarks' section. The interpretation is very interesting:

“In this study, we did not compare the contribution of aphid infestation alone to the plant transcriptome. However, recent work (Annacondia et al., 2021) on the transcriptome changes of healthy Arabidopsis plants infested or not with M. persicae for 72 h, identified a limited number of DEGs (265) suggesting that the contribution of aphid infestation to the transcriptome changes in

healthy (and probably also in virus-infected and aphid-infested plants) is minor. Although it is difficult to directly compare their data with ours, we looked for Arabidopsis genes that were upregulated by aphid infestation alone but downregulated by aphid infestation plus virus infection. The rationale was that these genes might reflect viral effects to reduce the host plant's capability to recognize aphid infestation and establish defenses, thus favoring aphid infestation. For TuYV, only one gene that was upregulated by aphid infestation was downregulated by concomitant TuYV-infection (the transcription factor DREB26, AT1G21910). But 36 genes were downregulated by CaMV while upregulated by Myzus infestation alone (none inversely). GO analysis of these genes indicated an enrichment of genes related to 'response to chitin', 'response to salicylic acid', 'response to salt stress', 'response to wounding', 'hormone-mediated signaling pathway', 'defense response to fungus', 'regulation of defense response' and 'signal transduction' (see Supplementary Table S3). This indicates that at least CaMV might dampen plant perception of aphid infestation and defenses against aphids, which might manifest itself in that aphids reach the phloem faster and feed longer on CaMV-infected plants. The fact that Myzus fecundity was lower on these plants, might be explained by the profound changes in other GOs, especially photosynthesis and carbohydrate metabolism, which probably reduce the nutritional value of CaMV-infected plants. Some of these genes could merit further exploration."

Legend to Supplementary Table S3:

"Supplementary Table S3: List of plant genes upregulated by aphid infestation alone (extracted from the RNA-seq data by Annacondia et al. (2021)) but downregulated in plants concomitantly aphid-infested and infected by TuYV or CaMV."

Introduction contains all relevant background required to determine the importance of the study, suggest no changes.

Thank you.

References seem well ordered and formatted, suggest no changes.

Thank you.

Scientific method and experiments are well conducted, suggest addressing why Camelina was not confirmed via qRT-PCR. Also include the above mentioned comment about why 18 days was utilized as a time point.

Please see above.

June 19, 2022

Dr. Martin Drucker
SVQV INRAE Centre Grand Est Colmar
Université Strasbourg
Colmar
France

Re: Spectrum00136-22R1 (Comparative plant transcriptome profiling of Arabidopsis and Camelina infested with Myzus persicae aphids acquiring circulative and non-circulative viruses reveals virus- and plant-specific alterations relevant to aphid feeding behavior and transmission)

Dear Dr. Martin Drucker:

Your manuscript has been accepted, and I am forwarding it to the ASM Journals Department for publication. You will be notified when your proofs are ready to be viewed.

Sincerely,

Lindsey Burbank
Editor, Microbiology Spectrum

Journals Department
Supplemental Dataset S1: Accept
Supplemental Dataset S3: Accept
Supplemental Dataset S2: Accept
Supplemental Material: Accept